# GRADIENT CORRELATION IS A KEY INGREDIENT TO ACCELERATE SGD WITH MOMENTUM

**Julien Hermant**[*1]**, Marien Renaud**[1]**, Jean-François Aujol**[1]**, Charles Dossal**[2]**, Aude Rondepierre**[2]
1. Univ. Bordeaux, Bordeaux INP, CNRS, IMB, UMR 5251, F-33400 Talence, France
2. IMT, Univ. Toulouse, INSA Toulouse, Toulouse, France

## ABSTRACT

Empirically, it has been observed that adding momentum to Stochastic Gradient Descent (SGD) accelerates the convergence of the algorithm. However, the literature has been rather pessimistic, even in the case of convex functions, about the possibility of theoretically proving this observation. We investigate the possibility of obtaining accelerated convergence of the Stochastic Nesterov Accelerated Gradient (SNAG), a momentum-based version of SGD, when minimizing a sum of functions in a convex setting. We demonstrate that the average correlation between gradients allows to verify the strong growth condition, which is the key ingredient to obtain acceleration with SNAG. Numerical experiments, both in linear regression and deep neural network optimization, confirm in practice our theoretical results.

## 1 INTRODUCTION

Supervised machine learning tasks can often be formulated as the following optimization problem (Hastie et al., 2009):

$$f^* = \min_{x \in \mathbb{R}^d} f(x), \text{ with } f(x) := \frac{1}{N} \sum_{i=1}^{N} f_i(x) \tag{FS}$$

where for all $i \in \{1, \ldots, N\}$, $f_i : \mathbb{R}^d \to \mathbb{R}$ is associated to one data. As there is often no closed form solution to problem (FS), optimization algorithms are commonly used. Considering high dimensional problems, first order algorithms such as gradient descent, namely algorithms that make use of the gradient information, are popular due to their relative cheapness, *e.g.* compared to second order methods which involve the computation of the Hessian matrix. In the case of problem (FS), the gradient itself may be computationally heavy to obtain when $N$ is large, *i.e.* for large datasets. This is the reason why, instead of the exact gradient, practitioners rather use an average of a subsampled set of several $\nabla f_i$, resulting in algorithms such as Stochastic Gradient Descent (SGD) (Robbins & Monro, 1951) or ADAM algorithm (Kingma & Ba, 2015). Although an estimation of the gradient is used, SGD performs well in practice (Goyal et al., 2017; Schmidt et al., 2021; Renaud et al., 2024).

**Interpolation** One of the key points that explain the good performance of SGD is that large machine learning models, such as over parameterized neural networks, are generically able to perfectly fit the learning data (Cooper, 2021; Allen-Zhu et al., 2019; Nakkiran et al., 2021; Zhang et al., 2021). From an optimization point of view, this fitting phenomenon translates into an **interpolation** phenomenon (Assumption 1).

**Assumption 1** (Interpolation). $\exists x^* \in \arg \min f, \forall 1 \leq i \leq N, \ x^* \in \arg \min f_i$.

Strikingly, under this assumption, theoretical results show that SGD performs as well as deterministic gradient descent (Ma et al., 2018; Gower et al., 2019; 2021).

**Momentum algorithms: hopes and disappointments** Within the convex optimization realm, it is well known that a first order momentum algorithm, named Nesterov Accelerated Gradient (NAG),

---

[*]julien.hermant@math.u-bordeaux.fr

outperforms the gradient descent in term of convergence speed (Nesterov, 1983; 2018). A natural concern is thus whether, assuming interpolation and convexity, a stochastic version of NAG, **Stochastic Nesterov Accelerated Gradient** (SNAG), can be faster than SGD. Unfortunately, existing works has expressed skepticism about the possibility of acceleration via momentum algorithms, even assuming interpolation (Assumption 1). Devolder et al. (2014); Aujol & Dossal (2015) indicate that momentum algorithms are very sensitive to errors on the gradient, due to error accumulation. Also, the choice of parameters that offers acceleration in the deterministic case can make SNAG diverge (Kidambi et al. (2018); Assran & Rabbat (2020); Ganesh et al. (2023)).

**What keeps us hopeful**    Firstly, in the case of linear regression, depending on the data, SNAG can accelerate over SGD (Jain et al., 2018; Liu & Belkin, 2020; Varre & Flammarion, 2022). Unfortunately the methods used in those works are hardly generalizable outside of the linear regression case. On the other hand, in a convex setting, Vaswani et al. (2019) show that under an assumption over the gradient noise, named the **Strong Growth Condition** (SGC), SNAG can be stabilized, and it could accelerate over SGD. However, it is not clear which functions satisfy this assumption, and in which cases acceleration occurs.

**On the convexity assumption**    For many machine learning models, such as neural networks, the associated loss function is not convex (Li et al., 2018).

However, even in the convex setting, there is still work to do concerning the possibility of accelerating SGD with SNAG. For example, up to our knowledge, characterizing convex smooth functions that satisfy SGC has not been addressed yet.

Finally, note that our core results about gradient correlation (Propositions 1-2) do not assume convexity, and thus could be used in future works beyond the convex setting.



**Can SNAG accelerate over SGD in a convex setting ?**



**Contributions**    **(i)** We give a new characterization of the Strong Growth Condition (SGC) constant by using the correlation between gradients, quantified by RACOGA (Propositions 1-2), and we exploit this link to study the efficiency of SNAG. **(ii)** Using our framework, we study the theoretical impact of batch size on the algorithm performance, depending on the correlation between gradients (Theorem 5). **(iii)** We complete convergence results of Vaswani et al. (2019); Gupta et al. (2023) with new almost sure convergence rates (Theorem 4). **(iv)** We provide numerical experiments that show that RACOGA is a key ingredient to have good performances of SNAG compared to SGD.

## 2    BACKGROUND

For a function $f : \mathbb{R}^d \to \mathbb{R}$, continuously differentiable, we introduce the following definitions.

**Definition 1.** *Let $L > 0$. $f : \mathbb{R}^d \to \mathbb{R}$ is **L-smooth** if $\nabla f$ is L-Lipschitz.*

Definition 1 implies that $\forall x, y \in \mathbb{R}^d$, $f(x) \leq f(y) + \langle \nabla f(y), x - y \rangle + \frac{L}{2}\|x - y\|^2$ (Nesterov, 2018) and it ensures that the curvature of the function $f$ is upper-bounded by $L$.

**Definition 2.** *$f : \mathbb{R}^d \to \mathbb{R}$ is $\mu$-**strongly convex** if there exists $\mu > 0$ such that: $\forall x, y \in \mathbb{R}^d$, $f(x) \geq f(y) + \langle \nabla f(y), x - y \rangle + \frac{\mu}{2}\|x - y\|^2$. $f : \mathbb{R}^d \to \mathbb{R}$ is **convex** if it verifies this property with $\mu = 0$.*

Definition 2 implies that the curvature of the function $f$ is lower-bounded by $\mu \geq 0$. Convex functions are very convenient for optimization and widely studied (Nesterov, 2018; Beck, 2017). For instance, a critical point of a convex function is the global minimum of this function.

**Assumption 2** (smoothness). *Each $f_i$ in (FS) is $L_i$-smooth. We note $L_{(K)} := \max_B \frac{1}{K} \sum_{i \in B} L_i$ where $B \subset \{1, \ldots, N\}$, $Card(B) = K$ and we note $L_{\max} := L_{(1)} = \max_{1 \leq i \leq N} L_i$.*

Assumption 2 implies that $f$ is $L$-smooth with $L \leq \frac{1}{N} \sum_{i=1}^{N} L_i \leq L_{(K)} \leq L_{max}$, and will be used for the convergence results for SGD (Theorems 1-2).

**SGD** The stochastic gradient descent (Algorithm 1) is widely used despite its simplicity. It can be viewed as a gradient descent where the exact gradient is replaced by a batch estimator

$$\tilde{\nabla}_K(x) := \frac{1}{K} \sum_{i \in B} \nabla f_i(x), \tag{1}$$

where $B$ is a batch of indices of size $K$ sampled uniformly in $\{B \subset \{1, \ldots, N\} \mid \mathrm{Card}(B) = K\}$. Note that $\tilde{\nabla}_K(x)$ is a random variable depending on $K$, $f$ and $x$.

---

**Algorithm 1** Stochastic Gradient Descent (SGD)

---

1: **input:** $x_0 \in \mathbb{R}^d$, $s > 0$
2: **for** $n = 0, 1, \ldots, n_{it} - 1$ **do**
3: $\quad x_{n+1} = x_n - s\tilde{\nabla}_K(x_n)$
4: **end for**
5: **output:** $x_{n_{it}}$

---

**Algorithm 2** Stochastic Nesterov Accelerated Gradient (SNAG)

---

1: **input:** $x_0 = z_0 \in \mathbb{R}^d$, $s > 0$, $\beta \in [0, 1]$, $(\alpha_n)_{n \in \mathbf{N}} \in [0, 1]^{\mathbf{N}}$, $(\eta_n)_{n \in \mathbf{N}} \in \mathbf{R}_+^{\mathbf{N}}$
2: **for** $n = 0, 1, \ldots, n_{it} - 1$ **do**
3: $\quad y_n = \alpha_n x_n + (1 - \alpha_n) z_n$
4: $\quad x_{n+1} = y_n - s\tilde{\nabla}_K(y_n)$
5: $\quad z_{n+1} = \beta z_n + (1 - \beta) y_n - \eta_n \tilde{\nabla}_K(y_n)$
6: **end for**
7: **output:** $x_{n_{it}}$

---

**SNAG** The Nesterov accelerated gradient algorithm (Algorithm 7 in Appendix B.1) allows to achieve faster convergence than gradient descent when considering $L$-smooth functions that are convex or strongly convex, see Nesterov (1983; 2018). Intuitively, a momentum mechanism accelerates the gradient descent. As proposed in Nesterov (2012), a stochastic version of the Nesterov accelerated gradient algorithm can be developed, see Algorithm 2. Note that there exists several ways to write it (see Appendix B.2).

**Strong Growth Condition** To our knowledge, this assumption was introduced in Polyak (1987), and further used by Cevher & Vu (2019) as a relaxation of the maximal strong growth condition (Tseng, 1998; Solodov, 1998).

**Definition 3.** *The function $f$, with a gradient estimator $\tilde{\nabla}_K$ (Equation 1), is said to verify the **Strong Growth Condition** if there exists $\rho_K \geq 1$ such that*

$$\forall x \in \mathbb{R}^d, \ \mathbb{E}\left[\|\tilde{\nabla}_K(x)\|^2\right] \leq \rho_K \|\nabla f(x)\|^2. \tag{SGC}$$

$\rho_K$ quantifies the amount of noise: the larger $\rho_K$, the higher the noise. In some sense, the strong growth condition allows to replace the norm of the stochastic gradient by the norm of the exact gradient, up to a degrading constant $\rho_K$. We will see in Section 3 that it allows to recover similar convergence results as for the deterministic case.

**Example 1.** *Vaswani et al. (2019) show that if the function $f$ is strongly convex and verifies Assumptions 1 and 2, then $f$ verifies the strong growth condition with $\rho_K = \frac{L_{(K)}}{\mu}$.*

**Remark 1.** *The SGC implies interpolation (Assumption 1) if each $f_i$ is convex, see Remark 11 in Appendix H.3. However in the following results (Theorems 1-4), as we will only assume that the sum of $f_i$ is convex, SGC will not enforce interpolation. It will imply instead that minimizers of $f$ are critical points of all $f_i$.*

**Remark 2.** *Considering linear regression, one can choose functions such that the SGC is verified only for arbitrary large values of $\rho_K$ (see Appendix D.1). Worse, if we discard the convexity assumption, then one can construct examples such that $\rho_K$ does not exist (see Appendix D.2). Finding classes of functions such that the SGC is verified for an interesting $\rho_K$ is thus not an obvious task.*

## 3 Convergence speed of SNAG and comparison with SGD

In this section, we present convergence results for SNAG (Algorithm 2) under the Strong Growth Condition (SGC). Before doing so, we introduce convergence results for SGD (Algorithm 1) in order

| Algorithm | Assumption over $f$ | Convergence | |
|---|---|---|---|
| SGD | convex | $\mathbb{E}, \varepsilon$ solution | $\mathcal{O}\left(\frac{L_{(K)}}{\varepsilon}\right)$, Thm 1 |
| | strongly convex | $\mathbb{E}, \varepsilon$ solution | $\mathcal{O}\left(\frac{L_{(K)}}{\mu}\log\left(\frac{1}{\varepsilon}\right)\right)$, Thm 1 |
| | convex | a.s., c.r | $o\left(\frac{1}{n}\right)$, Thm 2 |
| | strongly convex | a.s., c.r | $o\left((1-\frac{\mu}{L}+\varepsilon')^n\right)$, Thm 2 |
| SNAG | Convex | $\mathbb{E}, \varepsilon$ solution | $\mathcal{O}\left(\rho_K\sqrt{\frac{L}{\varepsilon}}\right)$, Thm 3 |
| | Strongly Convex | $\mathbb{E}, \varepsilon$ solution | $\mathcal{O}\left(\rho_K\sqrt{\frac{L}{\mu}}\log\left(\frac{1}{\varepsilon}\right)\right)$, Thm 3 |
| | Convex | a.s., c.r | $o\left(\frac{1}{n^2}\right)$, Thm 4 |
| | Strongly | a.s., c.r | $o\left((1-\frac{1}{\rho_K}\sqrt{\frac{\mu}{L}}+\varepsilon')^n\right)$, Thm 4 |

Table 1: Summary of all the convergence results presented in Section 3. Results stated as $\epsilon$-solution refer to convergence results of the form of Equation (2). *c.r.* stands for *convergence rate*. These results are stated as an upper bound of the form $f(x_n) - f^* = \mathcal{O}(\psi_n)$, where $\psi_n$ is a sequence decreasing to 0.

to compare the performance of these two algorithms. All the results introduced in this section are summarized in Table 1.

For $\varepsilon > 0$, we say that an algorithm $\{x_n\}_n$ reaches an $\varepsilon$-precision at rank $\nu_\varepsilon \in \mathbb{R}_+$ if

$$\forall n \geq \nu_\varepsilon, \ \mathbb{E}\left[f(x_n) - f^*\right] \leq \varepsilon. \tag{2}$$

We denote by $\Omega$ the set of realization of the noise. We say that an algorithm $\{x_n\}_n$ converges almost surely with a rate negligible compared to $a_n \in \mathbb{R}_{++}^{\mathbb{N}}$, denoted by $f(x_n) - f^* \overset{a.s.}{=} o(a_n)$, if and only if $\exists A \subset \Omega$, such that $\mathbb{P}(A) = 1$ and $\forall \omega \in A, \forall \epsilon > 0, \exists n_0 \in \mathbb{N}$, such that $\forall n \geq n_0$,

$$|f(x_n(\omega)) - f^*| \leq \epsilon a_n. \tag{3}$$

## 3.1 CONVERGENCE RESULTS FOR SGD

First, we state convergence results of SGD (Algorithm 1), in expectation and almost surely. The two following theorems are variations of Gower et al. (2019) and Gower et al. (2021) (results in expectation) and Sebbouh et al. (2021) (result almost surely). The difference is that our setting does not assume the convexity of each $f_i$ in (FS), but rather only the convexity of the sum.

**Theorem 1.** *Under Assumptions 1 and 2, SGD (Algorithm 1) guarantees to reach an $\varepsilon$-precision (2) at the following iterations:*

• *If $f$ is convex, $s = \frac{1}{2L_{(K)}}$,*

$$n \geq 2\frac{L_{(K)}}{\varepsilon}\|x_0 - x^*\|^2. \tag{4}$$

• *If $f$ is $\mu$-strongly convex, $s = \frac{1}{L_{(K)}}$,*

$$n \geq 2\frac{L_{(K)}}{\mu}\log\left(2\frac{f(x_0) - f^*}{\mu\varepsilon}\right). \tag{5}$$

For the convex case, the bound is of the order $\mathcal{O}\left(\frac{L_{(K)}}{\varepsilon}\right)$, while for the strongly convex case the key factor is $\frac{L_{(K)}}{\mu}$ that may be very large for ill conditioned problems. These results are very similar to those obtained in a deterministic setting, see Appendix B.1.

Additionally, almost sure convergence gives guarantees that apply to a single run of SGD.

**Theorem 2.** *Under Assumptions 1 and 2, SGD (Algorithm 1) guarantees, in the sense of (3), the following asymptotic results, :*

- *If $f$ is convex, $s = \frac{1}{2L_{(K)}}$,*

$$f(\overline{x}_n) - f^* \stackrel{a.s.}{=} o\left(\frac{1}{n}\right). \tag{6}$$

- *If $f$ is $\mu$-strongly convex, $s = \frac{1}{L_{(K)}}$,*

$$f(x_n) - f^* \stackrel{a.s.}{=} o\left((q + \varepsilon')^n\right), \tag{7}$$

*for all $\varepsilon' > 0$, where $q := 1 - \frac{\mu}{L_{(K)}}$, $\overline{x}_0 = x_0$ and $\overline{x}_{n+1} = \frac{2}{n+1}x_n + \frac{n-1}{n+1}\overline{x}_n$.*

Note that in the convex case, there is a need of averaging the trajectory along iterations. Proofs of Theorem 1 and Theorem 2 are in Appendix E.1.

## 3.2 CONVERGENCE IN EXPECTATION FOR SNAG

We now state the convergence speed of SNAG in expectation under the Strong Growth Condition (SGC) and we compare it with the convergence speed of SGD.

**Theorem 3.** *Assume $f$ is L-smooth, and that $\tilde{\nabla}_K$ verifies the SGC for $\rho_K \geq 1$. Then the SNAG (Algorithm 2) allows to reach an $\varepsilon$-precision (2) at the following iterations:*
- *If $f$ is convex, $s = \frac{1}{L\rho_K}$, $\eta_n = \frac{1}{L\rho_K^2}\frac{n+1}{2}$, $\beta = 1$, $\alpha_n = \frac{\frac{n^2}{n+1}}{2+\frac{n^2}{n+1}}$,*

$$n \geq \rho_K\sqrt{\frac{2L}{\varepsilon}}\|x_0 - x^*\|. \tag{8}$$

- *If $f$ is $\mu$-strongly convex, $s = \frac{1}{L\rho_K}$, $\eta_n = \eta = \frac{1}{\rho_K\sqrt{\mu L}}$, $\beta = 1 - \frac{1}{\rho_K}\sqrt{\frac{\mu}{L}}$, $\alpha_n = \alpha = \frac{1}{1+\frac{1}{\rho_K}\sqrt{\frac{\mu}{L}}}$,*

$$n \geq \rho_K\sqrt{\frac{L}{\mu}}\log\left(2\frac{f(x_0)-f^*}{\varepsilon}\right). \tag{9}$$

Theorem 3 is a variation of Vaswani et al. (2019); Gupta et al. (2023), see Appendix C.3. Indeed, our proof (Appendix G) leads to the same convergence result as Vaswani et al. (2019), although resulting in a slightly simpler formulation of the algorithm, as we do not have intermediate sequences of parameters, see Appendix C. Note that we only use the $L$-smoothness of $f$ and the SGC instead of Assumptions 1-2 because SGC allows us to make weaker assumptions.

Theorem 3 indicates that the performance degrades linearly with $\rho_K$. For the special case $\rho_K = 1$, bounds of Theorem 3 are the same as in the deterministic case (see Appendix B.1).

**Remark 3.** *According to Theorem 3 and Theorem 1, SNAG (Algorithm 2) is faster than SGD (Algorithm 1) when $\rho_K$ is small enough, more precisely when*

- $\rho_K < \sqrt{\frac{2L_{(K)}^2}{\varepsilon L}}\|x_0 - x^*\|$ *if $f$ convex.*

- $\rho_K < 2\sqrt{\frac{L_{(K)}^2}{\mu L}}$ *if $f$ $\mu$-strongly convex, ignoring the differences between logarithm terms.*

*If $f$ is convex and the required precision $\varepsilon$ small enough, SNAG is faster than SGD. It is not necessarily the case if $f$ is $\mu$-strongly convex, as the dependence on $\varepsilon$ disappears. In particular, the bound $\rho_K \leq \frac{L_{(K)}}{\mu}$ offered by strong convexity (see Example 1) does not guarantee acceleration.*

**Remark 4.** *In our comparison, we neither considered a convergence result for SGD (Algorithm 1) that assumes the SGC, nor considered a result for SNAG (Algorithm 2) that does not assume SGC. In both cases, doing so would lead to misleading comparisons, see summary in Remark 9.*

## 3.3 ALMOST SURE CONVERGENCE FOR SNAG

We provide new asymptotic almost sure convergence results for SNAG (Algorithm 2). Almost sure convergence has already been addressed in Gupta et al. (2023) without convergence rates.

**Theorem 4.** *Assume $f$ is $L$-smooth , and that $\tilde{\nabla}_K$ verifies the SGC for $\rho_K \geq 1$. Then SNAG (Algorithm 2) guarantees, in the sense of (3), the following asymptotic results:*

• *If $f$ is convex, $s = \frac{1}{\rho_K L}$, $\eta_n = \frac{1}{4}\frac{n^2}{n+1}$, $\beta = 1$, $\alpha_n = \frac{\frac{n^2}{n+1}}{4+\frac{n^2}{n+1}}$,*

$$f(x_n) - f^* \stackrel{a.s.}{=} o\left(\frac{1}{n^2}\right). \tag{10}$$

• *If $f$ is $\mu$-strongly convex, $s = \frac{1}{\rho_K L}$, $\eta_n = \eta = \frac{1}{\rho_K \sqrt{\mu L}}$, $\beta = 1 - \frac{1}{\rho_K}\sqrt{\frac{\mu}{L}}$, $\alpha_n = \alpha = \frac{1}{1+\frac{1}{\rho_K}\sqrt{\frac{\mu}{L}}}$,*

$$f(x_n) - f^* \stackrel{a.s.}{=} o\left((q+\varepsilon')^n\right) \tag{11}$$

*for all $\varepsilon' > 0$, where $q := 1 - \frac{1}{\rho_K}\sqrt{\frac{\mu}{L}}$.*

See proofs in Appendix G. These bounds are asymptotically better than the finite time bounds, with $o(\frac{1}{n^2})$ compared to $\mathcal{O}(\frac{1}{n^2})$ (Theorem 3) for instance in the convex setting. A similar asymptotic speedup phenomenon happens in the deterministic setting (Attouch & Peypouquet, 2016).

**Remark 5.** *Theorem 4 states that in the convex case, the parameter $\rho_K$ has a negligible impact on the asymptotic convergence, and thus SNAG always asymptotically outperforms SGD. For the strongly convex case, we need to ensure that $\rho_K < \sqrt{L_{(K)}^2/\mu L}$ to have SNAG faster than SGD.*

The possibility of acceleration of SNAG over SGD is highly depending on the SGC constant $\rho_K$. We need to investigate for a fine characterization of $\rho_K$ to ensure acceleration in realistic contexts.

## 4 CHARACTERIZING CONVERGENCE WITH STRONG GROWTH CONDITION AND GRADIENT CORRELATION

Although general bounds on the constant $\rho_K$ are difficult to obtain (see Remark 2), Example 2 shows that the characterization of the SGC constant given in Example 1 can be improved.

**Example 2** (Motivating example). *Consider the function $f(x) = \frac{1}{2}\left(\frac{\mu}{2}\langle e_1, x\rangle^2 + \frac{L}{2}\langle e_2, x\rangle^2\right)$, with $0 < \mu < L$ and $e_1, e_2$ standard basis vectors. This function satisfies Assumption 1, Assumption 2 with $L_{\max} = L$, and it is $\frac{\mu}{2}$-strongly convex. Following Example 1, $f$ satisfies the SGC with $\rho_1 = 2\frac{L}{\mu}$, which can be arbitrary large. However, by developing $\|\nabla f(x)\|^2$, we get that the SGC is actually verified for $\rho_1 = 2 < 2\frac{L}{\mu}$.*

Example 2 motivates to seek for new, eventually tighter, characterizations of the SGC constant.

### 4.1 AVERAGE POSITIVE CORRELATION CONDITION

In this section, we show how we can exploit the finite sum structure of $f$ to exhibit a condition that, if verified, allows for a new computation of $\rho_K$.

**Proposition 1.** *Considering batches of size $K$, we have*

$$\|\nabla f(x)\|^2 = \frac{K}{N}\mathbb{E}\left[\|\tilde{\nabla}_K(x)\|^2\right] + \frac{2}{N^2}\frac{N-K}{N-1}\sum_{1\leq i<j\leq N}\langle\nabla f_i(x), \nabla f_j(x)\rangle. \tag{12}$$

Proposition 1 is proved in Appendix H.1. Without any assumption on $f$, Proposition 1 splits the norm of $\nabla f$ into two terms. One relies on the gradient estimator, while the other one involves the average correlation of gradients. From Proposition 1, we deduce the following consequence.

**Corollary 1.** *Considering batches of size $K$, $f$ satisfies the SGC with $\rho_K = \frac{N}{K}$ if its gradients are, on average, positively correlated, i.e. if we have*

$$\forall x \in \mathbb{R}^d, \quad \sum_{1\leq i<j\leq N}\langle\nabla f_i(x), \nabla f_j(x)\rangle \geq 0. \tag{PosCorr}$$

Using condition PosCorr, Corollary 1 ensures that $f$ verifies a SGC for a constant $\rho_K$ only depending on $N$ and the batch size $K$, and not on geometrical parameters of $f$, *e.g.* $\mu$ or $L$.

**Example 3.** *Assume $f_i(x) = \Phi(\langle x, u_i \rangle)$, for some $\Phi : \mathbb{R} \to \mathbb{R}$ that are non necessarily convex, and some orthogonal basis $\{u_i\}_i$. We have $\nabla f_i(x) = \Phi'(\langle x, u_i \rangle) u_i$. Then, condition PosCorr is verified, and $f$ satisfies the SGC with $\rho_K = \frac{N}{K}$. Note that the upper-bound $\rho_K \leq \frac{L_{(K)}}{\mu}$ given in Example 1 can be arbitrary large independently of $N$ and $K$ (see Example 2), meaning that eventually $\frac{N}{K} \ll \frac{L_{(K)}}{\mu}$. Thus, the new upper-bound for $\rho_K$ (Corollary 1) can be much tighter, resulting in improved convergence bounds for SNAG (Theorem 3-4).*

In some cases, condition PosCorr could be too restrictive (see Appendix I). In the following section, we show how to ensure the SGC with a relaxed version of PosCorr, named RACOGA.

### 4.2 RACOGA: RELAXING THE POSCORR CONDITION

We introduce a new condition named RACOGA which is related, but not the same as two other conditions named gradient diversity (Yin et al., 2018) and gradient confusion (Sankararaman et al., 2020). We discuss these different conditions in Appendix C.

**Definition 4.** *We say that $f$ verifies the Relaxed Averaged COrrelated Gradient Assumption (RACOGA) if there exists $c \in \mathbb{R}$ such that the following inequality holds:*

$$\forall x \in \mathbb{R}^d \backslash \overline{\mathcal{X}}, \quad \frac{\sum_{1 \leq i < j \leq N} \langle \nabla f_i(x), \nabla f_j(x) \rangle}{\sum_{i=1}^N \|\nabla f_i(x)\|^2} \geq c, \qquad \text{(RACOGA)}$$

*where $\overline{\mathcal{X}} = \{x \in \mathbb{R}^d, \forall i \in \{1, \ldots, N\}, \|\nabla f_i(x)\| = 0\}$.*

RACOGA is a generalisation of the condition PosCorr which allows to quantify anti correlation ($c < 0$) or correlation ($c > 0$) between gradients.

**Proposition 2.** *Assume RACOGA holds with $c > -\frac{1}{2}$. Then, considering batch of size $1$, $f$ verifies the SGC with $\rho_1 = \frac{N}{1+2c}$.*

Proposition 2 is proved in Appendix H.2. Note that RACOGA is always verified with $c = -\frac{1}{2}$, and we have $c \leq \frac{N-1}{2}$ (Appendix H.5).

**Remark 6.** *Proposition 2 creates a direct link between RACOGA and SGC. It indicates that the more correlation between gradients there is, the lower is the SGC constant, which results in improved convergence bounds for SNAG, see Theorems 3-4. Importantly if the gradients are too anti correlated, Proposition 2 could only be verified with $c$ arbitrary close to $-\frac{1}{2}$, resulting in a bound for $\rho_1$ increasing to $+\infty$.*

**Remark 7.** *It is well known that considering high dimensional vectors drawn uniformly on the unit sphere, they will be pairwise quasi orthogonal with high probability (Milman & Schechtman, 1986). Sankararaman et al. (2020) show theoretically and empirically that considering neural networks with data drawn uniformly on the unit sphere, under some assumptions, linked to over parameterization, each scalar product $\langle \nabla f_i(x), \nabla f_j(x) \rangle$ is not too negative. In this case, RACOGA is thus verified for a $c$ that is at worst close to zero.*

### 4.3 THE STRONG GROWTH CONDITION WITH BATCH SIZE $1$ DETERMINES HOW THE PERFORMANCE SCALES WITH BATCH SIZE

In this section, we build over Proposition 2, which only covers batches of size $1$, to take into account bigger batches.

**Lemma 1.** *Assume that for batches of size $1$, $f$ verifies the SGC with constant $\rho_1$. Then, for batches of size $K$, $f$ verifies the strong growth condition with constant $\rho_K$ where*

$$\rho_K \leq \frac{1}{K(N-1)} \left( \rho_1(N-K) + (K-1)N \right). \qquad (13)$$

Lemma 1 is proved in Appendix H.3. It shows that if the SGC is verified for batches of size $1$, it is verified for any size of batch $K$ and we can compute an estimation of $\rho_K$.

Strikingly, Lemma 1 allows to study the effect of increasing batch size $K$ on the number of $\nabla f_i$ evaluations we need to reach a $\varepsilon$-solution. We only consider the convex case, as the same reasoning and results hold for strongly convex functions.

**Theorem 5.** *Assume $f$ is convex and $L$-smooth, and that $\tilde{\nabla}_1$ verifies the SGC with $\rho_1 \geq 1$. Then, SNAG (Algorithm 2) with batch size $K$ allows to reach an $\varepsilon$-precision (2) at this amount of $\nabla f_i$ evaluations:*

$$\Delta_K . \rho_1 \sqrt{\frac{2L}{\varepsilon}} \|x_0 - x^*\|, \tag{14}$$

*where $\Delta_K := \left( \frac{N-K}{N-1} + \frac{N}{\rho_1} \frac{K-1}{N-1} \right)$, and $\rho_1 \sqrt{\frac{2L}{\varepsilon}} \|x_0 - x^*\|$ is the number of $\nabla f_i$ evaluations needed to reach an $\varepsilon$-precision when using batches of size $1$ according to Theorem 3.*

Compared to the theorems of section 3, Theorem 5 gives a bound on the number of $\nabla f_i$ we have to evaluate, not the number of iterations of the algorithm, see the proof in Appendix H.4.

Theorem 5 indicates that using batches of size $K$, we need $\Delta_K$ times the number of $\nabla f_i$ that is needed when using batches of size $1$ to reach an $\epsilon$-precision. Note also that it assumes the knowledge of $\rho_1$, that can be determined using RACOGA, see Proposition 2.

**Remark 8.** *From Theorem 5, we distinguish 3 regimes, among which the orthogonality of gradients is a critical state.*

1. *$\rho_1 = N$. This is notably true when the gradients are orthogonal. $\Delta_K = 1$ for any value of $K$, and the number of $\nabla f_i$ evaluations is exactly the same independently of batch size.*

2. *$\rho_1 < N$. The gradients are in average positively correlated, i.e. RACOGA is verified with $c > 0$. $\Delta_K > 1$, and increasing batch size leads to an increasing amount of $\nabla f_i$ evaluations. So, increasing batch size will make parallelization sublinearily efficient, a phenomenon known as performance saturation, see Ma et al. (2018); Liu & Belkin (2020).*

3. *$\rho_1 > N$. The gradients are in average negatively correlated, i.e. RACOGA is verified with $c < 0$. $\Delta_K < 1$ and larger batches leads to a decreasing amount of $\nabla f_i$ evaluations.*

Theorem 5 and Remark 8 state that, considering convergence speed, replacing the exact gradient by a stochastic approximation is not necessarily cheaper, in term of number of $\nabla f_i$ we evaluate.

## 5 NUMERICAL EXPERIMENTS

Our theory indicates that the correlation between gradients, evaluated through RACOGA is needed to have SNAG (Algorithm 2) outperforming SGD (Algorithm 1). We provide numerical experiments to validate this statement by running SNAG to optimize classic machine learning models such as linear regression (Section 5.1) or classification neural network (Section 5.2). We also compare its performance with its deterministic version NAG (Algorithm 7), together with GD (Algorithm 6). Our code is available at `https://github.com/J-Hermant/Momentum_Stochastic_GD`.

**RACOGA in practice** We introduced the RACOGA as an inequality that holds over all the space. However, in practice, one only needs to consider the RACOGA values along the optimization path. So this quantity will be computed only along this path.

**Performance metrics** Our interest will be how the algorithms make the training loss function decrease. In order to make a fair comparison between algorithms, our $x$-axis is the number of $\nabla f_i$ evaluations, not $n_{it}$, the number of iterations of the algorithms.

### 5.1 LINEAR REGRESSION

For a dataset $\{a_i, b_i\}_{i=1}^N \in \mathbb{R}^d \times \mathbb{R}$, we want to solve a linear regression formulated as Problem LR.

$$f(x) := \frac{1}{N} \sum_{i=1}^{N} f_i(x) := \frac{1}{N} \sum_{i=1}^{N} \frac{1}{2} (\langle a_i, x \rangle - b_i)^2. \tag{LR}$$

We consider the overparameterized case, *i.e.* $d > N$. Note that the linear regression problem is convex and smooth, so we are in the theoretical setting of this paper. Moreover, we can directly compute the parameters involved in the algorithms except for a parameter $\lambda$ that replaces the unknown $\rho_K$ constant in the case of SNAG, see details in Appendix A.1.

An interesting characteristic of linear regression is that as $\nabla f_i(x) = (\langle a_i, x \rangle - b_i)a_i$, we have

$$\underbrace{\langle \nabla f_i(x), \nabla f_j(x) \rangle}_{\text{gradient correlation}} = (\langle a_i, x \rangle - b_i)(\langle a_j, x \rangle - b_j) \underbrace{\langle a_i, a_j \rangle}_{\text{data correlation}}. \tag{15}$$

The pairwise correlation between the gradients depends explicitly on the pairwise correlation inside the data. Therefore, we expect the correlation inside data to impact on the RACOGA values, and thus on the performance of stochastic algorithms such as SNAG (Algorithm 2).

To validate this intuition experimentally, we build two different datasets with $N = 100$ and $d = 1000$. The first set of $\{a_i\}_{i=1}^{N}$ is generated uniformly onto the $d$-dimensional sphere, such that the data are fewly correlated. The second one is generated by a Gaussian mixture law with ten modes, which induces correlation inside data. In both cases, the $\{b_i\}_{i=1}^{N}$ are generated by a Gaussian law.

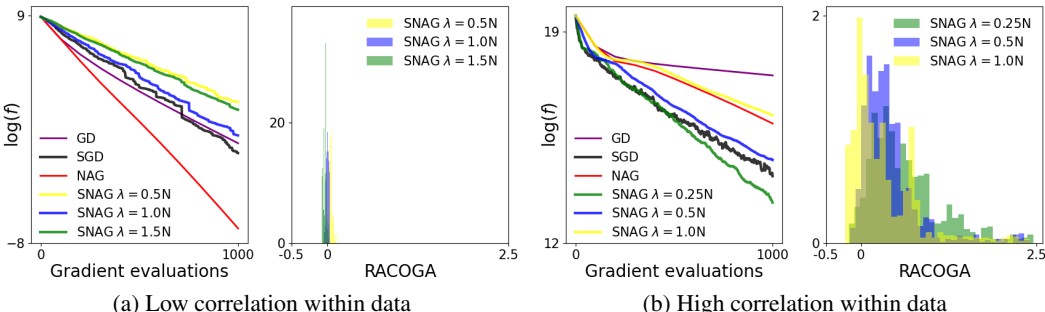

(a) Low correlation within data        (b) High correlation within data

Figure 1: Illustration of the convergence speed of GD (Algorithm 6), SGD (Algorithm 1, batch size 1), NAG (Algorithm 7) and SNAG (Algorithm 2, batch size 1) on a linear regression problem, together with an histogram distribution of RACOGA values along the iterations of SNAG. Stochastic algorithms results are averaged over ten runs. On the left, data are generated by a law that make them fewly correlated, while on the right the data are generated by a gaussian mixture, leading to higher correlation. The $\lambda$ parameter replaces the unknown SGC constant in the algorithm, see Appendix A.1. Note that the data correlation results in better performance of SNAG, whereas uncorrelated data lead to smaller RACOGA values, reducing the benefit of using SNAG.

**SGD vs SNAG** Comparing right parts of Figure 1a and Figure 1b, we observe that the lack of correlation inside data leads to smaller values of RACOGA. In the case of Figure 1a, these lower RACOGA values coincide with SGD being faster than SNAG. On the other hand we see on Figure 1b that the presence of correlation inside the data makes the optimization path crosses areas with higher RACOGA values, allowing SNAG to be faster than SGD. These experimental results support our theoretical findings.

**Deterministic vs stochastic** Strikingly, one can also observe that the lack of correlation results in poor performance of all the stochastic algorithms, especially compared to NAG (Figure 1a). Conversely, presence of correlation results in the opposite phenomenon (Figure 1b). These observations are consistent with our theoretical findings of Section 4.3, which indicate that is some cases, stochastic algorithms are not necessarily cheaper to use than their deterministic counterparts in term of $\nabla f_i$ evaluations.

**Role of $\lambda$** In the case of SNAG (Algorithm 2), the choice of parameters from Theorem 3 involves the $\rho_K$ constant, that we do not know. We thus replace $\rho_K$ by a parameter $\lambda$. The higher the

RACOGA are, the smaller $\rho_K$ is, and the smaller $\lambda$ can be chosen which results to more aggressive steps of the algorithm, as $s = \frac{1}{L\lambda}$. See details in Appendix A.1.

## 5.2 NEURAL NETWORKS

In this second experiment, we aim to test if the crucial role of correlation inside data observed for linear regression (Section 5.1) extend to more general models.

For a dataset $\{a_i, b_i\}_{i=1}^N \in \mathbb{R}^d \times \mathbb{R}$, we consider a classification problem tackled with a neural network model with the cross-entropy loss. Importantly, this problem is **not convex**, so we are not anymore in the setting of our theoretical results.

We use the SNAG version implemented in *Pytorch* (Algorithm 3), that is equivalent to Algorithm 2, see Appendix B.2

As for the linear regression problem, we use two different datasets. The first one, CIFAR10 (Krizhevsky & Hinton, 2009), is composed of images and serves as the correlated dataset. The second one, generated onto the $d$-dimensional sphere with 2 different labels according to which hemisphere belongs each data (see details in Appendix A.2), serves as the uncorrelated dataset. For the CIFAR10 experiment, we use a Convolutional Neural Network (CNN, LeCun et al. (1998)), and for the sphere experiment we use a Multi Layer Perceptron (MLP, Rumelhart et al. (1986)).

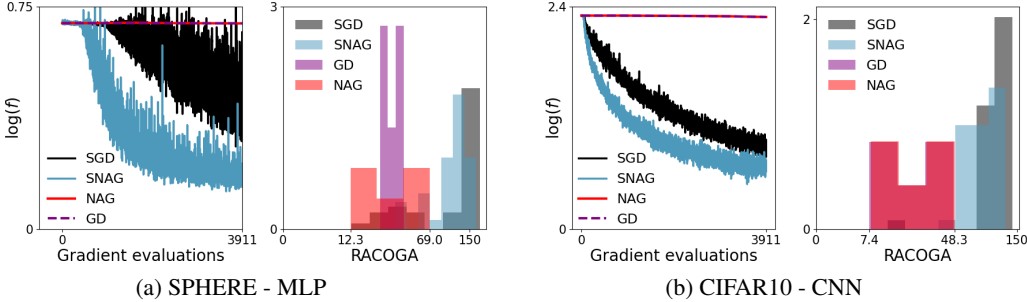

(a) SPHERE - MLP          (b) CIFAR10 - CNN

Figure 2: Illustration of the convergence speed of GD (Algorithm 6), SGD (Algorithm 1, batch size 64), NAG (Algorithm 3, full batch) and SNAG (Algorithm 3, batch size 64) averaged over 10 different initializations, together with an histogram distribution of RACOGA values taken along the optimization path, averaged over 10 different initialisations, where the $x$-axis scale is logarithmic. On the left, we use a MLP to classify data sampled from a law such that they are fewly correlated. On the right we use a CNN to classify CIFAR10 images. Note that contrarily to Figure 1, the presence of correlation within data no longer influence the RACOGA values, that remains high in both cases, resulting in better performances of SNAG.

Strikingly, it appears on Figures 2a and 2b that the correlation inside data has no longer direct impact on the RACOGA values along the iterations path. In each case the RACOGA values are high, which results is SNAG outperforming other algorithms. In particular, both deterministic algorithms are significantly less efficient.

These experiments indicate that neural networks offer high RACOGA values, that SNAG can take advantage of to converge faster.

## 6 CONCLUSION

In this paper, we introduced RACOGA to help us to understand in which case the Stochastic Nesterov Accelerated Gradient algorithm (SNAG) allows to outperform the Stochastic Gradient Descent (SGD) for convex or strongly convex functions, as it happens for the deterministic counterparts of these algorithms. We demonstrate theoretically and empirically that **large RACOGA values allows to accelerate SGD with momentum**. RACOGA may be the, up to now, missing ingredient to understand the acceleration possibilities offered by SNAG in this setting, outside of the linear regression problems.

ACKNOWLEDGMENTS

This work was supported by the ANR project PEPR PDE-AI, and the French Direction Générale de l'Armement. Experiments presented in this paper were carried out using the PlaFRIM experimental testbed, supported by Inria, CNRS (LABRI and IMB), Universite de Bordeaux, Bordeaux INP and Conseil Regional d'Aquitaine (see https://www.plafrim.fr). We thank Bilel Bensaid to bringing to our knowlegde the works about gradient diversity and gradient confusion, and Eve Descomps for her help for figure design.

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

## SUPPLEMENTARY MATERIAL

These supplements contain additional details on the numerical experiments, proofs of our theoretical results, and some additional insights. In Supplement A, we present additional details about numerical experiments and supplementary experiments. In Supplement B, an optimization background is provided for completeness. In Supplement C, we present and comment related works. In Supplement D, we provide simple examples of functions that do not verify SGC, or verify it for a large constant. In Supplement E, we provide convergence proofs for SGD (Algorithm 1). In Supplement F, we introduce new convergence results for SNAG (Algorithm 2) without assuming that the SGC holds. In Supplement G, we present convergence proof for SNAG (Algorithm 2). In Supplement H, we provide the proof of the results presented in Section 4. Finally in Supplement I, we give deeper explanations concerning the link between RACOGA and algorithms considered in this paper when applied to the problem of linear regression.

### REPRODUCTIBILITY STATEMENT

Source code used in our experiments can be found in supplementary material. It contains a README.md file that explains step by step how to run the algorithm and replicate the results of the paper. We detail our datasets, network architectures and parameter choices in Section A.2. Theoretical results presented in the paper are proved in the appendices.

### IMPACT STATEMENT

The present paper, from an optimization point of view, aims to strengthen our understanding of the theory of machine learning. A good comprehension of the tools that are broadly used is important in order to quantify their impacts on the world. Moreover, considering environmental impact, it is crucial to understand the process that makes learning more efficient, especially accelerate current optimization algorithms, without necessarily using huge models.

## A  ADDITIONAL EXPERIMENTS AND DETAILS

This section presents additional details on the experiments for the sake of reproducibility. We also provide additional experiments for a deeper analysis.

### A.1  LINEAR REGRESSION

**Algorithms and parameters**   For the problem LR, we can explicitly compute geometrical constants. Indeed, $f$ is $L$-smooth with $L = \frac{1}{N}\lambda_{\max}\left(\sum_{i=1}^{N} a_i a_i^T\right)$, and each $f_i$ is $L_i$-smooth with $L_i = \lambda_{\max}\left(a_i a_i^T\right)$. In the overparametrized case *i.e.* $N < d$, $f$ is not $\mu$-strongly convex. However, up to a restriction to the vectorial subspace $\mathbf{V}$ spanned by $\{a_1, \ldots, a_N\}$, $f|_{\mathbf{V}}$ is $\mu = \frac{1}{N}\lambda_{\min}\left(\sum_{i=1}^{N} a_i a_i^T\right)$-strongly convex. Therefore in our experiments, in order to run GD, SGD and NAG (respectively Algorithms 6, 1 and 7), we chose the parameters respectively according to Theorem 6, 1 and 7. In the case of SNAG (Algorithm 2), in order to apply Theorem 3, we also need to know the SGC constant $\rho_K$, where $K$ is the selected batch size. However this constant is hard to compute. The knowledge of RACOGA along iterates would be sufficient, but we do not know this path before launching the algorithm. Thus, we run SNAG with this choice of parameters

$$s = \frac{1}{L\lambda}, \; \eta = \frac{1}{\sqrt{\mu\bar{L}\lambda}}, \; \beta = 1 - \frac{1}{\lambda}\sqrt{\frac{\mu}{L}}, \; \alpha = \frac{1}{1 + \frac{1}{\lambda}\sqrt{\frac{\mu}{L}}} \tag{16}$$

where $\lambda \geq 1$. In order to achieve better performance, provided that the iterates cross areas with higher RACOGA values, $\lambda$ can be chosen more aggressively, *i.e.* $\lambda$ smaller, as on Figure 1b. Decreasing the $\lambda$ parameter leads to a more aggressive, or less safe algorithm, because it will increase $s$ and $\eta$ in Equation (16). Taking a glance at Algorithm 2, we see that it results in making larger gradient steps. Recall that larger gradients steps allows the trajectory to move faster, and eventually to converge faster, but steps that are too big will make the algorithm diverge.

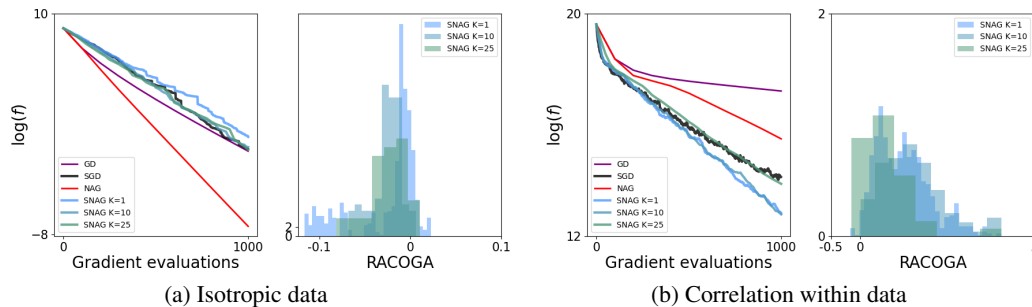

|  | (a) Isotropic data | (b) Correlation within data |
| --- | --- | --- |

Figure 3: Illustration of the convergence speed of GD (Algorithm 6), NAG (Algorithm 7), SGD (Algorithm 1) and SNAG (Algorithm 2) with varying batch sizes $K$, applied to a linear regression problem, together with an histogram distribution of RACOGA values along the iterations of SNAG. The stochastic algorithms results are average on ten runs. On the left, data are generated by a law such that they are fewly correlated, while on the right the data are generated by a gaussian mixture, such that some of the data are highly correlated. Note that the presence of correlation in data results in a decrease of performance for SNAG (Algorithm 2) when increasing too much the batch size, whereas uncorrelated data results in an improvement of performance when increasing batch size.

**Batch size influence**   In Section 5.1, we observed that contrarily to data generated uniformly onto the sphere, the presence of correlation inside data coincides with high RACOGA values and good performance of SNAG (Algorithm 2) with batch size 1 compared to other algorithms. Now in the same experimental setting, we study the impact of varying batch size. According to Remark 8, in the case of correlated data, we should observe a decrease of the performance up to a certain batch size. On Figure 3b, we observe this phenomenon. We see that we can multiply the batch size by a factor 10, and keeping the same performance. If performing parallelization, this results in 10 times faster computations. However, when increasing batch size from 10 to 25, we lose performance, inducing

---

**Algorithm 3** Stochastic Nesterov Accelerated Gradient - Machine learning version (SNAG ML)

1: **input:** $x_0, b_0 \in \mathbb{R}^d$, $s > 0$, $p \in [0, 1]$, $(\eta_n)_{n \in \mathbf{N}} \in \mathbf{R}_+^{\mathbf{N}}$
2: **for** $n = 0, 1, \ldots, n_{it} - 1$ **do**
3:     $b_n \leftarrow pb_{n-1} + \tilde{\nabla}_K(x_{n-1})$
4:     $x_n \leftarrow x_{n-1} - s\left(\tilde{\nabla}_K(x_{n-1}) + pb_n\right)$
5: **end for**
6: **output:** $x_{n_{it}}$

---

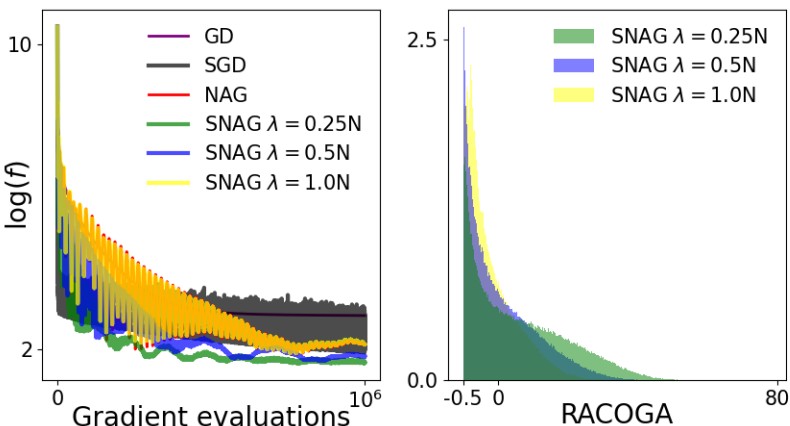

Figure 4: Illustration of the convergence speed of GD (Algorithm 6), NAG (Algorithm 7), SGD (Algorithm 1) and SNAG (Algorithm 2) for the linear regression problem with the Boston dataset. The parameters of SNAG are choosen accordingly to Equation 16. Some RACOGA values are high, while other are also close to the worst anti-correlation case $-\frac{1}{2}$. Note that SNAG appears to converge faster than the other algorithms when choosing $\lambda$ small enough.

paralellization will not results in 2.5 times faster computation. This phenomenon is often referred to as *performance saturation*, see Ma et al. (2018); Liu & Belkin (2020). On Figure 3a, we observe that conversely, increasing batch size improve the performance. Figure 3 is thus consistent with our theoretical findings, see Remark 8.

**Under-parameterization with a realistic dataset**  The Boston dataset (Harrison & Rubinfeld, 1978) concerns house pricing in the Boston area. This dataset can be used for linear regression, where we aim to predict house prices using 13 variables such as tax rates. There are 200 data, so $N = 200$ and $d = 13$, which means we are in an under-parameterized regime. In this case, we can hardly expect SGC to hold, as the noise of the gradient has no reason to be zero at the minimizers of $f$. On Figure 4, we tackle the linear regression problem on this dataset, and we observe that despite the under-parameterized setting, SNAG (Algorithm 2) appears to outperform GD (Algorithm 6), NAG (Algorithm 7) and SGD (Algorithm 1). RACOGA values can be either very large (*e.g.* for aggressive, *i.e.* small $\lambda$), but also close to $-\frac{1}{2}$, which corresponds to high anti-correlation. In this under-parameterized regime, the role of RACOGA is less clear. A possible explanation of the good performance of SNAG could be the variance reduction effect of momentum, see Gao et al. (2024).

## A.2   NEURAL NETWORKS

For a dataset $\{a_i, b_i\}_{i=1}^N \in \mathbb{R}^d \times \mathbb{R}$, we want to solve a classification problem formulated as Problem C.

$$f(x) := \frac{1}{N} \sum_{i=1}^N f_i(x) := \frac{1}{N} \sum_{i=1}^N CROSS(x; a_i, b_i). \tag{C}$$

Where $CROSS()$ is a cross entropy loss, as it is implemented in Pytorch with the function *nn.CrossEntropyLoss()*.

**Classification problem**    In Section 5.2, we considered two classification problems. The first one involves the classic CIFAR10 dataset (Krizhevsky & Hinton, 2009), that contains 60000 color images (dimension $32 \times 32$) with 10 different labels. See Figure 5a to see a data visualisation of the dataset, taken from Balasubramanian et al. (2022). This dataset serves as our correlated dataset. For the second classification problem, we generated data drawn uniformly onto the $d$-dimensional sphere, where $d$ is the same dimension as the image of CIFAR10, *i.e.* $3*32*32 = 3072$. We created 2 different labels depending on the positivity of the first coordinate of each data, which remains to associate a different label depending on which hemisphere belong each data, see Figure 5b for a 3d visualisation. This dataset is named SPHERE.

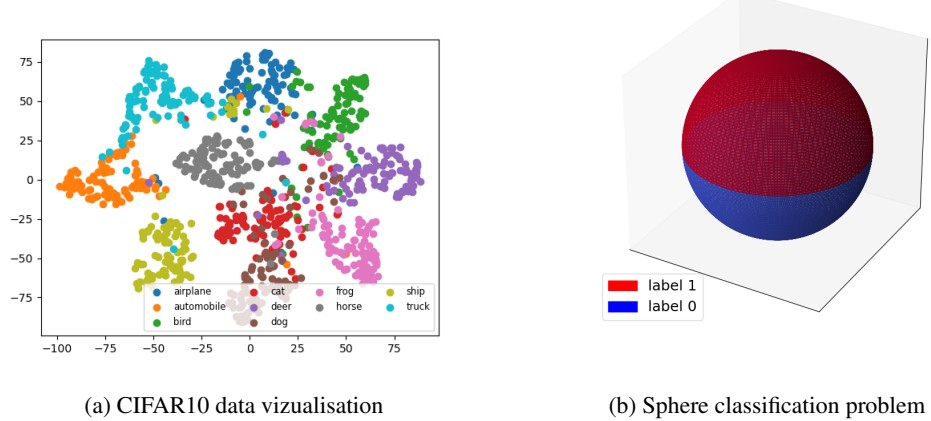

(a) CIFAR10 data vizualisation          (b) Sphere classification problem

Figure 5: Illustration of the two classification problems we consider in Section 5.2. On the left part, wee see a 2D vizualisation of CIFAR10 data set, proposed by Balasubramanian et al. (2022). On the right, we illustrate SPHERE dataset on the $3d$-sphere, where each hemisphere correspond to a different label.

**Network architecture**    For the classification problem involving the CIFAR10 dataset, we use a Convolutionnal Neural Network (CNN, LeCun et al. (1998)). For the classification problem involving the spherical data, we use a Multi Layer Perceptron (MLP, Rumelhart et al. (1986)). CNN are efficient architecture when it comes to image classification, at is exploit local information of the images. However, this architecture makes less sense for our classification problem on the sphere. This model indeed performed poorly in our experiment, and so our choice of a MLP architecture. We detail the architectures on Figure 6.

**Algorithms and parameters**    We ran the experiments using the *Pytorch* library. We used the *Pytorch* implementation to run the optimization algorithms, through the function *torch.optim.SGD*. This function contains a *nesterov = True* argument, which allows to run Algorithm 3. Note that this is indeed a formulation of the Nesterov algorithm, see Appendix B.2. The detailed parameters used to run the experiment, determined by grid search, are displayed on Figure 2 are presented on Table 2, together with the final test accuracy, averaged over 10 initialisations.

**Single run and RACOGA along iterations**    As a complement, we display on Figure 7 a slightly different view of Figure 2. On the left part of Figure 7a-Figure 7b, we display the typical behaviour of one single run of optimization algorithms, namely without averaging for several initialisations. As we can expect, we observe a higher variability, although the general behaviour remains similar. On the right part of Figure 7a-Figure 7b, we displayed the evolution of RACOGA values along the iterations, instead of the histogram distribution of Figure 2, which does not keep any temporal information. For correlated data, we observe that RACOGA values decrease when iterations converge.

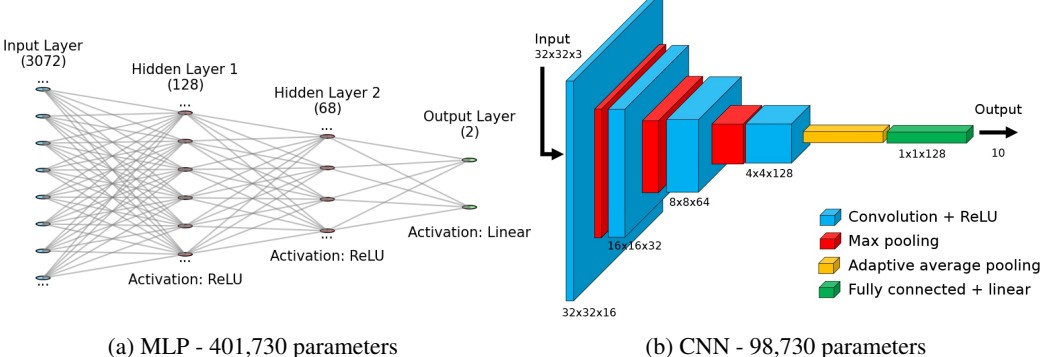

(a) MLP - 401,730 parameters                    (b) CNN - 98,730 parameters

Figure 6: Scheme of the architecture of the MLP and the CNN used in our experiments.

|  | MLP (SPHERE) | | | CNN (CIFAR10) | | |
|---|---|---|---|---|---|---|
|  | $s$ | $p$ | Accuracy test(%) | $s$ | $p$ | Accuracy test(%) |
| GD | 3 | - | 49.72 | 4 | - | 17.54 |
| SGD | 0.3 | - | 80.6 | 0.3 | - | 65.40 |
| NAG | 2 | 0.9 | 50.02 | 2 | 0.7 | 17.07 |
| SNAG | 0.1 | 0.9 | **88.73** | 0.05 | 0.9 | **70.88** |

Table 2: Parameters (the learning rate $s$ and the momentum $p$) used to run the experiments presented in Section 5.2, together with the final accuracy test in percent averaged over 10 different initializations. For precise formulation of the algorithms, see Algorithm 6 (GD), Algorithm 1 (SGD), Algorithm 8 (NAG) and Algorithm 3 (SNAG).

This phenomenon can be interpreted as the convergence of the iterations to a minimum with low curvature which is related to a low RACOGA value (see Appendix I for more details).

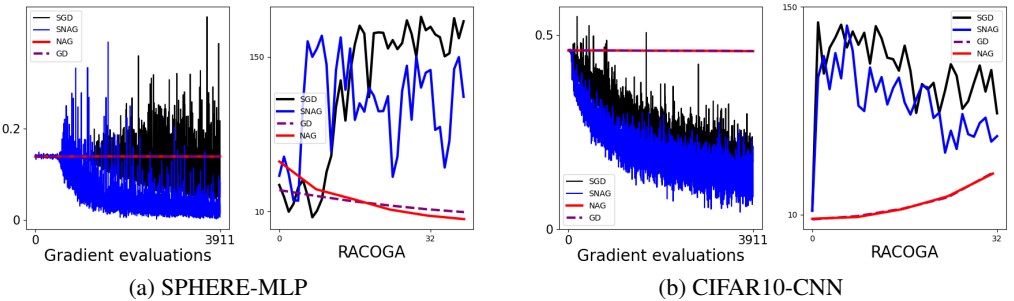

(a) SPHERE-MLP                    (b) CIFAR10-CNN

Figure 7: Illustration of the convergence speed of on run of GD (Algorithm 6 batch size N ), SGD (Algorithm 1, batch size 64), NAG (Algorithm 7) and SNAG (Algorithm 2, batch size 64), together with a display of the RACOGA values taken along the path, averaged over 5 different initializations. On the left, we use a MLP, and data are generated by an isotropic law and are fewly correlated. On the right we use a CNN, and the dataset used is CIFAR10.

**Comparison with ADAM and RMSprop**   RMSprop (Hinton, 2012) (Algorithm 4) and ADAM (Kingma & Ba, 2015) (Algorithm 5) are popular algorithms when it comes to optimize neural networks. RMSprop is similar to gradient descent, at the difference that, grossly, it divides componentwise the gradient by an average of the squared norm of the past gradients. There exists a variant of RMSprop that incorporates momentum. Adam combines both techniques of RMSprop and momentum, plus other mechanisms such as bias corrections.

On Figure 8, compared to Figure 2, we add the training convergence curve and RACOGA values of RMSprop and ADAM. On the CIFAR10 dataset, both algorithms do not converge faster than

---

**Algorithm 4** Root Mean Square Propagation (RMSprop)

---

1: **Input:** $\alpha, \beta, \epsilon > 0, x_0 \in \mathbb{R}^d$
2: **for** $n = 0, \ldots, n_{it} - 1$ **do**
3:     $g_n = \tilde{\nabla}_K(x_n)$
4:     $v_{n+1} = \beta v_n + (1 - \beta)g_n^2$
5:     $x_{n+1} = x_n - \alpha \frac{g_n}{\sqrt{v_{n+1}+\epsilon}}$
6: **end for**

---

**Algorithm 5** Adaptive Moment Estimation (ADAM)

---

1: **Input:** $\alpha, \beta_1, \beta_2, \epsilon > 0$, initial parameters $x_0 \in \mathbb{R}^d, m_0 = v_0 = 0$
2: **for** $n = 0, \ldots, n_{it}$ **do**
3:     $g_n = \tilde{\nabla}_K(x_n)$
4:     $m_{n+1} = \beta_1 m_n + (1 - \beta_1)g_n$
5:     $v_{n+1} = \beta_2 v_n + (1 - \beta_2)g_n^2$
6:     $\hat{m}_{n+1} = \frac{m_{n+1}}{1-\beta_1^{n+1}}$
7:     $\hat{v}_{n+1} = \frac{v_{n+1}}{1-\beta_2^{n+1}}$
8:     $x_{n+1} = x_n - \alpha \frac{\hat{m}_{n+1}}{\sqrt{\hat{v}_{n+1}+\epsilon}}$
9: **end for**

---

SNAG, and share similar RACOGA values. Interestingly for the SPHERE dataset, both algorithms are significantly faster than others. We observe that all the algorithms are, at the begining of the optimization process, stuck in a tray. SNAG steps out of this tray faster than SGD, and RMSprop and ADAM step out of it even faster. One may think that the normalization by the average of squared gradients induces larger stepsize and boost the convergence speed, as in this tray the gradient values are low. The average test accuracy at the end of the training for ADAM (Algorithm 5) is 93.73% for MLP, 69.12% for CNN. The average test accuracy at the end of the training for RMSprop (Algorithm 4) is 94.02% for MLP, 67.31% for CNN.

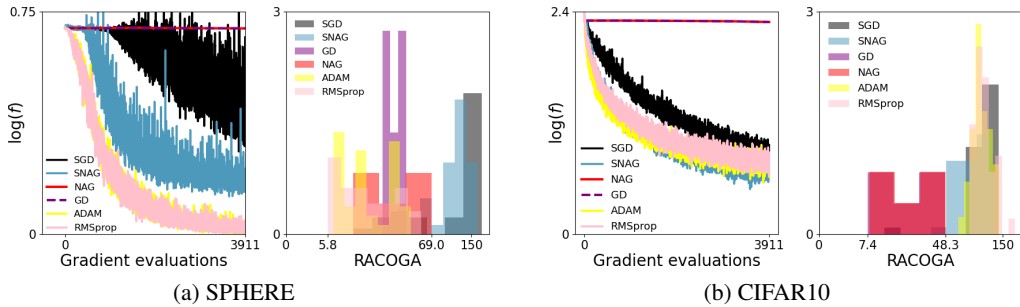

(a) SPHERE    (b) CIFAR10

Figure 8: Illustration of the convergence speed of GD (Algorithm 6), SGD (Algorithm 1, batch size 64), NAG (Algorithm 3, full batch) and SNAG (Algorithm 3, batch size 64), ADAM (Algorithm 5) and RMSprop (Algorithm 4) averaged over 10 different initializations, together with an histogram distribution of RACOGA values taken along the optimization path, averaged over 10 different initialisations. On the left, we use a MLP to classify fewly correlated data sampled from an isotropic law. On the right we use a CNN to classify CIFAR10 images. Note that if ADAM and RMSprop are both better than other algorithms for the SPHERE experiment, they are not faster than SNAG in the CIFAR10 experiment.

**Computation time**    Using the Python library *time*, the computation time needed to choose the best parameters for our two models (CNN for CIFAR10 and MLP for SPHERE) have been saved. Moreover, the computational time needed to generate Figures 1, 2, 3 and 8 is added to our computational budget. We saved the computation time needed to train the models and to compute the RACOGA values, 10 times per algorithms (due to the 10 initializations) and for both networks. Note that for

the stochastic algorithms, RACOGA was computed only every 100 iterations, because of the heavy computation time it demands. The experiments for the linear regression problem take less than one minute of computation. Experiments with SNAG, NAG, SGD and GD took approximately 4 hours. Additional experiments with ADAM and RMSprop took approximately 8 hours. Note that ADAM requires to tune 3 hyperparameters with the grid search, making this step significantly longer. The total computation time needed for all these experiments is approximately 12 hours of computation on a GPU NVIDIA A100 Tensor Core.

**Additional datasets results** On Figure 9, we present the convergence curves and RACOGA values for SGD (Algorithm 1) and SNAG (Algorithm 3) on various datasets, including hand-written numbers (MNIST), pictures of clothes (FashionMNIST Xiao et al. (2017)) and Kuzushiji characters (KMNIST Clanuwat et al. (2018)) and hand-written characters (EMNIST Cohen et al. (2017)). The used learning rate for SGD is $s = 0.1$ for MNIST and EMNIST and $s = 0.25$ for FashionMNIST and KMNIST. The learning rate of SNAG is $s = 0.05$ for MNIST and EMNIST and $s = 0.1$ for Fashion MNIST and KMNIST. The momentum of SNAG (Algorithm 3) is $p = 0.8$ for MNIST and FashionMNIST, and $p = 0.9$ for KMNIST and EMNIST. These quantities have been chosen after a grid search to find the parameters that maximize the test accuracy of the learned model. There were 3 epochs for the training.

Note that the RACOGA values are large for all these training paths. Moreover, in all these scenarios, we observe that the convergence of SNAG is faster than the convergence of SGD, although it is less clear for KMNIST. These additional experimental validations support our theoretical results.

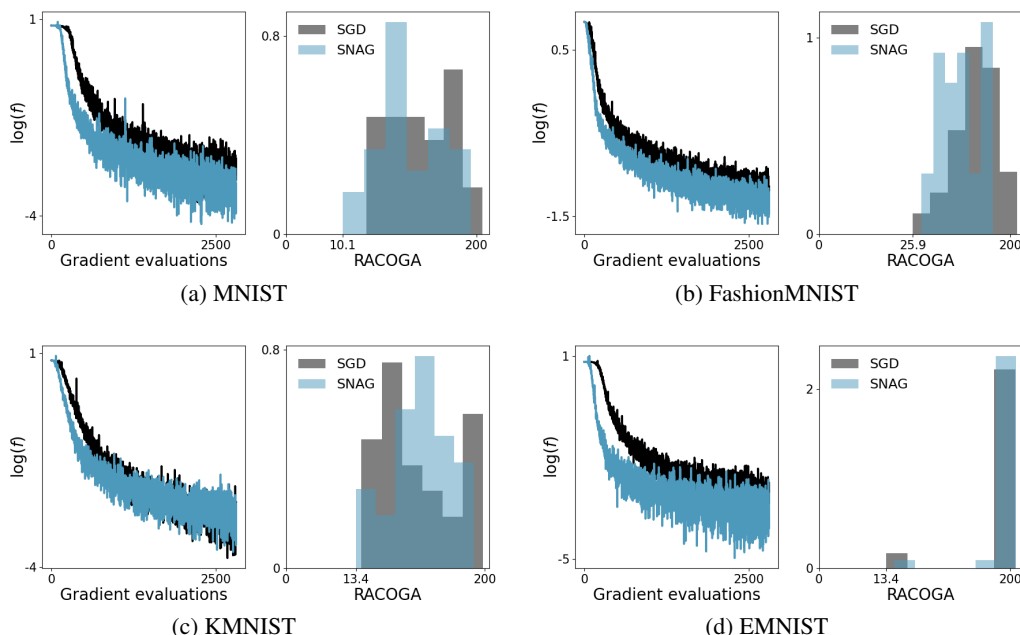

Figure 9: Illustration of the convergence speed of SGD (Algorithm 1 and SNAG (Algorithm 3, batch size 64), averaged over 10 different initializations, together with an histogram distribution of RACOGA values taken along one optimization path. We use a CNN described on Figure 6b, only changing the input layer with the dimension of images $1 \times 28 \times 28$. Note that SNAG is faster than SGD, although it is less clear for KMNIST, and the RACOGA values are large for all these datasets.

### A.3 LOGISTIC REGRESSION

On Figure 10, we present the convergence curves and RACOGA values for SGD (Algorithm 1) and SNAG (Algorithm 3) on images extracted from CIFAR10. Images of CIFAR10 have 10 classes and are of dimension $3,072$, so the logistic regression has $30,730$ parameters. In order to observe the convergence curves on under-parametrized regime, we run the optimization on all the CIFAR10

dataset ($60,000$ images). To analyse the over-parametrized regime, we run the optimization on a subset of $10,000$ images from CIFAR10. The learning rate of SNAG is $s = 0.2$ and the momentum $p = 0.7$. The learning rate for SGD is $s = 1.0$. These parameters have been chosen after a grid search to find the parameters that maximize the test accuracy of the learned model. There were 5 epochs for the training.

Note that for logistic regression the RACOGA values are positive but small (compared to the deep learning experiments). Therefore, we do not observe any acceleration of SNAG over SGD.

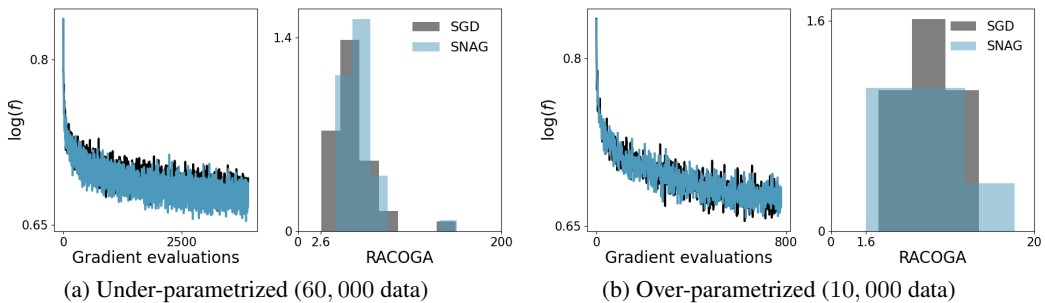

(a) Under-parametrized ($60,000$ data)  (b) Over-parametrized ($10,000$ data)

Figure 10: Illustration of the convergence speed of SGD (Algorithm 1 and SNAG (Algorithm 3, batch size $64$), averaged over 10 different initializations, together with an histogram distribution of RACOGA values taken along one optimization path. We run a logistic regression ($30,730$ parameters) on images extracted from CIFAR10. Note that SNAG and SGD have the same speed, which may be caused by the non convexity of the problem, or by the fact that the RACOGA values might be not large enough to observe an acceleration of SNAG.

## B  BACKGROUND : CONVEX OPTIMIZATION AND THE NESTEROV ALGORITHM

In this section, we give the reader some optimization background for completeness of our paper. First, we expose known results related to the convergence of the deterministic counterparts of SGD and SNAG, and then we exhibit different forms of the Nesterov algorithm that can be found in the literature.

### B.1  PERFORMANCE OF GD AND NAG

We present the speed of convergence of GD (Algorithm 6) and NAG (Algorithm 7). The two following theorems are well known results demonstrated by Nesterov (1983; 2018).

---

**Algorithm 6** Gradient Descent (GD)

---

1: **input:** $x_0 \in \mathbb{R}^d$, $s > 0$
2: **for** $n = 0, 1, \ldots, n_{it} - 1$ **do**
3: $\quad x_{n+1} = x_n - s\nabla f(x_n)$
4: **end for**
5: **output:** $x_{n_{it}}$

---

**Theorem 6.** *Let $f : \mathbb{R}^d \to \mathbb{R}$ be $L$-smooth, $\{x_n\}_{n\in\mathbb{N}}$ be generated by Algorithm 6 with stepsize $s = \frac{1}{L}$. Let $\nu_\epsilon \in \mathbb{N}$ the smallest integer such that $\forall n \geq \nu_\epsilon$, $f(x_n) - f^* \leq \varepsilon$.*

*1. If $f$ is convex, we have*

$$\nu_\epsilon \leq \frac{L}{2} \frac{\|x_0 - x^*\|^2}{\varepsilon}. \tag{17}$$

*2. If $f$ is $\mu$-strongly convex, we have*

---

**Algorithm 7** Nesterov Accelerated Gradient (NAG)

---

1: **input:** $x_0, z_0 \in \mathbb{R}^d$, $s > 0$, $\beta \in [0,1]$, $(\alpha_n)_{n \in \mathbf{N}} \in [0,1]^{\mathbf{N}}$, $(\eta_n)_{n \in \mathbf{N}} \in \mathbf{R}_+^{\mathbf{N}}$
2: **for** $n = 0, 1, \ldots, n_{it} - 1$ **do**
3:    $y_n \leftarrow \alpha_n x_n + (1 - \alpha_n) z_n$
4:    $x_{n+1} \leftarrow y_n - s \nabla f(y_n)$
5:    $z_{n+1} \leftarrow \beta z_n + (1 - \beta) y_n - \eta_n \nabla f(y_n)$
6: **end for**
7: **output:** $x_{n_{it}}$

---

$$\nu_\epsilon \leq \frac{L}{\mu} \log \left( \frac{f(x_0) - f^*}{\varepsilon} \right). \tag{18}$$

The result for convex $f$ can be found in Garrigos & Gower (2023).

**Theorem 7.** *Let* $f : \mathbb{R}^d \to \mathbb{R}$ *be L-smooth,* $\{x_n\}_{n \in \mathbb{N}}$ *be generated by Algorithm 7 with stepsize* $s = \frac{1}{L}$. *Let* $\nu_\epsilon \in \mathbb{N}$ *the smallest integer such that* $\forall n \geq \nu_\epsilon$, $f(x_n) - f^* \leq \varepsilon$.

1. *If* $f$ *is convex, choosing* $\alpha_n = \frac{n-1}{n+1}, \beta = 1, \eta_n = \frac{1}{L} \frac{n+1}{2}$, *we have*

$$\nu_\epsilon \leq \sqrt{2L} \frac{\|x_0 - x^*\|}{\sqrt{\varepsilon}}. \tag{19}$$

2. *If* $f$ *is* $\mu$-*strongly convex, choosing* $\alpha_n = \frac{1}{1 + \sqrt{\frac{\mu}{L}}}, \beta = 1 - \sqrt{\frac{\mu}{L}}, \eta_n = \frac{1}{\sqrt{\mu L}}$, *we have*

$$\nu_\epsilon \leq \sqrt{\frac{L}{\mu}} \log \left( \frac{2(f(x_0) - f^*)}{\varepsilon} \right). \tag{20}$$

By comparing the convergence speed for convex functions given by Theorems 6 and 7, NAG is faster than GD as long as the desire precision $\epsilon$ is small enough. As $L \geq \mu$, considering $\mu$-strongly convex functions NAG is always at least as good as GD. In practice, in particular for ill-conditioned problems, we can have $\mu \ll L$. In theses cases, NAG is a significant improvement over GD.

**Optimality of NAG**    Note that NAG (Algorithm 7) not only outperforms GD (Algorithm 6). It also offers bounds that are optimal among first order algorithms when minimizing strongly or non strongly convex functions, in the sense that it is possible to find examples of functions within these classes such that these bounds are achieved up to a constant (Nemirovskij & Yudin, 1983).

### B.2    The different forms of Nesterov algorithm

There exists several ways of writing Nesterov momentum algorithm, that can be linked together (see *e.g.* Defazio (2019) for 6 different forms in the strongly convex case). Different research fields are used to their own typical formulation. In this section, we present formulations that can be found in the optimization and machine learning communities. Within the optimization community NAG (Algorithm 7) (Zhu & Orecchia, 2017; Hinder et al., 2020) or NAG 2S (Algorithm 9) (Su et al., 2016; Aujol et al., 2023; 2024) are often used. The machine learning community is rather used to the formulation of NAG ML (Algorithm 8). In fact, this last algorithm is the version implemented in Pytorch with the function *torch.optim.SGD()* with the argument *nesterov = True*, up to the condition that $\tau = 0$.

The links between the several forms have been studied previously (Defazio, 2019; Lee et al., 2021). As our results are related to the three points version of NAG (Algorithm 7), we state now results that allow to generate the same optimisation scheme than Algorithm 7 with Algorithms 8-9.

**Proposition 3** (Hermant et al. (2024)). *Consider* $(x_n)_{n \in \mathbb{N}}$ *and* $(y_n)_{n \in \mathbb{N}}$ *generated by NAG (Algorithm 7). The same sequences are generated by Algorithm 9 with choice of parameters*

$$a_n = \frac{1 - \alpha_n}{1 - \alpha_{n-1}} \alpha_{n-1} \beta_{n-1}, \; b_n = (1 - \alpha_n) \left( \frac{\eta_{n-1}}{s} - \frac{\alpha_{n-1} \beta_{n-1}}{1 - \alpha_{n-1}} - 1 \right). \tag{21}$$

---

**Algorithm 8** Nesterov Accelerated Gradient - Machine Learning version (NAG ML)

---

1: **input:** $x_0, b_0 \in \mathbb{R}^d$, $s > 0$, $p \in [0, 1]$, $\tau \in \mathbb{R}$, $(\eta_n)_{n \in \mathbf{N}} \in \mathbf{R}_+^{\mathbf{N}}$
2: **for** $n = 0, 1, \ldots, n_{it} - 1$ **do**
3:     $b_n \leftarrow p b_{n-1} + (1 - \tau) \nabla f(x_{n-1})$
4:     $x_n \leftarrow x_{n-1} - s \left( \nabla f(x_{n-1}) + p b_n \right)$
5: **end for**
6: **output:** $x_{n_{it}}$

---

**Algorithm 9** Nesterov Accelerated Gradient - Two Sequences version (NAG 2S)

---

1: **input:** $x_0 \in \mathbb{R}^d$, $s > 0$, $(a_n)_{n \in \mathbf{N}} \in [0, 1]^{\mathbf{N}}$, $(b_n)_{n \in \mathbf{N}} \in [0, 1]^{\mathbf{N}}$
2: **for** $n = 0, 1, \ldots, n_{it} - 1$ **do**
3:     $y_n \leftarrow x_n + a_n(x_n - x_{n-1}) + b_n(x_n - y_{n-1})$
4:     $x_{n+1} \leftarrow y_n - s \nabla f(y_n)$
5: **end for**
6: **output:** $x_{n_{it}}$

---

Note that Algorithm 8 have constant parameters in the Pytorch implementation. The link between Algorithm 7 and Algorithm 8 can be deduced from Defazio (2019) in the case when $\tau = 1$ for Algorithm 8. We demonstrate a generalisation of this result for $\tau \neq 1$ and we present it in Proposition 4

**Proposition 4.** *Consider $(y_n)_{n \in \mathbb{N}}$ generated by Algorithm 7. The sequence $(x_n)_{n \in \mathbb{N}}$ generated by Algorithm 8 is the same as $(y_n)_{n \in \mathbb{N}}$ with choices of parameters*

$$p = \alpha\beta, \ \tau = \frac{s(1 + \alpha(\beta - 1)) - (1 - \alpha)\eta}{\alpha\beta s}. \tag{22}$$

*Proof.* Our strategy is to write both algorithms in a one point form algorithm to compare the parameters.

*(i) Algorithm 8.* By the line 4 of Algorithm 8, we have the relation

$$b_n = \frac{x_{n-1} - x_n}{ps} - \frac{1}{p} \nabla f(x_{n-1}). \tag{23}$$

Now we can replace the $b_n$ and $b_{n-1}$ in line 3 of Algorithm 8, thereby

$$\frac{x_{n-1} - x_n}{ps} - \frac{1}{p} \nabla f(x_{n-1}) = p \left( \frac{x_{n-2} - x_{n-1}}{ps} - \frac{1}{p} \nabla f(x_{n-2}) \right) + (1 - \tau) \nabla f(x_{n-1}). \tag{24}$$

We multiply both sides by $-ps$, and rearrange

$$x_n = x_{n-1} + p(x_{n-1} - x_{n-2}) - s\nabla f(x_{n-1}) + ps\nabla f(x_{n-2}) - ps(1 - \tau)\nabla f(x_{n-1}) \tag{25}$$
$$= x_{n-1} + p(x_{n-1} - x_{n-2}) - s\nabla f(x_{n-1}) - ps\left(\nabla f(x_{n-1}) - \nabla f(x_{n-2})\right) + \tau ps\nabla f(x_{n-1}) \tag{26}$$

Interestingly, we recognize with the three first terms on the right hand side Polyak's Heavy Ball (HB) equation (Polyak, 1964)

$$x_n = x_{n-1} + p(x_{n-1} - x_{n-2}) - s\nabla f(x_{n-1}). \tag{HB}$$

The $-ps\left(\nabla f(x_{n-1}) - \nabla f(x_{n-2})\right)$ term is often referred as a gradient correction term (Shi et al., 2022), and is characteristic of the difference between Polyak's Heavy Ball and Nesterov's algorithm. The last $+\tau ps\nabla f(x_{n-1})$ is considered as a damping term, and is often needed to obtain the tightest convergence results (Kim & Fessler, 2016), also to achieve accelerated convergence outside of the deterministic convex realm (Hermant et al., 2024).

*(ii) Algorithm 7* We set that $\forall n \in \mathbb{N}$, $\alpha_n = \alpha \in [0, 1]$, and $\eta_n = \eta > 0$. First, combining lines 3 and 4 of Algorithm 7, we have:

$$y_n = \alpha x_n + (1 - \alpha) z_n = \alpha(y_{n-1} - s\nabla f(y_{n-1})) + (1 - \alpha) z_n \tag{27}$$

Hence:

$$z_n = \frac{y_n - \alpha y_{n-1} + \alpha s \nabla f(y_{n-1})}{1 - \alpha}. \tag{28}$$

Now we inject this expression in line 5 of the Algorithm 7

$$\frac{y_{n+1} - \alpha y_n + \alpha s \nabla f(y_n)}{1 - \alpha} = \beta \left( \frac{y_n - \alpha y_{n-1} + \alpha s \nabla f(y_{n-1})}{1 - \alpha} \right) + (1 - \beta)y_n - \eta \nabla f(y_n). \tag{29}$$

Then, multiply both sides by $1 - \alpha$ and rearrange to get

$$y_{n+1} = \alpha y_n - \alpha s \nabla f(y_n) + \beta y_n - \beta \alpha y_{n-1} + \alpha \beta s \nabla f(y_{n-1}) \tag{30}$$
$$+ (1 - \alpha)(1 - \beta)y_n - (1 - \alpha)\eta \nabla f(y_n). \tag{31}$$

By simplifying and regrouping terms, we get

$$y_{n+1} = y_n + \alpha\beta(y_n - y_{n-1}) - s\nabla f(y_n) - \alpha\beta s \left( \nabla f(y_n) - \nabla f(y_{n-1}) \right)$$
$$+ s\nabla f(y_n) + \alpha\beta s \nabla f(y_n) - \alpha s \nabla f(y_n) - (1 - \alpha)\eta \nabla f(y_n) \tag{32}$$
$$= y_n + \alpha\beta(y_n - y_{n-1}) - s\nabla f(y_n) - \alpha\beta s \left( \nabla f(y_n) - \nabla f(y_{n-1}) \right)$$
$$+ \left[ s(1 + \alpha(\beta - 1)) - (1 - \alpha)\eta \right] \nabla f(y_n). \tag{33}$$

By comparing, Equation 26 and Equation 33, we can identify the different parameters

$$p = \alpha\beta \tag{34}$$
$$\tau p s = s(1 + \alpha(\beta - 1)) - (1 - \alpha)\eta \tag{35}$$

So, we have the correspondence

$$p = \alpha\beta \tag{36}$$
$$\tau = \frac{s(1 + \alpha(\beta - 1)) - (1 - \alpha)\eta}{\alpha\beta s} \tag{37}$$

$\square$

**Extension to Stochastic Nesterov algorithms** Proposition 3 and Proposition 4 can be extended to the stochastic versions of NAG (Algorithm 7), NAG ML (Algorithm 8) and NAG 2S (Algorithm 9), where the $\nabla f$ terms are replaced by $\hat{\nabla}_K$. The same correspondence between parameters of these algorithms holds.

## C    RELATED WORKS

### C.1    GRADIENT CORRELATION CONDITIONS

In this paper, we introduce two assumptions related to the average correlations of the gradients of the functions that form the sum in problem (FS), namely

$$\forall x \in \mathbb{R}^d, \quad \sum_{1 \le i < j \le N} \langle \nabla f_i(x), \nabla f_j(x) \rangle \ge 0. \tag{PosCorr}$$

$$\forall x \in \mathbb{R}^d \backslash \overline{\mathcal{X}}, \quad \frac{\sum_{1 \le i < j \le N} \langle \nabla f_i(x), \nabla f_j(x) \rangle}{\sum_{i=1}^N \|\nabla f_i(x)\|^2} \ge -c, \tag{RACOGA}$$

where $\overline{\mathcal{X}} = \{ x \in \mathbb{R}^d, \forall i \in \{1, \ldots, N\}, \|\nabla f_i(x)\| = 0 \}$. PosCorr is a special case of RACOGA, with the choice $c = 0$. The key role of correlation between gradients has been already observed in previous works, through related but different assumptions.

**Gradient diversity (Yin et al., 2018)**   Gradient diversity is defined at a point $x \in \mathbb{R}^d$ as the following ratio

$$\Delta(x) = \frac{\sum_{i=1}^{N} \|\nabla f_i(x)\|^2}{\|\sum_{i=1}^{N} \nabla f_i(x)\|^2}. \tag{GradDiv}$$

This quantity is closely related to the SGC condition. Indeed, for batch size 1, we clearly have

$$\frac{1}{N} \sum_{i=1}^{N} \|\nabla f_i(x)\|^2 \leq N \sup_{x \in \mathbb{R}^d} \Delta(x) \|\nabla f(x)\|^2. \tag{38}$$

Thus, assuming $\sup_{x \in \mathbb{R}^d} \Delta(x) < +\infty$, $f$ verifies SGC with $\rho_1 \leq N \sup_{x \in \mathbb{R}^d} \Delta(x)$. The authors show that increasing batch size is less efficient with a ratio depending on the values that $N\Delta(\cdot)$ takes along the optimization path. The smaller these quantities (high correlation), the smaller this ratio. Conversely, the higher it is (low or anti correlation), the higher the ratio is, inducing a large gain with parallelization of large batches. GradDiv is another measure to quantify gradient correlation. Indeed by developing the squared norm, we have the relation

$$\Delta(x)^{-1} = 1 + 2 \frac{\sum_{1 \leq i < j \leq N} \langle \nabla f_i(x), \nabla f_j(x) \rangle}{\sum_{i=1}^{N} \|\nabla f_i(x)\|^2} \tag{39}$$

However, RACOGA appears naturally in our study of the SGC, and is a direct measure of the correlation between gradients.

**Gradient confusion (Sankararaman et al., 2020)**   $f$ has gradient confusion $\eta \geq 0$ at a point $x \in \mathbb{R}^d$ if

$$\langle \nabla f_i(x), \nabla f_j(x) \rangle \geq -\eta, \; i \neq j \in \{1, \ldots, j\}. \tag{GradConf}$$

The authors show that for some classes of functions and considering SGD, satisfying this assumption allows to reduce the size of the neighbourhood of a stationary point towards which the algorithm converges. Also, they study theoretically and empirically how the gradient confusion behaves when considering neural networks. They show that practices that improve the learning, such as increasing width, batch normalization and skip connections, induce a lower gradient confusion at the end of the training, *i.e.* GradConf is verified with a small $\eta$. **In other words, well tuned neural networks avoid anti-correlation between gradients.** However, compared to RACOGA, GradConf asks for a uniform bound over the correlation of each pair of gradients, which can be much more restrictive.

## C.2   LINEAR REGRESSION ACCELERATION RESULTS

We mentioned in the introduction of this paper that there exists positive results concerning the possibility of acceleration considering the linear regression problem (Jain et al., 2018; Liu & Belkin, 2020; Varre & Flammarion, 2022). However, our setting is a bit different. Indeed, we consider the finite sum setting, while they consider the following problem

$$\min_{x \in \mathbb{R}^d} P(x), \; P(x) := \mathbb{E}_{(a,b) \sim \mathcal{D}} \left[ \frac{1}{2} (\langle a, x \rangle - b)^2 \right]. \tag{40}$$

They assume that they have access to a stochastic first order oracle, that returns the quantity

$$\tilde{\nabla} P(x) = (\langle a, x \rangle - b) a \tag{41}$$

where $(a, b) \sim \mathcal{D}$. Noting $H = \mathbb{E}_{(a,b) \sim \mathcal{D}} [aa^T]$, they denote $R^2$ and $\tilde{\kappa}$ the smallest positive constants such that

$$\mathbb{E}_{(a,b) \sim \mathcal{D}} [\|a\|^2 aa^T] \preceq R^2 H, \; \mathbb{E}_{(a,b) \sim \mathcal{D}} [\|a\|_{H^{-1}}^2 aa^T] \preceq \tilde{\kappa} H. \tag{42}$$

Finally, they note $\mu$ the smallest eigenvalue of $H$ which is assumed to be invertible, and $\kappa := \frac{R^2}{\mu}$. In this setting, there is interpolation if there exists $x^* \in \mathbb{R}^d$ such that almost surely, $b = \langle x^*, a \rangle$. Under Assumption 1, noting $n_{it}$ the algorithm iterations, the convergence speed of GD (Algorithm 6) is of the order $\mathcal{O}(e^{-\frac{n_{it}}{\kappa}})$, while momentum style algorithms allow to have a $\mathcal{O}(e^{-\frac{n_{it}}{\sqrt{\kappa \tilde{\kappa}}}})$ convergence (Jain et al., 2018; Liu & Belkin, 2020). The acceleration is data dependant. In particular, fixing the distribution to be uniform over the orthonormal basis $\{e_1, \ldots, e_d\}$, one have $\kappa = \tilde{\kappa}$ and there

is no acceleration. Our results (Example 3 + Theorem 5) extend outside linear regression this non-acceleration result, to the case of convex functions with orthogonal gradients. However, the ideas behind those results and ours are different, and $\tilde{\kappa}$ is not a correlation measure. According to Jain et al. (2018), $\tilde{\kappa}$ measure the number of $a_i$ we need to sample such that the empirical covariance is close to the Hessian matrix, $H$.

### C.3 SGC RELATED CONVERGENCE RESULTS

The linear convergence of SGD under a variation of SGC, for smooth convex and strongly convex functions, has been addressed in (Schmidt & Roux, 2013). Later, convergence of SNAG type algorithms under strong growth condition, for functions that are convex or strongly convex, also attracted interest. In this section, we discuss how these works relate to Theorems 3-4.

**Vaswani et al. (2019)**   As mentioned in Section 3, the bounds of Theorem 3 were already achieved by (Vaswani et al., 2019). It was achieved with a quite unusual formulation of the SNAG algorithm, which is equivalent, to the following one

---

**Algorithm 10** (Vaswani et al., 2019))

1: **input:** $x_0 = z_0 \in \mathbb{R}^d$, $s > 0$, $\beta \in [0,1]$, $(\alpha_n)_{n \in \mathbf{N}} \in [0,1]^{\mathbf{N}}$, $(\eta_n)_{n \in \mathbf{N}} \in \mathbf{R}_+^{\mathbf{N}}$, $g_n(.)$ stochastic approximation of $\nabla f(.)$ at iteration $n$.
2: **for** $n = 0, 1, \ldots, n_{it} - 1$ **do**
3: $\quad y_n = \alpha_n x_n + (1 - \alpha_n) z_n$
4: $\quad x_{n+1} = y_n - s g_n(y_n)$
5: $\quad z_{n+1} = \beta z_n + (1 - \beta) y_n - \gamma_n s g_n(y_n)$
6: **end for**
7: **output:** $x_{n_{it}}$

---

For $f$ $L$-smooth, and such that SGC is verified with constant $\rho$, the bounds of Theorem 3 are verified considering SNAG (Vaswani et al., 2019) with the following parameters:

• If $f$ $\mu$-strongly convex:

$$\gamma_n = \frac{1}{\sqrt{\mu s \rho}}, \quad \beta_n = 1 - \sqrt{\frac{\mu s}{\rho}}, \quad b_{n+1} = \frac{\sqrt{\mu}}{\left(1 - \sqrt{\frac{\mu s}{\rho}}\right)^{(n+1)/2}},$$

$$a_{n+1} = \frac{1}{\left(1 - \sqrt{\frac{\mu s}{\rho}}\right)^{(n+1)/2}}, \quad \alpha_n = \frac{\gamma_n \beta_n b_{n+1}^2 s}{\gamma_n \beta_n b_{n+1}^n s + a_n^2}, \quad s = \frac{1}{\rho L}.$$

• If $f$ convex:

$$\gamma_n = \frac{\frac{1}{\rho} + \sqrt{\frac{1}{\rho^2} + 4\gamma_{n-1}^2}}{2}, \quad a_{n+1} = \gamma_n \sqrt{s \rho},$$

$$\alpha_n = \frac{\gamma_n s}{\gamma_n s + a_n^2}, \quad s = \frac{1}{\rho L}.$$

In the strongly convex case, intermediate sequences of parameters $(a_n)$ and $(b_n)$ appear. In the convex case, the sequence $(\gamma_n)$ is defined with a recursive formula. Our different proof of Theorem 3 does not make these features appear.

**Gupta et al. (2023)**   Another line of research leads to a similar expectation result to ours, using AGNES (Algorithm 11), and so with a different proof. The authors of Gupta et al. (2023) also get almost sure convergence, nevertheless without convergence rates contrarily to our Theorem 4.

---

**Algorithm 11** Accelerated Gradient descent with Noisy EStimators (AGNES), Gupta et al. (2023)

1: **input:** $f$ (objective/loss function), $x_0$ (initial point), $\alpha = 10^{-3}$ (learning rate), $\eta = 10^{-2}$ (correction step size), $\rho = 0.99$ (momentum), $N$ (number of iterations)
2: $v_0 \leftarrow 0$
3: **for** $n = 0, 1, \ldots, N$ **do**
4:    $g_n \leftarrow \nabla_x f(x_n)$    (gradient estimator)
5:    $v_{n+1} \leftarrow \rho(v_n - g_n)$
6:    $x_{n+1} \leftarrow x_n + \alpha v_{n+1} - \eta g_n$
7: **end for**
8: $g_N \leftarrow \nabla_x f(x_N)$
9: $x_N \leftarrow x_N - \eta g_N$
10: **output:** $x_N$

---

# D    SOME EXAMPLES WITH CRITICAL STRONG GROWTH CONDITION CONSTANT

In this section, we show that even for some simple examples, the SGC constant can be very large or not existing, justifying that finding interesting characterizations of it is a challenging problem.

## D.1    LARGE $\rho_1$ WITH LINEAR REGRESSION

We consider the function

$$f(x) = \frac{1}{2}(\frac{1}{2}\underbrace{\langle e_1, x\rangle^2}_{:=f_1(x)} + \frac{1}{2}\underbrace{\langle a, x\rangle^2}_{:=f_2(x)}),$$

where $e_1 = (1, 0)$, $a = (1, \varepsilon)$. We have $\nabla f_1(x) = \langle e_1, x\rangle e_1$ and $\nabla f_2(x) = \langle a, x\rangle a$. Assume $x_0 = (-\frac{\varepsilon}{2}, \lambda)$. We have

$$\nabla f(x_0) = -\frac{\varepsilon}{4}(1, 0) + \frac{\varepsilon}{2}(\lambda - \frac{1}{2})(1, \varepsilon) \tag{43}$$

$$= \frac{\varepsilon}{2}(\lambda - 1, (\lambda - \frac{1}{2})\varepsilon). \tag{44}$$

Thus we obtain

$$\|\nabla f(x_0)\|^2 = \frac{\varepsilon^2}{4}((\lambda - 1)^2 + (\lambda - \frac{1}{2})^2\varepsilon^2), \tag{45}$$

whereas

$$\mathbb{E}[\|\tilde{\nabla}_1(x_0)\|^2] = \frac{\varepsilon^2}{8} + \frac{\varepsilon^2}{2}(\lambda - \frac{1}{2})^2(1 + \varepsilon^2). \tag{46}$$

Simply note that with the choice $\lambda = 1$, we have $\|\nabla f(x_0)\|^2 = \frac{\varepsilon^4}{16}$, while $\mathbb{E}[\|\tilde{\nabla}_1(x_0)\|^2] = \varepsilon^2 \frac{1}{4} + o(\varepsilon^2)$. Thus

$$\frac{\mathbb{E}[\|\tilde{\nabla}_1(x_0)\|^2]}{\|\nabla f(x_0)\|^2} \approx \frac{1}{\varepsilon^2}. \tag{47}$$

So the strong growth condition can be arbitrarily large as $\varepsilon$ vanishes.

## D.2    NON CONVEX FUNCTIONS SUCH THAT $\rho_1$ DOES NOT EXIST

We consider:

$$f(x) = \frac{1}{2}(f_1(x) + f_2(x))$$

where $x = (x_1, x_2) \in \mathbb{R}^2$, $f_1(x) = \frac{1}{2}(x_1 - \tanh(x_2))^2$, $f_2(x) = \frac{1}{2}(x_1 + \tanh(x_2))^2$. We have:

$$\nabla f_1(x) = (x_1 - \tanh(x_2), -(x_1 - \tanh(x_2))(1 - \tanh^2(x_2))) \tag{48}$$

$$\nabla f_2(x) = (x_1 + \tanh(x_2), (x_1 + \tanh(x_2))(1 - \tanh^2(x_2))) \tag{49}$$

Consider the line $y = (0, y_0)$, $y_0 \in \mathbb{R}$. We have:

$$\nabla f_1(y) = (-\tanh(y_0), \tanh(y_0)(1 - \tanh^2(y_0))) \tag{50}$$

$$\nabla f_2(y) = (\tanh(y_0), \tanh(y_0)(1 - \tanh^2(y_0))). \tag{51}$$

Then:

$$\nabla f(y) = \frac{1}{2}(\nabla f_1(y) + \nabla f_2(y)) = (0, \tanh(y_0)(1 - \tanh(y_0)^2) \underset{|y_0| \to +\infty}{\to} (0, 0). \tag{52}$$

However, we have $\nabla f_1(y) \to (-1, 0)$ and $\nabla f_2(y) \to (1, 0)$ as $y_0 \to +\infty$. Thus, for any $\rho_1 > 0$, the SGC condition is not verified when considering the whole space $\mathbb{R}^2$.

# E   CONVERGENCE OF SGD

In this section, we prove convergence results of SGD (Algorithm 1) stated in Section 3, namely Theorem 1 and Theorem 2. Also, in Subsection E.2 we present a convergence result for SGD under SGC and we justify that it is not a relevant result in order to compare the convergence speed of SGD and SNAG (Algorithm 2).

## E.1   PROOFS OF THEOREM 1 AND THEOREM 2

We can deduce Theorem 1-Theorem 2 by respectively adapting the proof from Garrigos & Gower (2023) and Sebbouh et al. (2021). We can not directly apply their original results because their setting is slightly different: in problem (FS) their convexity assumption holds for each $f_i$, while in our case it solely holds over the whole sum. In both expectation and almost sure cases, the core Lemma is to bound $\mathbb{E}\left[\|\tilde{\nabla}_K(x)\|^2\right]$, which bounds the variance of the estimator. It is Lemma 6.7 in Garrigos & Gower (2023), and Lemma 1.3 in Sebbouh et al. (2021). These two lemmas do not hold in our setting. We will use instead the following result.

**Lemma 2.** *Under assumptions (1)-(2), we have*

$$\mathbb{E}\left[\|\tilde{\nabla}_K(x)\|^2\right] \leq 2L_{(K)}(f(x) - f^*). \tag{53}$$

*Proof.* Let $x \in \mathbb{R}^d$. Consider a fixed batch $B$ with $|B| = K$. We note $f_B(x) = \frac{1}{K}\sum_{i \in B} f_i(x)$, $\nabla f_B(x) := \frac{1}{K}\sum_{i \in B} \nabla f_i(x)$. We first show that $f_B$ is $\frac{\sum_{i \in B} L_i}{K}$-smooth.

$$\forall x, y \in \mathbb{R}^d, \ \|\nabla f_B(x) - \nabla f_B(y)\| \leq \frac{1}{K}\sum_{i \in B}\|\nabla f_i(x) - \nabla f_i(y)\| \tag{54}$$

$$\leq \frac{1}{K}\sum_{i \in B} L_i\|x - y\| \tag{55}$$

$$= \frac{\sum_{i \in B} L_i}{K}\|x - y\|. \tag{56}$$

The first inequality uses triangular inequality, the second inequality uses the assumption that each $f_i$ is $L_i$-smooth, *i.e.* $\|\nabla f_i(x) - \nabla f_i(y)\| \leq L_i\|x - y\|$. Now, it is well known that if $f_B$ is $\frac{\sum_{i \in B} L_i}{K}$-smooth, fixing $x^* \in \arg\min f$ we have $\|\nabla f_B(x)\|^2 \leq 2\frac{\sum_{i \in B} L_i}{K}(f_B(x) - f_B(x^*))$, see Nesterov (2018) page 30. We upper bound this quantity uniformly in the constants $L_i$ over all the batches.

$$\|\nabla f_B(x)\|^2 \leq 2\frac{\sum_{i \in B} L_i}{K}(f_B(x) - f_B(x^*) \leq 2L_{(K)}(f_B(x) - f_B(x^*)). \tag{57}$$

where $L_{(K)} := \max_{B', |B'|=K}\left(\frac{1}{K}\sum_{i \in B'} L_i\right)$. Now, we get back to the random variable $\tilde{\nabla}_K(\cdot)$, we take the expectation over all the batches of size $K$, and we get

$$\mathbb{E}\left[\|\tilde{\nabla}_K(x)\|^2\right] \leq 2L_{(K)}\mathbb{E}\left[\sum_{i \in B}\frac{1}{K}(f_i(x) - \min f_i)\right]. \tag{58}$$

Note that $\mathbb{E}\left[\frac{1}{K}\sum_{i \in B} f_i(x)\right] = f(x)$. By interpolation (Assumption 1), there exists $x^* \in \arg\min f$ such that $x^* \in \arg\min f_i$ for all $1 \le i \le N$, which implies

$$\mathbb{E}\left[\frac{1}{K}\sum_{i \in B} \min f_i\right] = \mathbb{E}\left[\frac{1}{K}\sum_{i \in B} f_i(x^*)\right] = f(x^*) := f^*. \tag{59}$$

$\square$

**Proof of Theorem 1** Now to prove Theorem 1, note that the same proof as for Theorem 6.8 and Theorem 6.12 from Garrigos & Gower (2023) holds, replacing their Lemma 6.7 by our Lemma 2, setting in their proof $\sigma_b^* = 0$ (Assumption 1) and replacing their $2\mathcal{L}_b$ by $L_{(K)}$, allowing in our case to take a stepsize $s \le \frac{1}{2L_{(K)}}$, instead of their $\frac{1}{4\mathcal{L}_b}$ in the convex case, and $s \le \frac{1}{L_{(K)}}$ instead of their $\frac{1}{2\mathcal{L}_b}$ in the strongly convex case. Note that for the convex case, the $\varepsilon$-precision is actually reach with the sequence $\overline{x}_n := \frac{1}{n}\sum_{i=0}^{N-1} x_i$, *i.e.* we get a number of iterations such that $\mathbb{E}\left[f(\overline{x}_n) - f^*\right] \le \varepsilon$.

**Proof of Theorem 2** For the case of a convex function, the almost sure result from Sebbouh et al. (2021) follows from a decrease in expectation. It is obtained in our case replacing their Lemma 1.3 by our Lemma 2. As for the result in expectation, Lemma 2 allows us to choose $s \le \frac{1}{L_{(K)}}$. The rest of the proof follows as in theirs, as no supplementary assumption is needed in the interpolated case (Assumption 1). For the case of strongly convex functions, we apply the same proof as for Proposition 6, fixing $E_n = \|x_n - x^*\|^2$, where $x^* \in \arg\min f$ with $s = \frac{1}{L_{(K)}}$.

### E.2 CONVERGENCE OF SGD WITH STRONG GROWTH CONDITION

Theorem 1 states a convergence result for SGD (Algorithm 1) without making use of the strong growth condition. It is however possible, as done in Gupta et al. (2023). In Theorem 8 we give a very similar result.

**Theorem 8.** *Assume $f$ is L-smooth, and that $\tilde{\nabla}_K$ verifies the SGC for $\rho_K \ge 1$. Then SGD (Algorithm 1) with stepsize $s = \frac{1}{\rho_K L}$ allows to reach an $\varepsilon$-precision (2) at the following iterations*

$$n \ge \frac{\rho_K}{2}\frac{L}{\varepsilon}\|x_0 - x^*\|^2 \quad (f \text{ convex}), \tag{60}$$

$$n \ge \rho_K \frac{L}{\mu}\log\left(2\frac{f(x_0) - f^*}{\varepsilon}\right) \quad (f \text{ } \mu\text{-strongly convex}). \tag{61}$$

*Proof.* These results can be obtained by adapting Theorem 3.4 and Theorem 3.6 from Garrigos & Gower (2023). Indeed, the two equations from their Lemma 2.28 can be adapted to our case in the following way, $\forall x \in \mathbb{R}^d$

$$\mathbb{E}\left[f(x - s\tilde{\nabla}_K(x)) - f(x)\right] \le -s\left(1 - \frac{\rho_K s L}{2}\right)\|\nabla f(x)\|^2. \tag{62}$$

Then, we can reuse their proof just replacing the condition $s \le \frac{1}{L}$ by $s \le \frac{1}{\rho_K L}$. The conclusion proceeds fixing $s = \frac{1}{\rho_K L}$ from their Corollaries 3.5 and 3.8. $\square$

It is straightforward to compare these bounds with those of Theorem 3. This comparison indicates that the bounds of SNAG always outperform the bounds of SGD. The only exception that could occur is in the convex case if $\|x_0 - x^*\| \ll 1$, *i.e.* if we start close from the minimum.
Making such conclusions would be misleading. Indeed in some cases, the characterization of Theorem 8 is suboptimal compared to the one of Theorem 1. Consider the strongly convex bounds: we then compare $\rho_K \frac{L}{\mu}\log\left(2\frac{f(x_0) - f^*}{\varepsilon}\right)$ (Theorem 8) with $4\frac{L_{\max}}{\mu}\log\left(2\frac{f(x_0) - f^*}{\mu\varepsilon}\right)$ (Theorem 1). Now, the question is:

*How do we compare $\rho_K L$ with $L_{\max}$?*

We recall the example of Section D.1, that is

$$f(x) = \frac{1}{2}(\underbrace{\frac{1}{2}\langle e_1, x\rangle^2}_{:=f_1(x)} + \underbrace{\frac{1}{2}\langle a, x\rangle^2}_{:=f_2(x)}). \tag{63}$$

where $e_1 = (1, 0)$, $a = (1, \varepsilon)$, $\varepsilon > 0$. We already computed the gradient, that is

$$\nabla f(x) = \frac{1}{2}\langle e_1, x\rangle e_1 + \frac{1}{2}\langle a, x\rangle a. \tag{64}$$

One can check that the Hessian matrix is

$$\nabla^2 f(x) = \frac{1}{2}\begin{pmatrix} 2 & \varepsilon \\ \varepsilon & \varepsilon^2 \end{pmatrix}. \tag{65}$$

The $L$-smoothness constant is the larger eigenvalue, that is $\frac{1}{4}\left((\varepsilon^4 + 4)^{\frac{1}{2}} + \varepsilon^2 + 2\right) = 1 + o(\varepsilon)$. Moreover, $L_{(1)} = L_{\max} = 1 + \varepsilon^2$. Now, we saw in Section D.1 that $\rho_K$ is at least of the order $O\left(\frac{1}{\varepsilon^2}\right)$. Thereby, for small values of $\varepsilon$, we have $\rho_K L \gg L_{(1)}$, inducing the bound of Theorem 8 is significantly suboptimal. Therefore, Theorem 1 is more relevant than Theorem 8 to compare SGD and SNAG convergence speeds.

## F  CONVERGENCE OF SNAG WITHOUT STRONG GROWTH CONDITION

In this Section, we derive a finite time convergence result in expectation for SNAG (Algorithm 2) without assuming SGC. In this case, as for Theorems 1-2, the bound over the noise is derived from the geometrical properties of the functions.

**Lemma 3.** *Assume $f$ is such that assumptions (1)-(2) hold.*

*If $f$ is $\mu$-strongly convex, we have*

$$\mathbb{E}\left[\|\nabla f(x) - \tilde{\nabla}_K(x)\|^2\right] \le 2(L_{(K)} - \mu)(f(x) - f^*). \tag{66}$$

*If $f$ is convex, we have*

$$\mathbb{E}\left[\|\nabla f(x) - \tilde{\nabla}_K(x)\|^2\right] \le 2L_{(K)}(f(x) - f^*). \tag{67}$$

We prove Lemma 3 in Appendix F.3. We note $L$ the smoothness constant of $f$. Under assumption (2), we have $L \le \frac{1}{N}\sum_{i=1}^{N} L_i \le L_{(K)}$.

**Theorem 9.** *Under Assumptions 1 and 2, SNAG (Algorithm 2) with batch size $K$ guarantees to reach an $\varepsilon$-precision (2) at the following iterations:*

• *If $f$ is convex, $s_n = \frac{1}{2L_{(K)}}\frac{1}{n+1}$, $\alpha_n = \frac{n}{n+2}$, $\eta_n = \frac{1}{4}\frac{1}{L_{(K)}}$, $\beta = 1$,*

$$n \ge \frac{4L_{(K)}}{\varepsilon}\|x_0 - x^*\|^2. \tag{68}$$

• *If $f$ is $\mu$-strongly convex, $s = \frac{1}{16}\frac{\mu}{(L_{(K)} - \mu)^2}$, $\beta = 1 - \frac{1}{8}\frac{\mu}{(L_{(K)} - \mu)}$, $\eta = \frac{1}{4}\frac{1}{(L_{(K)} - \mu)}$, $\alpha = \frac{1}{1 + \frac{1}{4}\frac{\mu}{(L_{(K)} - \mu)}}$,*

$$n \ge 8\frac{L_{(K)} - \mu}{\mu}\log\left(\frac{2(f(x_0) - f^*)}{\varepsilon}\right). \tag{69}$$

In this Section, we denote $\mathcal{F}_n$ the $\sigma$-algebra generated by the $n + 1$ first iterates $\{x_i\}_{i=0}^{n}$ generated by SNAG (Algorithm 2), *i.e.* $\mathcal{F}_n = \sigma(x_0, \ldots, x_n)$. Also, we note $\mathbb{E}_n[\cdot]$ the conditional expectation with respect to $\mathcal{F}_n$.

**Comparison with the bounds of Theorem 1 on SGD (Algorithm 1)**

- $f$ **convex.** In this case, the bound for SGD of Theorem 1 is

$$n \geq 2\frac{L_{(K)}}{\varepsilon}\|x_0 - x^*\|^2.$$

  This bound for SGD is always better than the bound of SNAG from Theorem 9. However this comparison would be misleading, as Theorems 3 show that in the convex case, if aiming for a small enough precision $\varepsilon$, SNAG's bound can always be smaller than the one of SGD.

- $f$ **strongly convex.** In this case, the bound for SGD of Theorem 1 is

$$n \geq 2\frac{L_{(K)}}{\mu}\log\left(2\frac{f(x_0) - f^*}{\mu\varepsilon}\right).$$

  The bound on SNAG from Theorem 9 is better if $\frac{L_{(K)}}{\mu} \leq \frac{4}{3}$. In realistic setting, $\frac{L_{(K)}}{\mu} \gg 1$. In particular if $f$ is quadratic strongly convex, to ensure that the bound of SNAG is better, we need to ensure that $\frac{\lambda_{\max}}{\lambda_{\min}} \leq \frac{4}{3}$, where $\lambda_{\max}$ and $\lambda_{\min}$ are respectively the highest and the lowest eigenvalues of the Hessian matrix of $f$.

Using Assumptions 1-2 to study the convergence of SNAG almost always leads to a worst bound than SGD independently of the value of $L_{(K)}$, which is misleading. This is because, in this case, we have to take safer parameters for SNAG. For example in the convex case, $s_n \to 0$ in order to ensure convergence, compared to the Theorem using SGC (Theorem 3) where $s_n = s > 0$. This means that this is the SGC characterization of the noise that allows to stabilize SNAG and which tells us how to chose parameters such that the convergence is of the order $\mathcal{O}\left(n^{-2}\right)$. Our results of Section 4 indicate that what is behind, in the finite sum setting, is the question of measuring the correlation between gradients, that is encapsulated within the SGC.

**Comparison with Theorem 3**    Compared to Theorem 3 that makes use of SGC, in Theorem 9 we had to choose more conservative parameters in order to ensure convergence, leading to similar convergence bound as for SGD (Algorithm 1). The comparison of the bounds of Theorems 3-9 are similar to the ones of Remark 3.

**Remark 9.** *In Remark 3, we compare SGD and SNAG under different assumptions, in the sense that we assume that the SGC holds when studying convergence of SNAG, but not for SGD. In Section E.2, we see that using SGC to study SGD leads to convergence bounds that are always worse than for SNAG, which is misleading. In this Section, with Theorem 9, we show that not using SGC when studying SNAG leads to an opposite phenomenon, with bounds for SGD that are almost always better than for SNAG, which is also misleading. Indeed, in particular, as seen in Remark 3, in the convex case, using SGC to study SNAG allows to have acceleration over SGD in finite-time, as long as we aim for a small enough $\varepsilon$-precision. Also, in our experiments, e.g. on Figures 1-2, we see that both cases are possible, namely SGD outperforming SNAG or SNAG outperforming SGD.*

### F.1    Proof of Theorem 9, convex case

We first recall the SNAG algorithm (Algorithm 2), with step-size $s_n$ and $\beta = 1$

$$\begin{cases} y_n = \alpha_n x_n + (1 - \alpha_n)z_n \\ x_{n+1} = y_n - s_n\tilde{\nabla}_K(y_n) \\ z_{n+1} = z_n - \eta_n\tilde{\nabla}_K(y_n) \end{cases} \quad \text{(SNAG)}$$

with $\tilde{\nabla}_K(\cdot)$ defined in (1).

$$\frac{1}{2}\|z_{n+1} - x^*\|^2 = \frac{1}{2}\|z_n - x^*\|^2 + \frac{\eta_n^2}{2}\|\tilde{\nabla}_K(y_n)\|^2 + \eta_n\langle x^* - z_n, \tilde{\nabla}_K(y_n)\rangle \quad (70)$$

$$= \frac{1}{2}\|z_n - x^*\|^2 + \frac{\eta_n^2}{2}\|\tilde{\nabla}_K(y_n)\|^2 + \eta_n\langle x^* - y_n, \tilde{\nabla}_K(y_n)\rangle \quad (71)$$

$$+ \eta_n\frac{\alpha_n}{1 - \alpha_n}\langle x_n - y_n, \tilde{\nabla}_K(y_n)\rangle \quad (72)$$

Taking the expectation, using, under the assumption $s_n \leq \frac{1}{L}$, Lemma 4, we have

$$\mathbb{E}_n \left[ \frac{1}{2} \|z_{n+1} - x^*\|^2 \right] \leq \frac{1}{2} \|z_n - x^*\|^2 + \eta_n \langle x^* - y_n, \nabla f(y_n) \rangle \tag{73}$$

$$+ \eta_n \frac{\alpha_n}{1 - \alpha_n} \langle x_n - y_n, \nabla f(y_n) \rangle + \frac{\eta_n^2}{s_n} \mathbb{E}_n \left[ f(y_n) - f(x_{n+1}) \right] + \eta_n^2 \mathbb{E}_n \left[ \|\nabla f(y_n) - \tilde{\nabla}_K(y_n)\|^2 \right]. \tag{74}$$

Using convexity of $f$ on both scalar products, we get

$$\mathbb{E}_n \left[ \frac{1}{2} \|z_{n+1} - x^*\|^2 + \frac{\eta_n^2}{s_n} (f(x_{n+1}) - f^*) \right] \leq \frac{1}{2} \|z_n - x^*\|^2 + \eta_n \frac{\alpha_n}{1 - \alpha_n} (f(x_n) - f(y_n)) \tag{75}$$

$$+ \left( \frac{\eta_n^2}{s_n} - \eta_n \right) (f(y_n) - f^*) + \eta_n^2 \mathbb{E}_n \left[ \|\nabla f(y_n) - \tilde{\nabla}_K(y_n)\|^2 \right]. \tag{76}$$

In order to bound $\mathbb{E}_n \left[ \|\nabla f(y_n) - \tilde{\nabla}_K(y_n)\|^2 \right]$, we use Lemma 3

$$\mathbb{E}_n \left[ \frac{1}{2} \|z_{n+1} - x^*\|^2 + \frac{\eta_n^2}{s_n} (f(x_{n+1}) - f^*) \right] \leq \frac{1}{2} \|z_n - x^*\|^2 + \eta_n \frac{\alpha_n}{1 - \alpha_n} (f(x_n) - f^*)$$

$$+ \left( \frac{\eta_n^2}{s_n} - \eta_n \frac{\alpha_n}{1 - \alpha_n} - \eta_n + 2\eta_n^2 L_{(K)} \right) (f(y_n) - f^*). \tag{77}$$

We set

$$\frac{\eta_n^2}{s_n} = \frac{C}{L_{(K)}} (n + 1)^\alpha, \quad \eta_n \frac{\alpha_n}{1 - \alpha_n} = \frac{C}{L_{(K)}} n^\alpha,$$

for some positive constants $C$ and $\alpha$. We want to have $\alpha$ **to be the highest possible**, while having the factor behind $(f(y_n) - f^*)$ non-positive. We have

$$\frac{\eta_n^2}{s_n} = \frac{C}{L_{(K)}} (n + 1)^\alpha \Rightarrow \eta_n = \sqrt{\frac{C s_n}{L_{(K)}}} (n + 1)^{\frac{\alpha}{2}}. \tag{78}$$

And then

$$\eta_n \frac{\alpha_n}{1 - \alpha_n} = \frac{C}{L_{(K)}} n^\alpha \Rightarrow \frac{\alpha_n}{1 - \alpha_n} = \frac{1}{\eta_n} \frac{C}{L_{(K)}} n^\alpha \tag{79}$$

$$\Rightarrow \frac{\alpha_n}{1 - \alpha_n} = \sqrt{\frac{C}{L_{(K)} s_n}} \frac{n^\alpha}{(n + 1)^{\frac{\alpha}{2}}}. \tag{80}$$

We plug the above equations in the factor behind $(f(y_n) - f^*)$

$$\frac{\eta_n^2}{s_n} - \eta_n \frac{\alpha_n}{1 - \alpha_n} - \eta_n + 2\eta_n^2 L_{(K)} = \frac{C}{L_{(K)}} (n + 1)^\alpha - \frac{C}{L_{(K)}} n^\alpha \tag{81}$$

$$- \sqrt{\frac{C s_n}{L_{(K)}}} (n + 1)^{\frac{\alpha}{2}} + 2 s_n C (n + 1)^\alpha \tag{82}$$

$$= \frac{C}{L_{(K)}} \left( (n + 1)^\alpha - n^\alpha - \sqrt{\frac{L_{(K)} s_n}{C}} (n + 1)^{\frac{\alpha}{2}} \right. \tag{83}$$

$$\left. + 2 s_n L_{(K)} (n + 1)^\alpha \right). \tag{84}$$

Now, we note $s_n = \frac{C_1}{(n+1)^\beta}$. Then, we have

$$\frac{\eta_n^2}{s_n} - \eta_n \frac{\alpha_n}{1 - \alpha_n} - \eta_n + 2\eta_n^2 L_{(K)} = \frac{C}{L_{(K)}} \left( (n + 1)^\alpha - n^\alpha - \sqrt{\frac{L_{(K)} C_1}{C}} (n + 1)^{\frac{\alpha - \beta}{2}} \right.$$

$$\left. + 2 L_{(K)} C_1 (n + 1)^{\alpha - \beta} \right). \tag{85}$$

We have to ensure that the positive term are not of larger order than the negative ones. We have $(n+1)^\alpha - n^\alpha \approx \alpha n^{\alpha-1}$. So, to have that this term is controlled, we need to have

$$\begin{cases} 2(\alpha-1) \leq \alpha - \beta \\ \alpha - \beta \geq 2(\alpha - \beta) \end{cases} \tag{86}$$

The second constraint implies $\alpha \leq \beta$. As we want $\alpha$ to be maximal, we set $\alpha = \beta$. The first constraint implies $\alpha + \beta \leq 2$, we get $\alpha = \beta = 1$. Plugging these values into Equation (85)

$$\frac{\eta_n^2}{s_n} - \eta_n \frac{\alpha_n}{1-\alpha_n} - \eta_n + 2\eta_n^2 L_{(K)} = 0 \Rightarrow \frac{C}{L_{(K)}}\left(1 - \sqrt{\frac{L_{(K)}C_1}{C}} + 2L_{(K)}C_1\right) = 0. \tag{87}$$

$$\Rightarrow \sqrt{\frac{L_{(K)}C_1}{C}} = 1 + 2L_{(K)}C_1 \tag{88}$$

$$\Rightarrow C = \frac{C_1 L_{(K)}}{(1 + 2L_{(K)}C_1)^2} \tag{89}$$

We choose $C_1$ such that $C$ is maximal, and $\frac{C_1 L_{(K)}}{(1+2L_{(K)}C_1)^2}$ is maximized at $C_1 = \frac{1}{2L_{(K)}} \Rightarrow C = \frac{1}{8}$.

So we have

$$s_n = \frac{1}{2L_{(K)}}\frac{1}{n+1}, \quad \eta_n = \frac{1}{4L_{(K)}}, \tag{90}$$

and $\frac{\alpha_n}{1-\alpha_n} = \frac{1}{\eta_n}\frac{C}{L_{(K)}}n = \frac{n}{2}$, so $\alpha_n = \frac{n}{n+2}$. Note that as $L_{(K)} \geq L$, this choice of $s_n$ satisfies the constraint $s_n \leq \frac{1}{L}$, needed to apply Lemma 4. With these choices of parameters, taking expectation on Equation (77), we have

$$\mathbb{E}\left[\frac{1}{2}\|z_{n+1} - x^*\|^2 + \frac{(n+1)}{8L_{(K)}}(f(x_{n+1}) - f^*)\right] \leq \mathbb{E}\left[\frac{1}{2}\|z_n - x^*\|^2 + \frac{n}{8L_{(K)}}(f(x_n) - f(x^*))\right]. \tag{91}$$

By induction, we get

$$\mathbb{E}\left[\frac{1}{2}\|z_{n+1} - x^*\|^2 + \frac{(n+1)}{8L_{(K)}}(f(x_{n+1}) - f^*)\right] \leq \mathbb{E}\left[\frac{1}{2}\|z_0 - x^*\|^2\right] = \frac{1}{2}\|x_0 - x^*\|^2. \tag{92}$$

Now, as $\frac{1}{2}\|z_{n+1} - x^*\|^2 + \frac{(n+1)}{8L_{(K)}}(f(x_{n+1}) - f^*) \geq \frac{(n+1)}{8L_{(K)}}(f(x_{n+1}) - f^*)$, we get

$$\mathbb{E}\left[(f(x_{n+1}) - f^*)\right] \leq \frac{4L_{(K)}}{(n+1)}\|x_0 - x^*\|^2 \tag{93}$$

We see that, compared to the case where we assumed SGC, where we had a decrease of the order $\mathcal{O}\left(n^{-2}\right)$, the decrease here is of the order $\mathcal{O}\left(n^{-1}\right)$. In term of $\varepsilon$ solution, it leads to

$$n \geq \frac{4L_{(K)}}{\varepsilon}\|x_0 - x^*\|^2. \tag{94}$$

### F.2 Proof of Theorem 9, strongly convex case

We use the following Lyapunov function.

$$E_n := f(x_n) - f^* + \frac{\mu}{2}\|z_n - x^*\|^2 \tag{95}$$

Proceeding to the same computations as in the proof of Theorem 3 in the strongly convex case, we arrive to Equation (148), namely

$$\mathbb{E}_n[E_{n+1}] = \beta E_n + \mathbb{E}_n[f(x_{n+1}) - f^*] - \beta(f(x_n) - f^*) + \frac{\mu}{2}(1-\beta)\|y_n - x^*\|^2 \tag{96}$$

$$+ \frac{\mu}{2}\eta^2 \mathbb{E}_n\left[\|\tilde{\nabla}_K(y_n)\|^2\right] - \frac{\mu}{2}\beta(1-\beta)\left(\frac{\alpha}{1-\alpha}\right)^2\|y_n - x_n\|^2$$

$$- \frac{\alpha\beta\eta\mu}{1-\alpha}\langle\mathbb{E}_n\left[\tilde{\nabla}_K(y_n)\right], y_n - x_n\rangle - \mu\eta\langle\mathbb{E}_n\left[\tilde{\nabla}_K(y_n)\right], y_n - x^*\rangle.$$

Recall $\mathbb{E}_n\left[\tilde{\nabla}_K(y_n)\right] = \nabla f(y_n)$. By Lemma 4, and Lemma 3 we respectively have

$$\mathbb{E}_n\left[\|\tilde{\nabla}_K(y_n)\|^2\right] \leq \frac{2}{s_n}\mathbb{E}_n\left[(f(y_n) - f(x_{n+1}))\right] + 2\mathbb{E}_n\left[\|\nabla f(y_n) - \tilde{\nabla}_K(y_n)\|^2\right], \qquad (97)$$

and

$$\mathbb{E}\left[\|\nabla f(y_n) - \tilde{\nabla}_K(y_n)\|^2\right] \leq 2(L_{(K)} - \mu)(f(y_n) - f^*). \qquad (98)$$

So, by combining the two previous equations, we have

$$\mathbb{E}_n\left[\|\tilde{\nabla}_K(y_n)\|^2\right] \leq \frac{2}{s}\mathbb{E}_n\left[(f(y_n) - f(x_{n+1}))\right] + 4(L_{(K)} - \mu)(f(y_n) - f^*). \qquad (99)$$

Now, we inject this in (96), also using strong convexity.

$$\mathbb{E}_n\left[E_{n+1}\right] \leq \beta E_n + \mathbb{E}_n\left[f(x_{n+1}) - f^*\right] - \beta\left(f(x_n) - f^*\right) + \frac{\mu}{2}(1-\beta)\|y_n - x^*\|^2 \qquad (100)$$
$$+ \frac{\mu}{s}\eta^2\mathbb{E}_n\left[f(y_n) - f(x_{n+1})\right] + 2\mu\eta^2(L_{(K)} - \mu)(f(y_n) - f^*)$$
$$- \frac{\mu}{2}\beta(1-\beta)\left(\frac{\alpha}{1-\alpha}\right)^2\|y_n - x_n\|^2 - \frac{\alpha\beta\eta\mu}{1-\alpha}(f(y_n) - f(x_n)) - \mu\eta(f(y_n) - f^*)$$
$$- \frac{\mu^2\eta}{2}\|y_n - x^*\|^2.$$

Collecting terms and removing the $\|y_n - x_n\|^2$ term, we get

$$\mathbb{E}_n\left[E_{n+1}\right] \leq \beta E_n + \left(1 - \frac{\mu}{s}\eta^2\right)\mathbb{E}_n\left[f(x_{n+1}) - f^*\right] + \beta\left(\frac{\alpha\eta\mu}{1-\alpha} - 1\right)(f(x_n) - f^*) \qquad (101)$$
$$+ \frac{\mu}{2}(1 - \beta - \mu\eta)\|y_n - x^*\|^2 + \mu\eta\left(\frac{\eta}{s} - \frac{\alpha\beta}{1-\alpha} - 1 + 2\eta(L_{(K)} - \mu)\right)(f(y_n) - f^*)$$

We fix $s = \mu\eta^2$ and $\frac{\alpha}{1-\alpha} = \frac{1}{\eta\mu}$, which cancels $\mathbb{E}_n\left[f(x_{n+1}) - f^*\right]$ and the $f(x_n) - f^*$ terms. With these choices, we want

$$\frac{\eta}{s} - \frac{\alpha\beta}{1-\alpha} - 1 + 2\eta(L_{(K)} - \mu) = \frac{1}{\mu\eta} - \frac{\beta}{\mu\eta} - 1 + 2\eta(L_{(K)} - \mu) = 0 \qquad (102)$$
$$\Rightarrow 1 - \beta = \mu\eta(1 - 2\eta(L_{(K)} - \mu)) \qquad (103)$$

We want to maximize the right quantity with respect to $\eta$. It is maximized for $\eta = \frac{1}{4(L_{(K)} - \mu)}$, and in this case:

$$\beta = 1 - \mu\eta(1 - 2\eta(L_{(K)} - \mu)) = 1 - \frac{1}{4}\frac{\mu}{(L_{(K)} - \mu)}\left(1 - \frac{1}{2}\right) = 1 - \frac{1}{8}\frac{\mu}{(L_{(K)} - \mu)} \qquad (104)$$

Also with that choice of $\eta$ and $\beta$, we have $1 - \beta - \mu\eta = \frac{1}{8}\frac{\mu}{(L_{(K)} - \mu)} - \frac{1}{4}\frac{\mu}{(L_{(K)} - \mu)} = -\frac{1}{8}\frac{\mu}{(L_{(K)} - \mu)} < 0$, which ensures that we can control the $\|y_n - x^*\|^2$ term. Thus, with the choice of parameters

$$s = \frac{1}{16}\frac{\mu}{(L_{(K)} - \mu)^2}, \quad \beta = 1 - \frac{1}{8}\frac{\mu}{(L_{(K)} - \mu)}, \quad \eta = \frac{1}{4}\frac{1}{(L_{(K)} - \mu)}, \quad \alpha = \frac{1}{1 + \frac{1}{4}\frac{\mu}{(L_{(K)} - \mu)}}, \qquad (105)$$

we have

$$\mathbb{E}_n\left[E_{n+1}\right] \leq \left(1 - \frac{1}{8}\frac{\mu}{(L_{(K)} - \mu)}\right)E_n - \frac{1}{16}\frac{\mu^2}{(L_{(K)} - \mu)}\|y_n - x^*\|^2 \qquad (106)$$

Ignoring norm term, taking expectation and by induction

$$\mathbb{E}\left[E_n\right] \leq \left(1 - \frac{1}{8}\frac{\mu}{(L_{(K)} - \mu)}\right)^n \mathbb{E}\left[E_0\right] \qquad (107)$$

Finally, we just bound $E_0$, by strong convexity

$$E_0 = f(x_0) - f^* + \frac{\mu}{2}\|x_0 - x^*\|^2 \leq 2(f(x_0) - f^*), \tag{108}$$

as we assumed $x_0 = z_0$. Thus, we get

$$\mathbb{E}[f(x_n) - f^*)] \leq 2\left(1 - \frac{1}{8}\frac{\mu}{(L_{(K)} - \mu)}\right)^n \mathbb{E}\left[f(x_0) - f^*)\right] \tag{109}$$

We obtain a bound in term of $\varepsilon$ solution following the same reasoning as for the proof of Theorem 3 in the strongly convex case, and obtain jfathat such a solution is achieved if

$$n \geq 8\frac{L_{(K)} - \mu}{\mu}\log\left(\frac{2(f(x_0) - f^*)}{\varepsilon}\right) \tag{110}$$

## F.3 Additional Lemma

In this section, we prove Lemma 3. Then, we introduce and prove Lemma 4.

First, we recall the statement of Lemma 3.

**Lemma.** *Assume $f$ is such that assumptions (1)-(2) hold.*

*If $f$ is $\mu$-strongly convex, we have*

$$\mathbb{E}\left[\|\nabla f(x) - \tilde{\nabla}_K(x)\|^2\right] \leq 2(L_{(K)} - \mu)(f(x) - f^*). \tag{111}$$

*If $f$ is convex, we have*

$$\mathbb{E}\left[\|\nabla f(x) - \tilde{\nabla}_K(x)\|^2\right] \leq 2L_{(K)}(f(x) - f^*). \tag{112}$$

*Proof.* We have the elementary relation:

$$\mathbb{E}\left[\|\nabla f(x) - \tilde{\nabla}_K(x)\|^2\right] = \mathbb{E}\left[\|\tilde{\nabla}_K(x)\|^2\right] - \|\nabla f(x)\|^2. \tag{113}$$

If $f$ is $\mu$-strongly convex, it satisfies the Polyak-Łojasiewicz inequality (Necoara et al., 2019), namely

$$\|\nabla f(x)\|^2 \geq 2\mu(f(x) - f^*). \tag{114}$$

The convex case can be obtained by letting $\mu \to 0$. In this case, we only have $\|\nabla f(x)\|^2 \geq 0$. To conclude, we apply Lemma 2 and we get the result. $\qquad\square$

**Lemma 4.** *Assuming $f$ is $L$-smooth and $s_n \leq \frac{1}{L}$, iterates of SNAG give:*

$$\mathbb{E}_n\left[\|\tilde{\nabla}_K(y_n)\|^2\right] \leq \frac{2}{s_n}\mathbb{E}_n\left[(f(y_n) - f(x_{n+1}))\right] + 2\mathbb{E}_n\left[\|\nabla f(y_n) - \tilde{\nabla}_K(y_n)\|^2\right]. \tag{115}$$

*Proof.* Using smoothness, we get

$$f(x_{n+1}) \leq f(y_n) + \langle \nabla f(y_n), x_{n+1} - y_n\rangle + \frac{L}{2}\|x_{n+1} - y_n\|^2 \tag{116}$$

$$f(x_{n+1}) \leq f(y_n) - s_n\langle \nabla f(y_n), \tilde{\nabla}_K(y_n)\rangle + \frac{Ls_n^2}{2}\|\tilde{\nabla}_K(y_n)\|^2. \tag{117}$$

We have

$$\langle \nabla f(y_n), \tilde{\nabla}_K(y_n)\rangle = \langle \nabla f(y_n) - \tilde{\nabla}_K(y_n), \tilde{\nabla}_K(y_n)\rangle + \|\tilde{\nabla}_K(y_n)\|^2 \tag{118}$$

$$= -\|\nabla f(y_n) - \tilde{\nabla}_K(y_n)\|^2 + \|\tilde{\nabla}_K(y_n)\|^2 - \langle \nabla f(y_n) - \tilde{\nabla}_K(y_n), \nabla f(y_n)\rangle. \tag{119}$$

Note that taking conditional expectation with respect to $\mathcal{F}_n$, the scalar product in equation equation 119 gets cancelled. Inserting (119) in (117) and taking conditional expectation, we have

$$\mathbb{E}_n\left[f(x_{n+1})\right] \leq \mathbb{E}_n\left[f(y_n)\right] + s_n\mathbb{E}_n\left[\|\nabla f(y_n) - \tilde{\nabla}_K(y_n)\|^2\right] \tag{120}$$

$$- s_n\left(1 - \frac{Ls_n}{2}\right)\mathbb{E}_n\left[\|\tilde{\nabla}_K(y_n)\|^2\right] \tag{121}$$

$$\mathbb{E}_n\left[\|\tilde{\nabla}_K(y_n)\|^2\right] \leq \frac{2}{s_n}\mathbb{E}_n\left[f(y_n) - f(x_{n+1})\right] + 2\mathbb{E}_n\left[\|\nabla f(y_n) - \tilde{\nabla}_K(y_n)\|^2\right]. \tag{122}$$

Where we used in the second inequality that $s_n\left(1 - \frac{Ls_n}{2}\right) \geq \frac{1}{2}$, provided that $s_n \leq \frac{1}{L}$. $\qquad\square$

## G  CONVERGENCE OF SNAG WITH STRONG GROWTH CONDITION

In Sections G.1 and G.2, we provide for completeness a proof of Theorem 3, that is a similar result to the one from Vaswani et al. (2019). Our proof is a slightly simpler formulation of the algorithm. In Sections G.3 and G.4, we extend these results proving new almost sure convergences (Theorem 4), that are asymptotically better that the results in expectation.

In this Section G, we denote $\mathcal{F}_n$ the $\sigma$-algebra generated by the $n+1$ first iterates $\{x_i\}_{i=0}^n$ generated by SNAG (Algorithm 2), *i.e.* $\mathcal{F}_n = \sigma(x_0, \ldots, x_n)$. Also, we will note $\mathbb{E}_n[\cdot]$ the conditional expectation with respect to $\mathcal{F}_n$.

First, we present a technical result (Lemma 5) that will be useful in our proofs.

**Lemma 5.** *Assume $f$ is $L$-smooth, and that $\tilde{\nabla}_K$ verifies the SGC for $\rho_K \geq 1$. If $y_n$ and $x_{n+1}$ are generated by SNAG (Algorithm 2), then with $s = \frac{1}{L\rho_K}$*

$$\|\nabla f(y_n)\|^2 \leq 2L\rho_K \mathbb{E}_n\left[f(y_n) - f(x_{n+1})\right] \tag{123}$$

*where $\mathbb{E}_n$ stands for the conditional expectation with respect to $\mathcal{F}_n$.*

*Proof.* By $L$-smoothness, we have

$$f(x_{n+1}) \leq f(y_n) + \langle \nabla f(y_n), x_{n+1} - y_n \rangle + \frac{L}{2}\|x_{n+1} - y_n\|^2 \tag{124}$$

$$= f(y_n) - s\langle \nabla f(y_n), \tilde{\nabla}_K(y_n)\rangle + \frac{Ls^2}{2}\|\tilde{\nabla}_K(y_n)\|^2. \tag{125}$$

By taking conditional expectation,

$$\mathbb{E}_n\left[f(x_{n+1}) - f(y_n)\right] \leq -s\|\nabla f(y_n)\|^2 + \frac{Ls^2}{2}\mathbb{E}_n\left[\|\tilde{\nabla}_K(y_n)\|^2\right]. \tag{126}$$

Then, by strong growth condition (SGC), we have

$$\mathbb{E}_n\left[f(x_{n+1}) - f(y_n)\right] \leq s\left(\frac{L\rho_K}{2}s - 1\right)\|\nabla f(y_n)\|^2. \tag{127}$$

To maximize the decrease, we choose $s = \frac{1}{L\rho_K}$, leading to

$$\mathbb{E}_n\left[f(x_{n+1}) - f(y_n)\right] \leq -\frac{1}{2L\rho_K}\|\nabla f(y_n)\|^2. \tag{128}$$

$\square$

### G.1  CONVEX-EXPECTATION

In this section, we prove statement (8) of Theorem 3 which is the convergence rate of SNAG algorithm for a convex function.

We first recall the SNAG algorithm (Algorithm 2), with a fixed step-size $s$ and $\beta = 1$

$$\begin{cases} y_n = \alpha_n x_n + (1-\alpha_n)z_n \\ x_{n+1} = y_n - s\tilde{\nabla}_K(y_n) \\ z_{n+1} = z_n - \eta_n \tilde{\nabla}_K(y_n) \end{cases} \tag{SNAG}$$

with $\tilde{\nabla}_K(\cdot)$ defined in (1).

$$\frac{1}{2}\|z_{n+1} - x^*\|^2 = \frac{1}{2}\|z_n - x^*\|^2 + \frac{\eta_n^2}{2}\|\tilde{\nabla}_K(y_n)\|^2 + \eta_n\langle x^* - z_n, \tilde{\nabla}_K(y_n)\rangle \tag{129}$$

$$= \frac{1}{2}\|z_n - x^*\|^2 + \frac{\eta_n^2}{2}\|\tilde{\nabla}_K(y_n)\|^2 + \eta_n\langle x^* - y_n, \tilde{\nabla}_K(y_n)\rangle \tag{130}$$

$$+ \eta_n\frac{\alpha_n}{1-\alpha_n}\langle x_n - y_n, \tilde{\nabla}_K(y_n)\rangle. \tag{131}$$

After taking conditional expectation with respect to $\mathcal{F}_n$, by the convexity of $f$, SGC and Lemma 5, we have

$$\frac{1}{2}\mathbb{E}_n[\|z_{n+1} - x^*\|^2] \leq \frac{1}{2}\|z_n - x^*\|^2 - \eta_n(f(y_n) - f(x^*)) + \eta_n\frac{\alpha_n}{1 - \alpha_n}(f(x_n) - f(y_n)) \quad (132)$$

$$+ L\rho_K^2\eta_n^2\mathbb{E}_n[f(y_n) - f(x_{n+1})]. \quad (133)$$

We can reformulate as

$$\mathbb{E}_n[L\rho_K^2\eta_n^2\left(f(x_{n+1}) - f^*\right) + \frac{1}{2}\|z_{n+1} - x^*\|^2] \leq \eta_n\frac{\alpha_n}{1 - \alpha_n}(f(x_n) - f^*) + \frac{1}{2}\|z_n - x^*\|^2 \quad (134)$$

$$+ \left(L\rho_K^2\eta_n^2 - \eta_n - \eta_n\frac{\alpha_n}{1 - \alpha_n}\right)(f(y_n) - f^*). \quad (135)$$

We define parameters as

$$L\rho_K^2\eta_n^2 = \frac{C}{L}(n + 1)^2, \quad \eta_n\frac{\alpha_n}{1 - \alpha_n} = \frac{C}{L}n^2, \quad (136)$$

with $C \geq 0$.

This parameter setting implies $\eta_n = \frac{\sqrt{C}}{L\rho_K}(n + 1)$. Thus, we have

$$L\rho_K^2\eta_n^2 - \eta_n - \eta_n\frac{\alpha_n}{1 - \alpha_n} = \frac{C}{L}(2n + 1) - \frac{\sqrt{C}}{L\rho_K}(n + 1) \leq 0 \quad (137)$$

$$\Rightarrow \sqrt{C} \leq \frac{1}{\rho_K}\frac{n + 1}{2n + 1}. \quad (138)$$

As we have, for all $n \in \mathbb{N}$, $\frac{1}{2} \leq \frac{n+1}{2n+1} \leq 1$, at best we can set $\sqrt{C} = \frac{1}{2\rho_K}$. With this choice, we have $C = \frac{1}{4\rho_K^2}$, $\eta_n = \frac{1}{L\rho_K^2}\frac{n+1}{2}$ and

$$\frac{\alpha_n}{1 - \alpha_n} = \frac{C}{L}n^2\eta_n^{-1} = \frac{1}{2}\frac{n^2}{n + 1}. \quad (139)$$

This implies that $\alpha_n = \frac{\frac{n^2}{n+1}}{2 + \frac{n^2}{n+1}}$. With this choice of parameter, we have

$$\mathbb{E}_n\left[\frac{(n+1)^2}{4L\rho_K^2}(f(x_{n+1}) - f^*) + \frac{1}{2}\|z_{n+1} - x^*\|^2\right] \leq \frac{n^2}{4L\rho_K^2}(f(x_n) - f^*) + \frac{1}{2}\|z_n - x^*\|^2. \quad (140)$$

Finally, we get the convergence rate

$$\mathbb{E}[f(x_{n+1}) - f^*] \leq \frac{2L\rho_K^2}{(n+1)^2}\|x_0 - x^*\|^2. \quad (141)$$

To conclude:

$$\frac{2L\rho_K^2}{n^2}\|x_0 - x^*\|^2 \leq \varepsilon \Rightarrow \sqrt{\frac{2L}{\varepsilon}}\rho_K\|x_0 - x^*\| \leq n. \quad (142)$$

### G.2 STRONGLY CONVEX - EXPECTATION

In this section, we prove statement (9) of Theorem 3. Let us remind the algorithm

$$\begin{cases} y_n = \alpha_n x_n + (1 - \alpha_n)z_n \\ x_{n+1} = y_n - s\tilde{\nabla}_K(y_n) \\ z_{n+1} = \beta z_n + (1 - \beta)y_n - \eta_n\tilde{\nabla}_K(y_n) \end{cases} \quad \text{(SNAG)}$$

with $\tilde{\nabla}_K(\cdot)$ defined in (1). We introduce the following Lyapunov energy:

$$E_n = f(x_n) - f^* + \frac{\mu}{2}\|z_n - x^*\|^2 \quad (143)$$

We compute:

$$E_{n+1} - E_n = f(x_{n+1}) - f(x_n) + \frac{\mu}{2}\|z_{n+1} - x^*\|^2 - \frac{\mu}{2}\|z_n - x^*\|^2. \qquad (144)$$

We start considering the right term

$$
\begin{aligned}
\Delta_n &= \|z_{n+1} - x^*\|^2 - \|z_n - x^*\|^2 \\
&= \|\beta z_n + (1-\beta)y_n - \eta\tilde{\nabla}_K(y_n) - x^*\|^2 - \|z_n - x^*\|^2 \\
&= (\beta^2 - 1)\|z_n - x^*\|^2 + (1-\beta)^2\|y_n - x^*\|^2 + \eta^2\|\tilde{\nabla}_K(y_n)\|^2 \\
&\quad + 2\beta\langle z_n - x^*, (1-\beta)(y_n - x^*) - \eta\tilde{\nabla}_K(y_n)\rangle - 2(1-\beta)\eta\langle\tilde{\nabla}_K(y_n), y_n - x^*\rangle
\end{aligned}
$$

by construction of Algorithm 2. We now control the first scalar product: using the definition of Algorithm 2, we have $z_n = y_n + \frac{\alpha}{1-\alpha}(y_n - x_n)$, therefore

$$
\begin{aligned}
&\langle z_n - x^*, (1-\beta)(y_n - x^*) - \eta\tilde{\nabla}_K(y_n)\rangle \\
&= \langle y_n - x^*, (1-\beta)(y_n - x^*) - \eta\tilde{\nabla}_K(y_n)\rangle \\
&\quad + \frac{\alpha}{1-\alpha}\langle y_n - x_n, (1-\beta)(y_n - x^*) - \eta\tilde{\nabla}_K(y_n)\rangle \\
&= (1-\beta)\|y_n - x^*\|^2 - \eta\langle y_n - x^*, \tilde{\nabla}_K(y_n)\rangle - \frac{\alpha}{1-\alpha}\eta\langle y_n - x_n, \tilde{\nabla}_K(y_n)\rangle \\
&\quad + \frac{\alpha}{1-\alpha}(1-\beta)\langle y_n - x_n, y_n - x^*\rangle
\end{aligned}
$$

Now, applying the relation $2\langle a, b\rangle = \|a+b\|^2 - \|a\|^2 - \|b\|^2$ to $a = y_n - x^*$ and $b = \frac{\alpha}{1-\alpha}(y_n - x_n)$, we get

$$\frac{\alpha}{1-\alpha}\langle y_n - x_n, y_n - x^*\rangle = \frac{1}{2}\|z_n - x^*\|^2 - \frac{1}{2}\left(\frac{\alpha}{1-\alpha}\right)^2\|y_n - x_n\|^2 - \frac{1}{2}\|y_n - x^*\|^2, \quad (145)$$

so that

$$
\begin{aligned}
&\langle z_n - x^*, (1-\beta)(y_n - x^*) - \eta\tilde{\nabla}_K(y_n)\rangle \\
&= \frac{1-\beta}{2}\left(\|z_n - x^*\|^2 + \|y_n - x^*\|^2 - \left(\frac{\alpha}{1-\alpha}\right)^2\|y_n - x_n\|^2\right) - \eta\langle y_n - x^*, \tilde{\nabla}_K(y_n)\rangle \\
&\quad - \frac{\alpha}{1-\alpha}\eta\langle y_n - x_n, \tilde{\nabla}_K(y_n)\rangle
\end{aligned}
$$

and

$$
\begin{aligned}
\Delta_n &= -(1-\beta)\|z_n - x^*\|^2 + (1-\beta)\|y_n - x^*\|^2 + \eta^2\|\tilde{\nabla}_K(y_n)\|^2 \\
&\quad - \beta(1-\beta)\left(\frac{\alpha}{1-\alpha}\right)^2\|y_n - x_n\|^2 - 2\frac{\alpha\beta\eta}{1-\alpha}\langle\tilde{\nabla}_K(y_n), y_n - x_n\rangle \\
&\quad - 2\eta\langle\tilde{\nabla}_K(y_n), y_n - x^*\rangle.
\end{aligned}
$$

Reinjecting $\Delta_n$ in the expression of $E_{n+1} - E_n$ and by definition of $E_n$, we get

$$E_{n+1} - E_n = -(1-\beta)E_n + f(x_{n+1}) - f^* - \beta(f(x_n) - f^*) + \frac{\mu}{2}(1-\beta)\|y_n - x^*\|^2$$

$$+ \frac{\mu}{2}\eta^2\|\tilde{\nabla}_K(y_n)\|^2 - \frac{\mu}{2}\beta(1-\beta)\left(\frac{\alpha}{1-\alpha}\right)^2\|y_n - x_n\|^2 \qquad (146)$$

$$- \frac{\alpha\beta\eta\mu}{1-\alpha}\langle\tilde{\nabla}_K(y_n), y_n - x_n\rangle - \mu\eta\langle\tilde{\nabla}_K(y_n), y_n - x^*\rangle. \qquad (147)$$

We take the conditional expectation with respect to $\mathcal{F}_n$:

$$\mathbb{E}_n[E_{n+1}] = \beta E_n + \mathbb{E}_n[f(x_{n+1}) - f^*] - \beta(f(x_n) - f^*) + \frac{\mu}{2}(1-\beta)\|y_n - x^*\|^2 \qquad (148)$$

$$+ \frac{\mu}{2}\eta^2\mathbb{E}_n\left[\|\tilde{\nabla}_K(y_n)\|^2\right] - \frac{\mu}{2}\beta(1-\beta)\left(\frac{\alpha}{1-\alpha}\right)^2\|y_n - x_n\|^2$$

$$- \frac{\alpha\beta\eta\mu}{1-\alpha}\langle\mathbb{E}_n\left[\tilde{\nabla}_K(y_n)\right], y_n - x_n\rangle - \mu\eta\langle\mathbb{E}_n\left[\tilde{\nabla}_K(y_n)\right], y_n - x^*\rangle. \qquad (149)$$

Using strong convexity of $f$, the strong growth condition (SGC), and then Lemma 5, we have:

$$\mathbb{E}_n[E_{n+1}] \leq \beta E_n + \mathbb{E}_n[f(x_{n+1}) - f^*] - \beta(f(x_n) - f^*) + \frac{\mu}{2}(1-\beta)\|y_n - x^*\|^2$$

$$+ \mu L \rho_K^2 \eta^2 \mathbb{E}_n[f(y_n) - f(x_{n+1})] - \frac{\mu}{2}\beta(1-\beta)\left(\frac{\alpha}{1-\alpha}\right)^2 \|y_n - x_n\|^2 \qquad (150)$$

$$- \frac{\alpha\beta\eta\mu}{1-\alpha}(f(y_n) - f(x_n)) - \mu\eta(f(y_n) - f^*) - \frac{\mu^2\eta}{2}\|y_n - x^*\|^2. \qquad (151)$$

$$\leq \beta E_n + \left(1 - \mu L \rho_K^2 \eta^2\right)\mathbb{E}_n[f(x_{n+1}) - f^*] + \beta\left(\frac{\alpha\eta\mu}{1-\alpha} - 1\right)(f(x_n) - f^*)$$

$$+ \frac{\mu}{2}(1 - \beta - \mu\eta)\|y_n - x^*\|^2 + \mu\eta\left(L\rho_K^2\eta - \frac{\alpha\beta}{1-\alpha} - 1\right)(f(y_n) - f^*) \qquad (152)$$

We make the following choices: $\eta = \frac{1}{\sqrt{\mu L}\rho_K}$, $\beta = 1 - \mu\eta = 1 - \frac{1}{\rho_K}\sqrt{\frac{\mu}{L}}$ and $\frac{\alpha}{1-\alpha} = \frac{1}{\mu\eta} = \rho_K\sqrt{\frac{L}{\mu}} \Rightarrow \alpha = \frac{1}{1 + \frac{1}{\rho_K}\sqrt{\frac{\mu}{L}}}$. As we have:

$$L\rho_K^2\eta - \frac{\alpha\beta}{1-\alpha} - 1 = \rho_K\sqrt{\frac{L}{\mu}} - \rho_K\sqrt{\frac{L}{\mu}}(1 - \frac{1}{\rho_K}\sqrt{\frac{\mu}{L}}) - 1 = 0, \qquad (153)$$

these choices cancel all the terms. Thus we have:

$$\mathbb{E}_n[E_{n+1}] \leq \left(1 - \frac{1}{\rho_K}\sqrt{\frac{\mu}{L}}\right)E_n \Rightarrow \mathbb{E}[E_{n+1}] \leq \left(1 - \frac{1}{\rho_K}\sqrt{\frac{\mu}{L}}\right)^{n+1}E_0. \qquad (154)$$

Now, note that $E_0 = f(x_0) - f^* + \frac{\mu}{2}\|x_0 - x^*\|^2 \leq 2(f(x_0) - f^*)$, because $f$ is $\mu$-strongly convex. We deduce the following convergence rate:

$$\mathbb{E}[f(x_n) - f^*] \leq 2\left(1 - \frac{1}{\rho_K}\sqrt{\frac{\mu}{L}}\right)^n (f(x_0) - f^*). \qquad (155)$$

A sufficient condition on the number $n$ of iterations needed to achieve a precision $\varepsilon$ is then naturally given by

$$2\left(1 - \frac{1}{\rho_K}\sqrt{\frac{\mu}{L}}\right)^n (f(x_0) - f^*) \leq \varepsilon. \qquad (156)$$

By taking the log, we get

$$n\log\left(1 - \frac{1}{\rho_K}\sqrt{\frac{\mu}{L}}\right) + \log\left(\frac{2(f(x_0) - f^*)}{\varepsilon}\right) \leq 0. \qquad (157)$$

Or equivalently

$$n \geq \left|\log\left(1 - \frac{1}{\rho_K}\sqrt{\frac{\mu}{L}}\right)\right|^{-1}\log\left(\frac{2(f(x_0) - f^*)}{\varepsilon}\right). \qquad (158)$$

Using the inequality: $|\log(1-x)| \geq x$ for any $x \in (0,1)$, we observe that

$$\left|\log\left(1 - \frac{1}{\rho_K}\sqrt{\frac{\mu}{L}}\right)\right|^{-1}\log\left(\frac{2(f(x_0) - f^*)}{\varepsilon}\right) \leq \rho_K\sqrt{\frac{L}{\mu}}\log\left(\frac{2(f(x_0) - f^*)}{\varepsilon}\right) \qquad (159)$$

Hence a sufficient condition on the number of iterations to reach a given precision $\varepsilon$ is

$$n \geq \rho_K\sqrt{\frac{L}{\mu}}\log\left(\frac{2(f(x_0)-f^*)}{\varepsilon}\right)$$

$$\Rightarrow n \geq \left|\log\left(1 - \frac{1}{\rho_K}\sqrt{\frac{\mu}{L}}\right)\right|^{-1}\log\left(\frac{2(f(x_0)-f^*)}{\varepsilon}\right)$$

$$\Rightarrow \mathbb{E}[f(x_n) - f^*] \leq \varepsilon \qquad (160)$$

### G.3 CONVEX - ALMOST SURE

In this section we extend statement (8) of Theorem 3 to get a new almost sure convergence rate.

**Proposition 5.** *Assume $f$ is $L$-smooth, convex, and that $\tilde{\nabla}_K$ verifies the SGC for $\rho_K \geq 1$. Then SNAG (Algorithm 2) with parameter setting $s = \frac{1}{\rho_K L}$, $\beta = 1$, $\alpha_n = \frac{\frac{n^2}{n+1}}{4 + \frac{n^2}{n+1}}$, $\eta_n = \frac{1}{4} \frac{n^2}{n+1}$ generates a sequence $\{x_n\}_{n \in \mathbb{N}}$ such that*

$$f(x_n) - f^* \stackrel{a.s.}{=} o\left(\frac{1}{n^2}\right). \tag{161}$$

This result is asymptotically better than the result in expectation of Theorem 3. This asymptotic speedup happens similarly considering the deterministic version of the algorithm (Attouch & Peypouquet, 2016).
Following the scheme of the proof of Theorem 3.1 in Sebbouh et al. (2021), the Theorem 10 is the key result of our proof.

**Theorem 10** (Robbins & Siegmund (1971)). *Let $V_n$, $A_n$, $B_n$ and $\alpha_n$ be positive sequences, adapted to some filtration $\mathcal{F}_n$. Assume the following inequality is verified for all $n \in \mathbb{N}$ :*

$$\mathbb{E}[V_{n+1} | \mathcal{F}_n] \leq V_n(1 + \alpha_n) + A_n - B_n \tag{162}$$

*Then, on the set $\{\sum_{i \geq 0} \alpha_i < +\infty, \sum_{i \geq 0} A_i < +\infty\}$, $V_n$ converges almost surely to a random variable $V_\infty$, and we also have $\sum_{i \geq 0} B_i < +\infty$.*

Note that the choice of parameters stated in Proposition 5 are less agressive (multiplied by a factor $\frac{1}{2}$) compared to the results in expectation (Theorem 8).

*Proof.* We start back from Equation (134) that we recall

$$\mathbb{E}_n[L\rho_K^2 \eta_n^2 (f(x_{n+1}) - f^*) + \frac{1}{2}\|z_{n+1} - x^*\|^2] \leq \eta_n \frac{\alpha_n}{1 - \alpha_n}(f(x_n) - f^*) + \frac{1}{2}\|z_n - x^*\|^2 \tag{163}$$

$$+ \left(L\rho_K^2 \eta_n^2 - \eta_n - \eta_n \frac{\alpha_n}{1 - \alpha_n}\right)(f(y_n) - f^*). \tag{164}$$

We set

$$L\rho_K^2 \eta_n^2 = \frac{C}{L}(n + 1)^2, \quad \eta_n \frac{\alpha_n}{1 - \alpha_n} = \frac{C}{L}n^2, \tag{165}$$

with $C \geq 0$.

In this proof, compared to the one of Theorem 3, we do not want to cancel the last term but to exploit it. More precisely, we are looking forward to the following inequality

$$L\rho_K^2 \eta_n^2 - \eta_n - \eta_n \frac{\alpha_n}{1 - \alpha_n} \leq -\frac{\eta_n}{2} \tag{166}$$

Hence:

$$\frac{C}{L}(2n + 1) \leq \frac{\eta_n}{2} = \frac{1}{2}\frac{\sqrt{C}}{L\rho_K}(n + 1) \quad \Leftrightarrow \quad \sqrt{C} \leq \frac{1}{2\rho_K}\frac{n + 1}{2n + 1} \tag{167}$$

$$\Leftrightarrow \quad C \leq \frac{1}{16\rho_K^2}\left(\frac{n + 1}{n + \frac{1}{2}}\right)^2. \tag{168}$$

With the choice $C = \frac{1}{16\rho_K^2}$, the last inequality is verified. Then, we get the following parameters

$$\eta_n = \frac{1}{\rho_K^2 L}\frac{n + 1}{4}, \quad \frac{\alpha_n}{1 - \alpha_n} = \frac{1}{4}\frac{n^2}{n + 1}. \tag{169}$$

The latter induces $\alpha_n = \frac{\frac{n^2}{n+1}}{4+\frac{n^2}{n+1}}$. Thus, we have

$$\mathbb{E}_n\left[\frac{(n+1)^2}{16L\rho_K^2}(f(x_{n+1}) - f^*) + \frac{1}{2}\|z_{n+1} - x^*\|^2\right] \leq \frac{n^2}{16L\rho_K^2}(f(x_n) - f^*) + \frac{1}{2}\|z_n - x^*\|^2$$
$$- \frac{\eta_n}{2}(f(y_n) - f^*).$$

We can now apply Theorem 10 with

$$V_n := \frac{n^2}{16L\rho_K^2}(f(x_n) - f^*) + \frac{1}{2}\|z_n - x^*\|^2,$$
$$A_n := 0,$$
$$B_n := \frac{\eta_n}{2}(f(y_n) - f^*),$$
$$\alpha_n := 0.$$

So we have almost surely

$$\sum_{n\geq 0} B_n = \frac{1}{8\rho_K^2 L}\sum_{n\geq 0}(n+1)(f(y_n) - f^*) < +\infty, \tag{170}$$

which implies that

$$\sum_{n\geq 0}(n+1)(f(y_n) - f^*) < +\infty. \tag{171}$$

By the definition of Algorithm 2 ($\beta = 1$ in the convex setting), we have

$$x_n - z_n = \alpha_{n-1}(x_{n-1} - z_{n-1}) + (\eta_{n-1} - s)\tilde{\nabla}_K(y_{n-1}). \tag{172}$$

Moreover, we have

$$x_n - y_n = (1 - \alpha_n)(x_n - z_n). \tag{173}$$

By combining Equation (172) and Equation (173), we get

$$\|x_{n+1} - y_{n+1}\|^2 = (1 - \alpha_{n+1})^2\left(\frac{\alpha_n}{1 - \alpha_n}\right)^2\|x_n - y_n\|^2 + (1 - \alpha_{n+1})^2(\eta_n - s)^2\|\tilde{\nabla}_K(y_n)\|^2 \tag{174}$$

$$+ 2(1 - \alpha_{n+1})^2\frac{\alpha_n}{1 - \alpha_n}(\eta_n - s)\langle x_n - y_n, \tilde{\nabla}_K(y_n)\rangle. \tag{175}$$

By taking the expectation with respect to $\mathcal{F}_n$, and using $(\eta_n - s)^2 \leq \eta_n^2$ for $n$ large enough, we have

$$\mathbb{E}_n[\|x_{n+1} - y_{n+1}\|^2] \leq (1 - \alpha_{n+1})^2\left(\frac{\alpha_n}{1 - \alpha_n}\right)^2\|x_n - y_n\|^2 \tag{176}$$

$$+ (1 - \alpha_{n+1})^2\eta_n^2\mathbb{E}_n[\|\tilde{\nabla}_K(y_n)\|^2] + 2(1 - \alpha_{n+1})^2\frac{\alpha_n}{1 - \alpha_n}(\eta_n - s)\langle x_n - y_n, \nabla f(y_n)\rangle. \tag{177}$$

Using the convexity of $f$, we have $\langle x_n - y_n, \nabla f(y_n)\rangle \leq f(x_n) - f(y_n)$. Thanks to SGC and Lemma 5, we have

$$\mathbb{E}_n[\|x_{n+1} - y_{n+1}\|^2] \leq (1 - \alpha_{n+1})^2\left(\frac{\alpha_n}{1 - \alpha_n}\right)^2\|x_n - y_n\|^2 \tag{178}$$

$$+ 2(1 - \alpha_{n+1})^2\eta_n^2 L\rho_K^2\mathbb{E}_n[f(y_n) - f(x_{n+1})] \tag{179}$$

$$+ 2(1 - \alpha_{n+1})^2\frac{\alpha_n}{1 - \alpha_n}(\eta_n - s)(f(x_n) - f(y_n)). \tag{180}$$

We divide the previous inequality by $(1 - \alpha_{n+1})^2$

$$\mathbb{E}_n \left[ \frac{\|x_{n+1} - y_{n+1}\|^2}{(1 - \alpha_{n+1})^2} \right] \leq \left( \frac{\alpha_n}{1 - \alpha_n} \right)^2 \|x_n - y_n\|^2 + 2\eta_n^2 L \rho_K^2 \mathbb{E}_n[f(y_n) - f(x_{n+1})] \quad (181)$$

$$+ 2\frac{\alpha_n}{1 - \alpha_n}(\eta_n - s)(f(x_n) - f(y_n)). \quad (182)$$

Thus

$$\mathbb{E}_n \left[ \frac{\|x_{n+1} - y_{n+1}\|^2}{(1 - \alpha_{n+1})^2} \right] \leq \left( \frac{\alpha_n}{1 - \alpha_n} \right)^2 \|x_n - y_n\|^2 + 2\eta_n^2 L \rho_K^2 \mathbb{E}_n[(f^* - f(x_{n+1}))] \quad (183)$$

$$+ 2\frac{\alpha_n}{1 - \alpha_n}(\eta_n - s)(f(x_n) - f^*) \quad (184)$$

$$+ 2\left( \eta_n^2 L \rho_K^2 - \frac{\alpha_n}{1 - \alpha_n}(\eta_n - s) \right)(f(y_n) - f^*). \quad (185)$$

By the parameter setting (Equation (169)) and the step-size $s = \frac{1}{\rho_K L}$, we have

$$\eta_n^2 L \rho_K^2 - \frac{\alpha_n}{1 - \alpha_n}(\eta_n - s) = \frac{1}{16}\frac{2n + 1}{L\rho_K^2} + \frac{1}{L\rho_K}\frac{1}{4}\frac{n^2}{n + 1} = \mathcal{O}(n). \quad (186)$$

By setting $C_n := 2\left( \eta_n^2 L \rho_K^2 - \frac{\alpha_n}{1 - \alpha_n}(\eta_n - s) \right)(f(y_n) - f^*)$, Equation (171) and Equation (186) gives that almost surely

$$\sum_n C_n < +\infty. \quad (187)$$

By defining $\lambda_n := \frac{1}{1 - \alpha_n}$ and the parameter setting (Equation (169)), Equation 183 can be transformed into

$$\mathbb{E}_n[\lambda_{n+1}^2 \|x_{n+1} - y_{n+1}\|^2 + \frac{1}{8}\frac{(n + 1)^2}{L\rho_K^2}(f(x_{n+1}) - f^*)] \quad (188)$$

$$\leq (1 - \lambda_n)^2 \|x_n - y_n\|^2 + \frac{1}{8}\frac{n^2}{L\rho_K^2}(f(x_n) - f^*) - \frac{n^2}{2L\rho_K(n + 1)}(f(x_n) - f^*) + C_n \quad (189)$$

$$\leq \lambda_n^2 \|x_n - y_n\|^2 + \frac{1}{8}\frac{n^2}{L\rho_K^2}(f(x_n) - f^*) - \frac{n^2}{2L\rho_K(n + 1)}(f(x_n) - f^*) + C_n \quad (190)$$

$$- (2\lambda_n - 1)\|x_n - y_n\|^2. \quad (191)$$

Recalling $\sum_n C_n < +\infty$, we then use Theorem 10 with

$$\tilde{V}_n := \lambda_n^2 \|x_n - y_n\|^2 + \frac{1}{8}\frac{n^2}{L\rho_K^2}(f(x_n) - f^*),$$

$$\tilde{A}_n := C_n,$$

$$\tilde{B}_n := \frac{n^2}{2L\rho_K(n + 1)}(f(x_n) - f^*) + (2\lambda_n - 1)\|x_n - y_n\|^2,$$

$$\tilde{\alpha}_n := 0.$$

Note that $\lambda_n = \frac{1}{1 - \alpha_n} = \frac{4 + \frac{n^2}{n+1}}{4} \geq 1$, $2\lambda_n - 1 \geq \lambda_n \geq 0$ and $\tilde{B}_n$ is positive. So, we have that $\lim \tilde{V}_n := \tilde{V}_\infty$ exists almost surely, and $\sum_n \tilde{B}_n < +\infty$ almost surely. However, we have

$$\lambda_n \tilde{B}_n \geq \frac{\lambda_n n^2}{2L\rho_K(n + 1)}(f(x_n) - f^*) + \lambda_n^2 \|x_n - y_n\|^2. \quad (192)$$

Moreover, we can compute by the parameter setting (Equation 169)

$$\frac{\lambda_n n^2}{2L\rho_K(n + 1)} = \frac{4 + \frac{n^2}{n+1}}{4}\frac{1}{L\rho_K}\frac{1}{2}\frac{n^2}{n + 1} = \frac{n^2}{8L\rho_K^2}\frac{\rho_K(4 + \frac{n^2}{n+1})}{n + 1}. \quad (193)$$

As $\frac{\rho_K(4+\frac{n^2}{n+1})}{n+1} = \rho_K \frac{n^2+4n+4}{n^2+2n+1} > 1$, we have $\frac{\lambda_n n^2}{2L\rho_K(n+1)} > \frac{n^2}{8L\rho_K^2}$, and thus

$$\lambda_n \tilde{B}_n \geq \tilde{V}_n. \tag{194}$$

We can deduce from the previous inequality

$$\sum_{n\geq 0} \tilde{B}_n = \sum_{n\geq 0} \frac{1}{\lambda_n} \lambda_n \tilde{B}_n \geq \sum_{n\geq 0} \frac{1}{\lambda_n} \tilde{V}_n = 4\sum_{n\geq 0} \frac{\tilde{V}_n}{4+\frac{n^2}{n+1}} \tag{195}$$

As $\sum \tilde{B}_n < \infty$ almost surely, we have $\sum_{n\geq 0} \frac{\tilde{V}_n}{4+\frac{n^2}{n+1}} < +\infty$ almost surely, and necessarily $V_\infty = 0$ almost surely. Then, almost surely

$$\frac{1}{8}\frac{n^2}{L\rho_K^2}(f(x_n) - f^*) \overset{a.s.}{\to} 0. \tag{196}$$

Finally, we get the result of Proposition 5

$$f(x_n) - f^* \overset{a.s.}{=} o\left(\frac{1}{n^2}\right). \tag{197}$$

$\square$

## G.4 Strongly convex - almost sure

Similarly to Section G.3, we extend statement (8) of Theorem 3 to get a new, asymptotically better, almost sure convergence result.

**Proposition 6.** *Assume $f$ is $L$-smooth, $\mu$-strongly convex, and that $\tilde{\nabla}_K$ verifies the SGC for $\rho_K \geq 1$. Then SNAG (Algorithm 2) with parameter setting $\alpha = \frac{1}{1+\frac{1}{\rho_K}\sqrt{\frac{\mu}{L}}}$, $s = \frac{1}{\rho_K L}$, $\beta = 1 - \frac{1}{\rho_K}\sqrt{\frac{\mu}{L}}$ and $\eta = \frac{1}{\rho_K\sqrt{\mu L}}$ generates a sequence $(x_n)_{n\in\mathbb{N}}$ such that for all $\varepsilon > 0$, we have*

$$f(x_n) - f^* \overset{a.s.}{=} o\left((q+\varepsilon)^n\right), \tag{198}$$

$$\|z_n - x^*\|^2 \overset{a.s.}{=} o\left((q+\varepsilon)^n\right) \tag{199}$$

*where $q := 1 - \frac{1}{\rho_K}\sqrt{\frac{\mu}{L}}$.*

*Proof.* We use the following Lyapunov function

$$E_n := f(x_n) - f^* + \frac{\mu}{2}\|z_n - x^*\|^2. \tag{200}$$

We set $q := 1 - \frac{1}{\rho_K}\sqrt{\frac{\mu}{L}}$. We fix $\varepsilon' > 0$. By the Markov inequality and Equation (154), we get

$$\mathbb{P}\left(E_n \geq (q+\varepsilon')^n E_0\right) \leq \frac{\mathbb{E}[E_n]}{(q+\varepsilon')^n E_0} \leq \left(\frac{q}{q+\varepsilon'}\right)^n. \tag{201}$$

We sum on $n \geq 0$

$$\sum_{n\geq 0} \mathbb{P}\left(E_n \geq (q+\varepsilon')^n E_0\right) \leq \sum_{n\geq 0} \left(\frac{q}{q+\varepsilon'}\right)^n < +\infty. \tag{202}$$

Setting $A_n := \{E_n \geq (q+\varepsilon')^n E_0\}$, we have by the Borel Cantelli Lemma that $\mathbb{P}(\limsup A_n) = 0$, which implies $\mathbb{P}(\liminf A_n^c) = 1$, where $A_n^c$ is the complementary of $A_n$. In other words, as $A_n^c := \{E_n < (q+\varepsilon')^n E_0\}$, then for almost every $\omega \in \Omega$, $\exists N_0(\omega) \in \mathbb{N}$ such that for all $n \geq N_0(\omega)$, we have

$$E_n(\omega) < (q+\varepsilon')^n E_0. \tag{203}$$

Thus, we have

$$\frac{E_n(\omega)}{(q+2\varepsilon')^n} < \left(\frac{q+\varepsilon'}{q+2\varepsilon'}\right)^n E_0 \underset{n\to+\infty}{\to} 0 \tag{204}$$

The right term is independent of $\omega$, so almost surely, we have

$$E_n = o\left((q + 2\varepsilon')^n\right)$$

Now fix $\varepsilon = 2\varepsilon'$ and we get

$$E_n = o\left((q + \varepsilon)^n\right), \tag{205}$$

and thus the result by definition of $E_n$. $\qquad\square$

## H    PROOFS OF SECTION 4

In this section, we will prove our results that establish a link between the strong growth condition SGC and the average correlation between gradients, by exploiting the finite sum structure (FS).

### H.1    PROOF OF PROPOSITION 1

In order to demonstrate Proposition 1, we first establish Lemma 6.

**Lemma 6.** *Let $\{a_i\}_{i=1}^N$ be a sequence of vectors in $\mathbb{R}^d$ and $K \in \mathbb{N}$. We define $\mathcal{B}(K, N) = \{B \subset \{1, \ldots, N\} | Card(B) = K\}$. Then, we have*

$$\|\sum_{i=1}^N a_i\|^2 = \frac{N}{K}\frac{1}{\binom{N}{K}} \sum_{B \in \mathcal{B}(K,N)} \|\sum_{i \in B} a_i\|^2 + 2\frac{N-K}{N-1} \sum_{1 \le i < j \le N} \langle a_i, a_j \rangle. \tag{206}$$

*Proof.* We fix $\{i_1, \ldots, i_k\} \in \mathcal{B}(K, N)$. We have

$$\| \sum_{i \in \{i_1, \ldots, i_k\}} a_i\|^2 = \|\sum_{i=1}^N a_i - \sum_{i \notin \{i_1, \ldots, i_k\}} a_i\|^2 \tag{207}$$

$$= \|\sum_{i=1}^N a_i\|^2 + \|\sum_{i \notin \{i_1, \ldots, i_k\}} a_i\|^2 - 2\sum_{i=1}^N \sum_{j \notin \{i_1, \ldots, i_k\}} \langle a_i, a_j \rangle \tag{208}$$

$$= \|\sum_{i=1}^N a_i\|^2 + \sum_{i \notin \{i_1, \ldots, i_k\}} \|a_i\|^2 + \sum_{\substack{i,j \notin \{i_1, \ldots, i_k\} \\ i \ne j}} \langle a_i, a_j \rangle - 2\sum_{i=1}^N \sum_{j \notin \{i_1, \ldots, i_k\}} \langle a_i, a_j \rangle. \tag{209}$$

We sum over all the possible $B = \{i_1, \ldots, i_k\} \in \mathcal{B}(K, N)$. Note that $|\mathcal{B}(K, N)| = \binom{N}{K}$. We split each term in Equation (209), first

$$\sum_{B \in \mathcal{B}(K,N)} \|\sum_{i=1}^N a_i\|^2 = \binom{N}{K}\|\sum_{i=1}^N a_i\|^2. \tag{210}$$

The sum of the second term of Equation (209) is

$$\sum_{B \in \mathcal{B}(K,N)} \sum_{i \notin \{i_1, \ldots, i_k\}} \|a_i\|^2 = \sum_{B \in \mathcal{B}(N-K,N)} \sum_{i \in \{i_1, \ldots, i_{n-k}\}} \|a_i\|^2 \tag{211}$$

$$= \binom{N-1}{N-K-1} \sum_{i=1}^N \|a_i\|^2 \tag{212}$$

$$= \binom{N}{K}\frac{N-K}{N} \sum_{i=1}^N \|a_i\|^2 \tag{213}$$

$$= \binom{N}{K}\frac{N-K}{N}\|\sum_{i=1}^N a_i\|^2 - \binom{N}{K}\frac{N-K}{N} \sum_{\substack{i,j=1 \\ i \ne j}}^N \langle a_i, a_j \rangle. \tag{214}$$

The second equality comes from how many times the index $i$ is picked by the sum. We thus count the number of set $\{i_1, \ldots, i_{n-k}\} \in \{1, \ldots, N\}^{n-k}$ such that $i$ belongs to this set. This amounts to compute the cardinal of the set $\{\{i, i_1, \ldots, i_{n-k-1}\}, i_1, \ldots, i_{n-k-1} \in \{1, \ldots, N\}\backslash\{i\}\}$. This set has the same size as $\mathcal{B}(N-K-1, N-1)$, which is $\binom{N-1}{N-K-1}$.

The sum of the third term of Equation (209) is

$$\sum_{B\in\mathcal{B}(K,N)} \sum_{\substack{i,j\notin\{i_1,\ldots,i_k\}\\i\neq j}} \langle a_i, a_j \rangle = \sum_{B\in\mathcal{B}(N-K,N)} \sum_{\substack{i,j\in\{i_1,\ldots,i_{n-k}\}\\i\neq j}} \langle a_i, a_j \rangle \tag{215}$$

$$= \binom{N-2}{N-K-2} \sum_{\substack{i,j=1\\i\neq j}}^{N} \langle a_i, a_j \rangle \tag{216}$$

$$= \binom{N}{K} \frac{(N-K)(N-K-1)}{N(N-1)} \sum_{\substack{i,j=1\\i\neq j}}^{N} \langle a_i, a_j \rangle. \tag{217}$$

Here, the second equality comes from the fact that we compute the size of the set $\{\{i, j, i_1, \ldots, i_{n-k-2}\}, i_1, \ldots, i_{n-k-2} \in \{1, \ldots, N\}\backslash\{i, j\}\}$, which is of the same size as $\mathcal{B}(N-K-2, N-2)$, which is $\binom{N-2}{N-K-2}$.

Finally, we compute the sum of the fourth term of Equation (209), using Equation (213)

$$\sum_{B\in\mathcal{B}(K,N)} \sum_{i=1}^{N} \sum_{j\notin\{i_1,\ldots,i_k\}} \langle a_i, a_j \rangle = \sum_{i=1}^{N} \left\langle a_i, \sum_{B\in\mathcal{B}(K,N)} \sum_{j\notin\{i_1,\ldots,i_k\}} a_j \right\rangle \tag{218}$$

$$= \sum_{i=1}^{N} \left\langle a_i, \binom{N}{K} \frac{N-K}{N} \sum_{j=1}^{N} a_j \right\rangle \tag{219}$$

$$= \binom{N}{K} \frac{N-K}{N} \|\sum_{i=1}^{N} a_i\|^2. \tag{220}$$

Now that we have computed the sum of each terms in Equation (209), we have

$$\sum_{B\in\mathcal{B}(K,N)} \|\sum_{i\in\{i_1,\ldots,i_k\}} a_i\|^2 = \binom{N}{K} \|\sum_{i=1}^{N} a_i\|^2 + \binom{N}{K} \frac{N-K}{N} \left( \|\sum_{i=1}^{N} a_i\|^2 - \sum_{\substack{i,j=1\\i\neq j}}^{N} \langle a_i, a_j \rangle \right) \tag{221}$$

$$+ \binom{N}{K} \frac{(N-K)(N-K-1)}{N(N-1)} \sum_{\substack{i,j=1\\i\neq j}}^{N} \langle a_i, a_j \rangle - 2\binom{N}{K} \frac{N-K}{N} \|\sum_{i=1}^{N} a_i\|^2 \tag{222}$$

$$= \binom{N}{K} \left(1 - \frac{N-K}{N}\right) \|\sum_{i=1}^{N} a_i\|^2 + \binom{N}{K} \frac{N-K}{N} \left( \frac{N-K-1}{N-1} - 1 \right) \sum_{\substack{i,j=1\\i\neq j}}^{N} \langle a_i, a_j \rangle \tag{223}$$

$$= \binom{N}{K} \frac{K}{N} \|\sum_{i=1}^{N} a_i\|^2 - \binom{N}{K} \frac{K}{N} \frac{N-K}{N-1} \sum_{\substack{i,j=1\\i\neq j}}^{N} \langle a_i, a_j \rangle. \tag{224}$$

By rearranging the terms, we obtain the desired result

$$\|\sum_{i=1}^{N} a_i\|^2 = \frac{N}{K} \frac{1}{\binom{N}{K}} \sum_{B\in\mathcal{B}(K,N)} \|\sum_{i\in\{i_1,\ldots,i_k\}} a_i\|^2 + \frac{N-K}{N-1} \sum_{\substack{i,j=1\\i\neq j}}^{N} \langle a_i, a_j \rangle. \tag{225}$$

$\square$

The proof the Proposition 1 is simply an application of Lemma 6 with $a_i = \frac{1}{N}\nabla f_i(x)$. We obtain

$$\|\nabla f(x)\|^2 = \|\frac{1}{N}\sum_{i=1}^{N}\nabla f_i(x)\|^2 \tag{226}$$

$$= \frac{N}{K}\frac{1}{\binom{N}{K}}\sum_{B\in\mathcal{B}(K,N)}\|\frac{1}{N}\sum_{i\in\{i_1,\ldots,i_k\}}\nabla f_i(x)\|^2 + \frac{N-K}{N-1}\frac{1}{N^2}\sum_{\substack{i,j=1\\i\neq j}}^{N}\langle\nabla f_i(x),\nabla f_j(x)\rangle \tag{227}$$

$$= \frac{K}{N}\frac{1}{\binom{N}{K}}\sum_{B\in\mathcal{B}(K,N)}\|\frac{1}{K}\sum_{i\in\{i_1,\ldots,i_k\}}\nabla f_i(x)\|^2 + \frac{N-K}{N-1}\frac{1}{N^2}\sum_{\substack{i,j=1\\i\neq j}}^{N}\langle\nabla f_i(x),\nabla f_j(x)\rangle \tag{228}$$

$$= \frac{K}{N}\mathbb{E}[\|\tilde{\nabla}_K(x)\|^2] + \frac{N-K}{N-1}\frac{1}{N^2}\sum_{\substack{i,j=1\\i\neq j}}^{N}\langle\nabla f_i(x),\nabla f_j(x)\rangle \tag{229}$$

$$= \frac{K}{N}\mathbb{E}\left[\|\tilde{\nabla}_K(x)\|^2\right] + \frac{2}{N^2}\frac{N-K}{N-1}\sum_{1\leq i<j\leq N}\langle\nabla f_i(x),\nabla f_j(x)\rangle. \tag{230}$$

## H.2 PROOF OF PROPOSITION 2

In this part, we demonstrate Proposition 2. The result is a direct consequence of the RACOGA condition. Indeed, considering batch of size 1, by Proposition 1 we have $\forall x \in \mathbb{R}^d$

$$\|\nabla f(x)\|^2 = \frac{1}{N}\mathbb{E}\left[\|\tilde{\nabla}_1(x)\|^2\right] + \frac{2}{N^2}\sum_{1\leq i<j\leq N}\langle\nabla f_i(x),\nabla f_j(x)\rangle. \tag{231}$$

Now recall the RACOGA condition

$$\forall x \in \mathbb{R}^d, \quad \frac{\sum_{1\leq i<j\leq N}\langle\nabla f_i(x),\nabla f_j(x)\rangle}{\sum_{i=1}^{N}\|\nabla f_i(x)\|^2} \geq c. \tag{RACOGA}$$

We inject RACOGA in Equation (231) to get

$$\|\nabla f(x)\|^2 \geq \frac{1}{N}\mathbb{E}\left[\|\tilde{\nabla}_1(x)\|^2\right] + c\frac{2}{N^2}\sum_{i=1}^{N}\|\nabla f_i(x)\|^2 \tag{232}$$

$$= \frac{1}{N}\mathbb{E}\left[\|\tilde{\nabla}_1(x)\|^2\right] + c\frac{2}{N}\mathbb{E}\left[\|\tilde{\nabla}_1(x)\|^2\right] \tag{233}$$

$$= \frac{1}{N}\left(1 + 2c\right)\mathbb{E}\left[\|\tilde{\nabla}_1(x)\|^2\right]. \tag{234}$$

From Equation (234), that holds $\forall x \in \mathbb{R}^d$, we deduce that $f$ satisfy SGC with $\rho_1 \leq \frac{N}{1+2c}$.

## H.3 PROOF OF LEMMA 1

In this part, we demonstrate Lemma 1. Assume that for batches of size 1, $f$ verifies a $\rho_1$-SGC, *i.e.*

$$\forall x \in \mathbb{R}^d, \quad \frac{1}{N}\sum_{i=1}^{N}\|\nabla f_i(x)\|^2 \leq \rho_1\|\nabla f(x)\|^2. \tag{235}$$

By Proposition 1, we have

$$\|\nabla f(x)\|^2 = \frac{K}{N}\mathbb{E}\left[\|\tilde{\nabla}_K(x)\|^2\right] + \frac{2}{N^2}\frac{N-K}{N-1}\sum_{1\leq i<j\leq N}\langle\nabla f_i(x),\nabla f_j(x)\rangle. \tag{236}$$

Moreover, by developing the squared norm of $\nabla f(x)$ and rearranging, we get

$$\frac{1}{N} \sum_{1 \leq i < j \leq N} \langle \nabla f_i(x), \nabla f_j(x) \rangle = \frac{N}{2} \|\nabla f(x)\|^2 - \frac{1}{2N} \sum_{i=1}^{N} \|\nabla f_i(x)\|^2, \tag{237}$$

hence, by reinjecting (237) into (236),

$$
\begin{aligned}
\mathbb{E}\left[\|\tilde{\nabla}_K(x)\|^2\right] &= \frac{N}{K}\left(1 - \frac{N-K}{N-1}\right)\|\nabla f(x)\|^2 + \frac{1}{NK}\frac{N-K}{N-1}\sum_{i=1}^{N}\|\nabla f_i(x)\|^2 & (238)\\
&\leq \frac{N}{K}\left(1 - \frac{N-K}{N-1} + \frac{1}{N}\frac{N-K}{N-1}\rho_1\right)\|\nabla f(x)\|^2 & (239)\\
&\leq \frac{1}{K(N-1)}\left(N(K-1) + (N-K)\rho_1\right)\|\nabla f(x)\|^2 & (240)
\end{aligned}
$$

using the SGC assumption for batches of size 1.

Finally we deduce Lemma 1: the SGC is verified for every size of batch $K \geq 1$ and we have

$$\rho_K \leq \frac{1}{K(N-1)}\left(\rho_1(N-K) + (K-1)N\right). \tag{241}$$

**Remark 10.** *Lemma 1 offers an indirect way to demonstrate Corollary 1. Indeed, the result of Proposition 1 for the special case of batches of size 1 is easily computed, as we have*

$$\|\nabla f(x)\|^2 = \frac{1}{N^2}\sum_{i=1}^{N}\|\nabla f_i(x)\|^2 + \frac{2}{N^2}\sum_{1 \leq i < j \leq N}\langle \nabla f_i(x), \nabla f_j(x) \rangle. \tag{242}$$

*Thus, when PosCorr is verified, with batches of size 1, $f$ verifies SGC with constant $\rho_1 = N$. By applying Lemma 1 with $\rho_1 = N$, we get that for batches of size $K$, $f$ verifies SGC) with constant*

$$\rho_K \leq \frac{1}{K(N-1)}\left(N(N-K) + (K-1)N\right) = \frac{N}{K}. \tag{243}$$

*For the clarity of our presentation, we choose to present Proposition 1 before Lemma 1, even if it can be seen as a corollary of this result.*

One can obtain a reciprocal result of Lemma 1.

**Proposition 7.** *Assume that for batches of size $K$, $f$ verifies the SGC with constant $\rho_K$. Then, for batches of size 1, $f$ verifies the SGC with a constant $\rho_1$ which satisfies*

$$\rho_1 \leq \frac{K\rho_K(N-1) - N(K-1)}{N-K}. \tag{244}$$

*Proof.* From Equality (238), we have

$$\mathbb{E}\left[\|\tilde{\nabla}_K(x)\|^2\right] = \frac{1}{KN}\frac{N-K}{N-1}\sum_{i=1}^{N}\|\nabla f_i(x)\|^2 + \frac{K-1}{K}\frac{N}{N-1}\|\nabla f(x)\|^2. \tag{245}$$

Using the SGC, we obtain

$$\frac{1}{K}\frac{N-K}{N-1}\frac{1}{N}\sum_{i=1}^{N}\|\nabla f_i(x)\|^2 \leq \left(\rho_K - \frac{K-1}{K}\frac{N}{N-1}\right)\|\nabla f(x)\|^2. \tag{246}$$

So, the SGC is verified with batches of size 1 and (244) is proved. $\qquad\square$

**Remark 11.** *Proposition 7 indicates that satisfying the SGC for batches of size $K$ implies interpolation (Assumption 1) if each $f_i$ is convex. Indeed, Proposition 7 induces that the SGC is verified for batches of size 1 with some $\rho_1 \geq 1$, i.e.*

$$\forall x \in \mathbb{R}^d, \ \frac{1}{N} \sum_{i=1}^{N} \|\nabla f_i(x)\|^2 \leq \rho_1 \|\nabla f(x)\|^2. \tag{247}$$

*Interpolation is then a direct consequence of evaluating Inequality (247) at some $x^* \in \arg\min f$. Indeed, it implies that each minimizer of $f$ is a critical point of each $f_i$. Convexity of the $f_i$ allow to conclude.*

## H.4 Proof of Theorem 5

In this section, we demonstrate Theorem 5. Recall that we assume $f$ is convex, $L$-smooth, and that $\tilde{\nabla}_1$ verify SGC for $\rho_1 \geq 1$. We know that SNAG with good choice of parameters (see Theorem 3) guarantees to reach an $\varepsilon$-precision (2) if

$$n \geq \rho_K \sqrt{\frac{2L}{\varepsilon}} \|x_0 - x^*\|. \tag{248}$$

Note that by Lemma 1, we know that $\rho_k$, the SGC constant associated to $\tilde{\nabla}_K$, exists, and that

$$\rho_K \leq \frac{1}{K(N-1)} \left( \rho_1(N-K) + (K-1)N \right). \tag{249}$$

So, in particular, we are ensured to reach an $\varepsilon$-precision if

$$n \geq \frac{1}{K(N-1)} \left( \rho_1(N-K) + (K-1)N \right) \sqrt{\frac{2L}{\varepsilon}} \|x_0 - x^*\| \tag{250}$$

$$= \underbrace{\left( \frac{N-K}{N-1} + \frac{N}{\rho_1} \frac{K-1}{N-1} \right)}_{:=\Delta_K} \frac{\rho_1}{K} \sqrt{\frac{2L}{\varepsilon}} \|x_0 - x^*\|. \tag{251}$$

Now, to translate the result in term of number of $\nabla f_i$ evaluated, note that each iteration requires to evaluate $K$ different $\nabla f_i$, because we consider batches of size $K$. Finally, the number of $\nabla f_i$ to evaluate is

$$\Delta_K \rho_1 \sqrt{\frac{2L}{\varepsilon}} \|x_0 - x^*\|. \tag{252}$$

## H.5 Bound on RACOGA

In this section, we bound the ratio $\frac{\sum_{1 \leq i < j \leq N} \langle \nabla f_i(x), \nabla f_j(x) \rangle}{\sum_{i=1}^{N} \|\nabla f_i(x)\|^2}$, that is involved in the RACOGA condition.

**Proposition 8.** *Let $x \in \mathbb{R}^d \backslash \overline{\mathcal{X}}$. We have*

$$-\frac{1}{2} \leq \frac{\sum_{1 \leq i < j \leq N} \langle \nabla f_i(x), \nabla f_j(x) \rangle}{\sum_{i=1}^{N} \|\nabla f_i(x)\|^2} \leq \frac{N-1}{2}. \tag{253}$$

*where $\overline{\mathcal{X}} = \{x \in \mathbb{R}^d, \forall i \in \{1, \ldots, N\}, \|\nabla f_i(x)\| = 0\}$.*

*Proof.* The upper-bound relies on the inequality $\langle a, b \rangle \leq \frac{1}{2}\|a\|^2 + \frac{1}{2}\|b\|^2$, for $a, b \in \mathbb{R}^d$.

$$\sum_{1 \leq i < j \leq N} \langle \nabla f_i(x), \nabla f_j(x) \rangle \leq \frac{1}{2} \sum_{1 \leq i < j \leq N} \left( \|\nabla f_i(x)\|^2 + \|\nabla f_j(x)\|^2 \right) \tag{254}$$

$$= \frac{N-1}{2} \sum_{i=1}^{N} \|\nabla f_i(x)\|^2. \tag{255}$$

Using $\langle a, b \rangle \geq -\frac{1}{2}\|a\|^2 - \frac{1}{2}\|b\|^2$, for $a, b \in \mathbb{R}^d$, we could get a lower-bound. This lower-bound is not tight when considering a sum of two or more scalar products. This is because the critical equality case occurs when $a = -b$. However considering at least 3 vectors, they cannot be respectively opposite to each other. Interestingly, Proposition 1 provides a way to get a tighter lower-bound.

By contradiction, assume there exists $x_\varepsilon \in \mathbb{R}^d \backslash \overline{\mathcal{X}}$ such that

$$\frac{\sum_{1 \leq i < j \leq N} \langle \nabla f_i(x_\varepsilon), \nabla f_j(x_\varepsilon) \rangle}{\sum_{i=1}^N \|\nabla f_i(x_\varepsilon)\|^2} = -\frac{1}{2} - \varepsilon \tag{256}$$

with $\varepsilon > 0$. Using Proposition 1 with batches of size 1, we have

$$\|\nabla f(x_\varepsilon)\|^2 = \frac{1}{N^2}\sum_{i=1}^N \|\nabla f_i(x_\varepsilon)\|^2 + \frac{2}{N^2}\sum_{1 \leq i < j \leq N} \langle \nabla f_i(x_\varepsilon), \nabla f_j(x_\varepsilon) \rangle \tag{257}$$

$$= \frac{1}{N^2}\sum_{i=1}^N \|\nabla f_i(x_\varepsilon)\|^2 - \frac{2}{N^2}\left(\frac{1}{2} + \varepsilon\right)\sum_{i=1}^N \|\nabla f_i(x_\varepsilon)\|^2 \tag{258}$$

$$= -\frac{2\varepsilon}{N^2}\sum_{i=1}^N \|\nabla f_i(x_\varepsilon)\|^2 < 0. \tag{259}$$

We thus arrive at a contradiction, as $\|\nabla f(x_\varepsilon)\|^2$ is non negative. As a consequence, (256) cannot hold. Thus, at worst we have for all $x \in \mathbb{R}^d \backslash \overline{\mathcal{X}}$

$$-\frac{1}{2} \leq \frac{\sum_{1 \leq i < j \leq N} \langle \nabla f_i(x), \nabla f_j(x) \rangle}{\sum_{i=1}^N \|\nabla f_i(x)\|^2}. \tag{260}$$

So, Proposition 8 is proved. □

## I   RACOGA VALUES IN LINEAR REGRESSION

In this section, we give deeper insights considering RACOGA values in the case of the linear regression problem. Moreover, we investigate, in this linear regression context, the link between RACOGA values and the curvature.

We have $\{a_i, b_i\}_{i=1}^N$, where each $(a_i, b_i) \in \mathbb{R}^d \times \mathbb{R}$, and we want to minimize $f$, with

$$f(x) := \frac{1}{N}\sum_{i=1}^N f_i(x) := \frac{1}{N}\sum_{i=1}^N \frac{1}{2}(\langle a_i, x \rangle - b_i)^2. \tag{LR}$$

As mentioned in Section 5.1, in this case the correlation between gradients is directly linked to the correlation between data by

$$\underbrace{\langle \nabla f_i(x), \nabla f_j(x) \rangle}_{\text{gradient correlation}} = (\langle a_i, x \rangle - b_i)(\langle a_j, x \rangle - b_j)\underbrace{\langle a_i, a_j \rangle}_{\text{data correlation}}. \tag{261}$$

In particular uncorrelated data, *i.e.* $\langle a_i, a_j \rangle = 0$, will induce uncorrelated gradients, *i.e.* $\forall x \in \mathbb{R}^d$, $\langle \nabla f_i(x), \nabla f_j(x) \rangle = 0$. In this case, RACOGA is verified for $c = 0$. In this section, we will see that outside this special case of orthogonal data, the characterization of RACOGA values is a challenging problem.

### I.1   TWO FUNCTIONS IN TWO DIMENSIONS

We study in this subsection the simplified case with $d = 2$, $N = 2$ and $b_1 = b_2 = 0$, *i.e.* two functions defined on $\mathbb{R}^2$ such that 0 is their unique minimizer. Formally the function we consider is the following

$$f(x) := \frac{1}{2}\left(\frac{1}{2}\langle a_1, x \rangle^2 + \frac{1}{2}\langle a_2, x \rangle^2\right). \tag{2f}$$

In this special case, we look at the gradient correlation, with same sign of the RACOGA values,

$$\Delta(x) := \langle \nabla f_1(x), \nabla f_2(x) \rangle = \langle a_1, x \rangle \langle a_2, x \rangle \langle a_1, a_2 \rangle. \tag{262}$$

If $\langle a_1, a_2 \rangle \neq 0$, $\Delta(x)$ is not identically equal to zero. Without loss of generality, for the following reasoning we can assume $\langle a_1, a_2 \rangle > 0$. Choose $x_0 \in \mathbb{R}^2$ such that $\langle a_1, x_0 \rangle > 0$ and $\langle a_2, x_0 \rangle \neq 0$. The function $x \to \langle a_1, x \rangle$ has a kernel $a_1^\perp$, the orthogonal of $a_1$. Moving along this kernel, *i.e.* considering $x = x_0 + k$, $k \in a_1^\perp$, if $a_1$ and $a_2$ are not colinear, one can make the scalar product $\langle a_2, x \rangle$ be positive or negative while $\langle a_1, x \rangle > 0$ as it remains equal to $\langle a_1, x_0 \rangle$. Therefore necessarily, if $\langle a_1, a_2 \rangle \neq 0$ and $a_1$ and $a_2$ are not colinear, we have $\min_x \Delta(x) < 0$. So, if $\langle a_1, a_2 \rangle \neq 0$, the minimum of the RACOGA values on the space is necessarily negative.

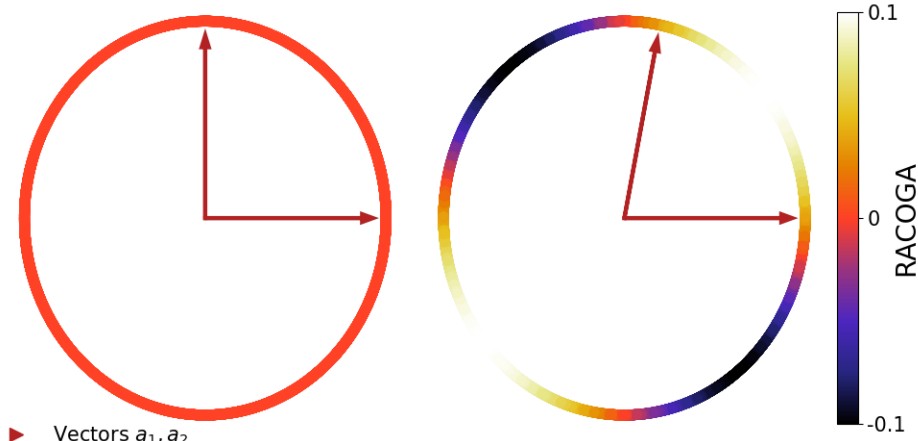

Vectors $a_1, a_2$

Figure 11: Illustration of the RACOGA values for problem (2f), along a circle around the solution. On the left part, $a_1$ and $a_2$ are orthogonal, inducing RACOGA is constant equal to zero. On the right part, $a_1$ and $a_2$ are slightly correlated, inducing positive and negative RACOGA values. Note that the non positive RACOGA areas exactly contain the points $x \in \mathbb{R}^d$ such that $\langle a_1, x \rangle \langle a_2, x \rangle \leq 0$.

Figure 11 illustrates this behaviour. We observe that non orthogonality of $a_1$ and $a_2$ creates non positive and non negative areas of RACOGA values.

According to Theorem 5, non positive RACOGA values indicate a bad performance of SNAG (Algorithm 2). The example of this section indicates that we can not hope to obtain theoretical results that would ensure high RACOGA values for any linear regression problem, and thus a good performance of SNAG. In the next section we see that, nevertheless, we can expect the RACOGA values to be positive over most of the space.

## I.2   RACOGA IS HIGH OVER MOST OF THE SPACE

In Section I.1, we considered a 2-dimensional example with 2-functions. Increasing dimension and adding functions, the problem of characterizing RACOGA values becomes harder. In the case of independent data, it is possible to give a theoretical result considering the RACOGA values in expectation over the data.

**Proposition 9.** *Let* $f(x) = \frac{1}{N} \sum_{i=1}^{N} (\Phi(x, a_i) - b_i)^2$ *where* $\{a_i, b_i\}_{i=1}^{N} \in \mathbb{R}^p \times \mathbb{R}$ *are i.i.d. and* $\Phi : \mathbb{R}^d \times \mathbb{R}^p \to \mathbb{R}$ *is differentiable. We have*

$$\mathbb{E}[\sum_{1 \leq i < j \leq N} \langle \nabla f_i(x), \nabla f_j(x) \rangle] = \frac{N(N-1)}{2} \| \mathbb{E}[(\Phi(x, a_1) - b_1) \nabla \Phi(x, a_1)] \|^2 \geq 0. \tag{263}$$

*In particular if* $\Phi(x, a_i) = \langle x, a_i \rangle$ *and* $a_1 \sim \mathcal{N}(m, \Gamma)$, $b_1 \sim \mathcal{N}(m_b, \sigma_b^2)$ *with* $a_1 \perp\!\!\!\perp b_1$, *we have*

$$\mathbb{E}\left[ \sum_{1 \leq i < j \leq N} \langle \nabla f_i(x), \nabla f_j(x) \rangle \right] = \frac{N(N-1)}{2} \| \Gamma x + m m^t x - m_b m \|^2 \geq 0. \tag{264}$$

*In both cases, the expectation is taken with respect to the data* $\{a_i, b_i\}_{i=1}^{N}$.

Proposition 9 indicates that when having a large amount of data, we can expect the RACOGA values to be positive over a large part of the space. The proof of Proposition 9 is deferred in Appendix I.4.

To confirm this statement empirically, for a fixed dataset $\{a_i, b_i\}_{i=1}^N$, we compute the RACOGA values on a sphere whose center is a minimizer of the function. Note that by the linearity of the gradient, the RACOGA values taken on this sphere are invariant by homothety. We set the bias, *i.e.* the $\{b_i\}_{i=1}^N$, at zero. This forces zero to be a minimizer, without loss of generality. The function we consider is the following

$$f(x) = \frac{1}{N} \sum_{i=1}^N \frac{1}{2} \langle a_i, x \rangle^2 . \tag{265}$$

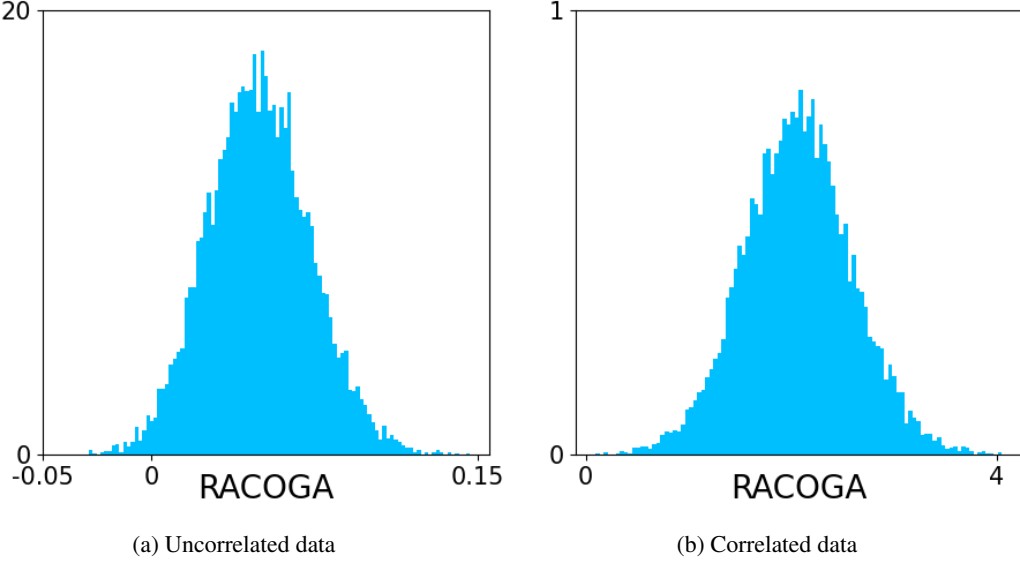

(a) Uncorrelated data          (b) Correlated data

Figure 12: Histogram distribution of the RACOGA values for points sampled uniformly on a sphere centered on a minimizer. On the left plot, the data are fewly correlated and the RACOGA values are mostly positive. On the right plot there is correlation inside data, and all RACOGA values are positive. Note that the RACOGA values are significantly higher on the right plot, because of the higher data correlation.

On Figure 12, we run this experiment in the case where $\{a_i\}_{i=1}^N$ are drawn uniformly onto the sphere, inducing low correlation inside data, and also in the case it is drawn following a gaussian mixture law, inducing higher correlation inside data. We set $d = 1000$, $N = 100$. It is the same problem as for Figure 1, except that here we set the bias to zero. We sample 10000 points on the sphere. In both cases, RACOGA is almost only non negative. More, all the RACOGA values are positive on Figure 12b, *i.e.* for the correlated dataset. Note that the observations we made in Section 5.1 are consistent: correlated data induce higher RACOGA values (Figure 12b), whereas with uncorrelated data the RACOGA values are smaller (Figure 12a).

However, one should not conclude from Figure 12b that RACOGA values are positive everywhere, as there could be non positive RACOGA value areas that are so small that our sampled points never fall in. Even more, we should not conclude from the fact that the eventual areas of non positive RACOGA values are small that the optimisation algorithms never cross them. We show in the following section that, actually, these small non positive RACOGA value areas exist and **attract** the optimization algorithms, and that stochasticity prevents the algorithm to get stuck inside.

### I.3 THE CURVATURE PROBLEM: FIRST ORDER ALGORITHMS ARE ATTRACTED BY LOW RACOGA VALUE AREAS

In Section I.2, we showed that in the case $d > N$ where $N$ is not too small, one can expect RACOGA values to be high over most of the space. This statement is reinforced in the presence of correlation inside data (Figure 12b).

However, if $d$ is high, even though we sample a large amount of points to evaluate RACOGA values, we could miss non positive RACOGA value areas if these areas are too small. On Figure 13, we see that these areas indeed exist. Moreover, although they are tiny with respect to the whole space (Section I.2), deterministic algorithms, namely GD (Algorithm 6) and NAG (Algorithm 7), dive into these areas and stayed trapped inside. Strikingly, SNAG behaves differently and it manages not to get stuck in the same area.

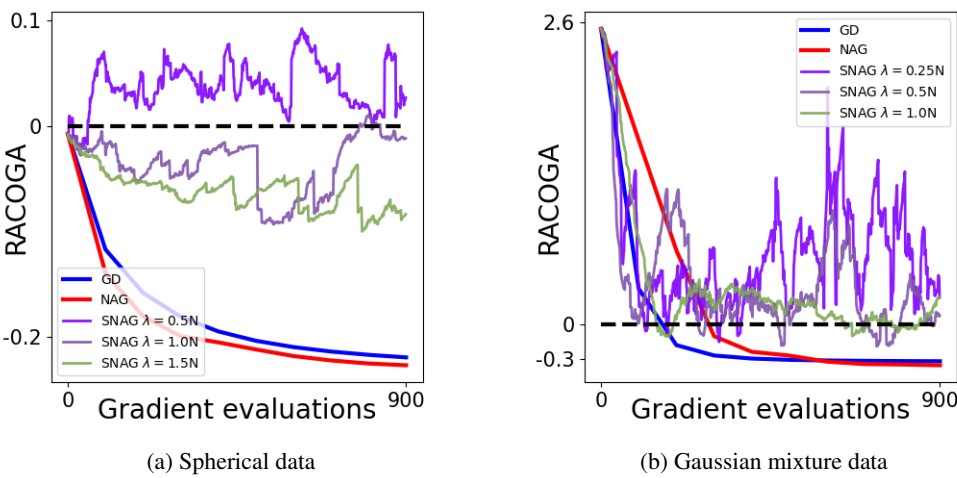

(a) Spherical data                    (b) Gaussian mixture data

Figure 13: Illustration of RACOGA values taken along iterations of GD (Algorithm 6), NAG (Algorithm 7) and SNAG (Algorithm 2, batch size 1) for the linear regression problem. On the left plot, the data are fewly correlated while on the right plot there is correlation inside data. Note that while deterministic algorithms, *i.e.* GD and NAG, dive and stay in a negative RACOGA value area, the stochasticity of SNAG enables it to not to be trapped in the same zone.

In the remaining of this section, we give an explanation of the behaviour observed on Figure 13.

**First order algorithms fall in low curvature area**    In the case of linear regression, which amounts to minimizing a quadratic function, it is well known that first order algorithms, namely algorithms that use only gradient information, converge faster in the direction of high curvature, *i.e.* directions such that the Hessian matrix has a high eigenvalue. We illustrate this phenomenon on Figure 14, where we plot the first iterations of GD (Algorithm 6) and NAG (Algorithm 7) applied to the function

$$g(x) = \frac{1}{2}x^T A x \tag{266}$$

where $A = \begin{pmatrix} L & 0 \\ 0 & \mu \end{pmatrix}$, $0 < \mu < L$. The algorithms GD and NAG dive and stay in a low curvature zone. However, note that the stochasticity of SNAG makes it unstable enough to not follow the exact same path.

**The RACOGA-Curvature link**    In this paragraph, we connect our observations about curvature and RACOGA values. Intuitively, for a point $x \in \mathbb{R}^d$ such that RACOGA is small, *i.e.* gradients are on average anti correlated, the gradients will compensate each other. Thus, we can expect that around this point, the gradient will have low values. Considering linear regression, this induce that non positive RACOGA areas tend to produce low curvature areas. We illustrate this phenomenon

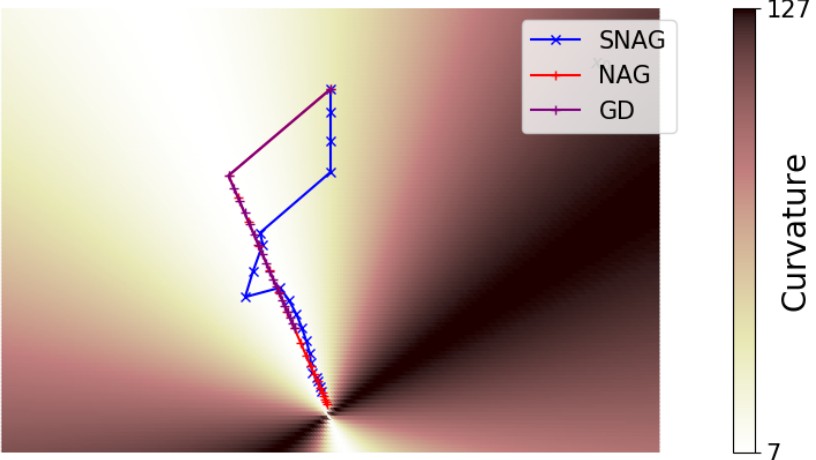

Figure 14: Illustration of the iterations of the trajectories of GD (Algorithm 6), NAG (Algorithm 7) and SNAG (Algorithm 2) applied the function (266). We also display the curvature of the function, which we define at $x \in \mathbb{R}^2_*$ as $\frac{x^T A x}{\|x\|^2}$. Note that the deterministic algorithms GD and NAG dive in the direction of smallest curvature, and then move following this direction. Note also that the instability of SNAG enables itself to follow less strictly this smallest curvature ravine.

on Figure 15, where we consider problem 2f. Actually if $a_1$ and $a_2$ have the same norm, the lowest RACOGA direction coincides exactly with the lowest curvature direction. As we mentioned in the previous paragraph that deterministic algorithms dive and stay in a low curvature zone, they actually dive and stay in low RACOGA areas. The instability provided by stochasticity allows SNAG not to get stuck inside these low RACOGA areas.

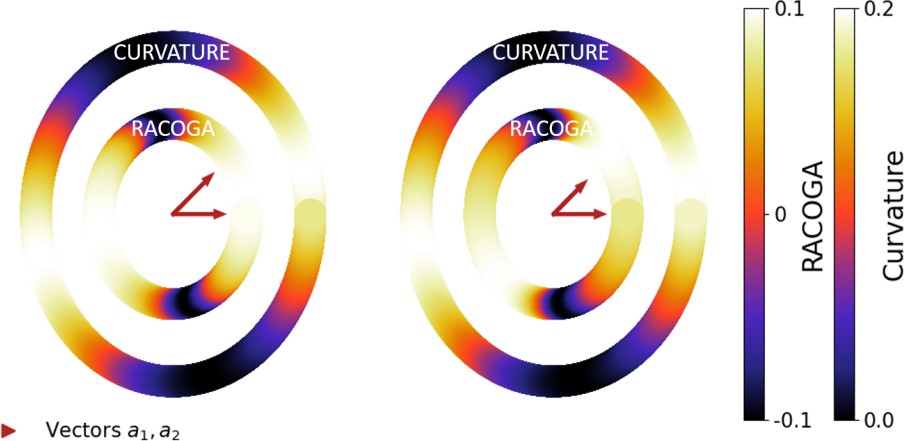

Figure 15: Comparison of the RACOGA and curvature values for problem (2f), along a circle around the solution. On the left plot, $a_1$ and $a_2$ have the same norm, which is not the case on the right plot. Note that the low curvature zone are close to the non positive RACOGA areas, and are exactly the same when $a_1$ and $a_2$ have same norm.

### I.4 PROOF OF PROPOSITION 9

**Proof of Equation (263)**  We consider the least square problem defined by

$$f(x) = \frac{1}{N} \sum_{i=1}^{N} (\Phi(x, a_i) - b_i)^2, \tag{267}$$

with $\Phi : \mathbb{R}^d \times \mathbb{R}^p \to \mathbb{R}$ differentiable and $\{a_i, b_i\}_{i=1}^{N}$, random variables drawn i.i.d.

By the independence of the variable, we have for $i \neq j$

$$\mathbb{E}[\langle \nabla f_i(x), \nabla f_j(x) \rangle] = \langle \mathbb{E}[\nabla f_i(x)], \mathbb{E}[\nabla f_j(x)] \rangle \tag{268}$$

$$= \|\mathbb{E}[\nabla f_1(x)]\|^2 \tag{269}$$

$$= \|\mathbb{E}[(\Phi(x, a_1) - b_1)\nabla \Phi(x, a_1)]\|^2 \geq 0 \tag{270}$$

Finally, we sum over $N$ to get Equation (263).

**Proof of Equation (264)**  First, we compute for $a = (a^{(1)}, \ldots, a^{(d)}) \in \mathbb{R}^d$ and $b \in \mathbb{R}$

$$\mathbb{E}[(\langle a, x \rangle - b)a] = (\mathbb{E}[\sum_i a^{(1)} a^{(i)} x_i] - \mathbb{E}[a^{(1)} b], \ldots, \mathbb{E}[\sum_i a^{(d)} a^{(i)} x_i] - \mathbb{E}[a^{(d)} b]). \tag{271}$$

We have $a \sim \mathcal{N}(m, \Gamma)$, $b \sim \mathcal{N}(m_b, \sigma_b^2)$ with $a \perp\!\!\!\perp b$. So we can deduce that $\mathbb{E}[(a^{(i)})^2] = \Gamma_{i,i} + m_i^2$, and $\mathbb{E}[a^{(i)} a^{(j)}] = \Gamma_{i,j} + m_i m_j$. Thereby, we have for a fixed $x \in \mathbb{R}^d$

$$\mathbb{E}[(\langle a, x \rangle - b)a] = (\mathbb{E}[\sum_i a^{(1)} a^{(i)} x_i] - \mathbb{E}[a^{(1)} b], \ldots, \mathbb{E}[\sum_i a^{(d)} a^{(i)} x_i] - \mathbb{E}[a^{(d)} b]) \tag{272}$$

$$= (\sum_i \mathbb{E}[a^{(1)} a^{(i)}] x_i - \mathbb{E}[a^{(1)}] \mathbb{E}[b], \ldots, \sum_i \mathbb{E}[a^{(d)} a^{(i)}] x_i - \mathbb{E}[a^{(d)}] \mathbb{E}[b]) \tag{273}$$

$$= (\sum_i \Gamma_{1,i} x_i + \sum_i m_1 m_i x_i - m_b m_1, \ldots, \sum_i \Gamma_{d,i} x_i + \sum_i m_d m_i x_i - m_b m_d) \tag{274}$$

$$= (\sum_i \Gamma_{1,i} x_i, \ldots, \sum_i \Gamma_{d,i} x_i) + (m_1, \ldots, m_d) \sum_i m_i x_i - m_b(m_1, \ldots, m_d) \tag{275}$$

$$= \Gamma x + (m^T x - m_b) m \tag{276}$$

From the previous computation and Equation (263), we deduce Equation (264).

