# OpenReview forum: "Gradient correlation is a key ingredient to accelerate SGD with momentum"
_ICLR.cc/2025/Conference — ICLR 2025 Poster_

### Official Review · Reviewer_wrcq · 2024-11-02

**Soundness:** 3
**Presentation:** 2
**Contribution:** 2
**Rating:** 6
**Confidence:** 4

**Summary:**

This paper studies the acceleration of Stochastic Nesterov's Accelerated Gradient (SNAG) method over the SGD algorithm. In particular, the study is based on Vaswani et al, 2019, in which the acceleration is first proved based on the Strong Growth Condition (SGC) of the stocastic gradient. This paper extends the previous paper by showing an accelerated almost sure convergence result for SNAG, and develops condition that lead to a better SGC coefficient. Based on this condition, they show how the SGC coefficient changes as the batch size increases. The paper also verifies the condition using experiments.

**Strengths:**

1. The paper provides an extension of the original convergence theorem in Vaswani et al, 2019 in the almost sure convergence form, leading to a more comprehensive understanding of the SNAG algorithm.

2. The paper's result covers both convex and strongly convex case. In particular, both the PosCorr and the RACOGA condition can lead to the SGC without assuming the strong convexity.

3. Centered around the SGC, the paper develops conditions that implies the SGC, which allows the paper to investigate the relationship between the batch size and the SGC coefficient.

**Weaknesses:**

1. Although the almost sure convergence result provided in Theorem 4 deepens our understanding of the SNAG method, I believe that the major focus of this paper is still on how the gradient correlation can lead to a better SGC coefficient, which gives acceleration for SNAG. From this perspective, the result in Theorem 4 seems a bit disjoint from the other sections of the paper.

2. Although the RACOGA condition holds in general with a coefficient of $c \geq -\frac{1}{2}$, it does not seem to be easy to find a tight $c$ for the objectives, as evaluating this lower bound involves analyzing the pairwise inner product between gradients for all choices of the parameters in the parameter space. Furthermore, when $c$ approaches to $-\frac{1}{2}$, the SGC coefficient $\rho = \frac{N}{1 + 2c}$ approaches infinity, leading to a trivial condition.

3. The experimental verification of the paper seems quite weird. It is noticed that, in the linear regression case, the gradient correlation involves both the inner product term and the $\mathbf{a}_i^\top \mathbf{a}_j$, sign of the residual terms $\mathbf{x} ^\top\mathbf{a}_i - b_i$'s. In particular, different signs of the residual terms could lead to completely different lower bound on the gradient correlation. However, it seems that in the experimental design the paper considered only the correlation between the data. Moreover, it may contradict the claim of the paper that RACOGA helps acceleration since in Figure 1.(a) the green curve, with a smaller RACOGA coefficient, led to a faster convergence than the blue curve.

**Questions:**

1. How is Theorem 2 different from the results in Vaswani et al, 2019? It would be nice if the paper could include a detailed comparison of the two results.

2. In Example 3, the paper demonstrates the benefits of PosCorr condition over the traditional way of verifying the SGC condition by showing that $\rho_K\leq \frac{N}{K}$. Why is $\frac{N}{K} \leq \frac{L_{(K)}}{\mu}$, so that it could be considered an improvement?

3. What is $\lambda$ in Figure 1?

---

> ### Author Response · Authors · 2024-11-19
>
> Thank you for your feedback and careful reading of our paper. Please see below our detailed answers
> + Although the almost sure convergence result provided in Theorem 4 deepens our understanding of the SNAG
> method, I believe that the major focus of this paper is still on how the gradient correlation can lead to a
> better SGC coefficient, which gives acceleration for SNAG. From this perspective, the result in Theorem 4
> seems a bit disjoint from the other sections of the paper.
>
> We thank the reviewer to share his interrogation about the relevance of Theorem 4. As the reviewer noticed,
> our paper main interest is the link between gradient correlation and the possibility of accelerating SGD
> with SNAG. In order to have a complete comparison between these two algorithms, we include almost sure
> convergence rates (Theorem 2-4) additionally to expectation rates (Theorem 1-3). Importantly, in the strongly
> convex case notice that, as explained in Remark 5, the SGC, linked with gradient correlation, appears in the
> rate of Theorem 4 so in this case the question of gradient correlation remains crucial.
>
>
> + Although the RACOGA condition holds in general with a coefficient of $c \geq -\frac{1}{2}$, it does not seem to be easy to find a tight $c$ for the objectives, as evaluating this lower bound involves analyzing the pairwise inner product between gradients for all choices of the parameters in the parameter space. Furthermore, when approaches to $-\frac{1}{2}$, the SGC coefficient $\rho = \frac{N}{1+2c}$ approaches infinity, leading to a trivial condition.
>
> We thank the reviewer for this relevant remark. As discussed in our Appendix H, it is not easy to find a tight RACOGA constant even in the relative simple case of linear regression, and it could be an interesting venue of research. Also, as the reviewer noticed, the case of RACOGA constant approaching $-\frac{1}{2}$ is indeed a critical case, that is exactly the case where SNAG will perform poorly. We propose a modification of Remark 6 in our revised version in order to emphasize this property.
>
> + The experimental verification of the paper seems quite weird. It is noticed that, in the linear regression case, the gradient correlation involves both the inner product term and the $a_i^T a_j$, sign of the residual terms $x^T a_i - b_i$'s. In particular, different signs of the residual terms could lead to completely different lower bound on the gradient correlation. However, it seems that in the experimental design the paper considered only the correlation between the data. Moreover, it may contradict the claim of the paper that RACOGA helps acceleration since in Figure 1.(a) the green curve, with a smaller RACOGA coefficient, led to a faster convergence than the blue curve.
>
>
>  We thank the reviewer for this interesting remark. As the reviewer noticed, the sign of $a_i^T a_j$ is not necessarily the same than the sign of RACOGA. We aim to stress that in the particular case of linear regression, Equation 15 gives a strong link between the correlation and the correlation between gradients (RACOGA). For instance, if the data are uncorrelated, then the gradients are also uncorrelated. This is exactly the example detailed in Example 3.
>
>  Moreover, experimentally, on Figure 1-2, we show RACOGA values and not the data correlation. Then, on Figure 1, we see that the values of RACOGA are very different if the data are weakly correlated (Figure 1.a) or highly correlated (Figure 1.b). We proposed a revised version of Figure 1 with the same scale for RACOGA on both plots to stress this behaviour.
>
> Finally, in Appendix H, we develop more deeply the link between RACOGA and the data correlation in some simple examples to get more intuition about this link.
>
> + How is Theorem 2 different from the results in Vaswani et al, 2019 ? It would be nice if the paper could include
> a detailed comparison of the two results.
>
> We thank the reviewer for his suggestion. We assumed that by "Theorem 2", the reviewer meant "Theorem 3", which is the SNAG result in expectation. The convergence result from [1] and Theorem 3 differ as we consider a version of the Algorithm that involves a fewer amount of parameters. For completeness, we added a Section C.3 in our revised version where we compare SNAG with the algorithm of [1].
>
> Also, we want to emphasize that our main theoretical contributions are rather our almost sure convergence results (Theorem 4) and our gradient correlation analysis (Proposition 1-2, Theorem 5).

---

> > ### Author Response · Authors · 2024-11-19
> >
> > + In Example 3, the paper demonstrates the benefits of PosCorr condition over the traditional way of verifying the SGC condition by showing that $\rho_K \leq \frac{N}{K}$. Why is $\frac{N}{K}\leq \frac{L_{(K)}}{\mu}$, so that it could be considered an improvement ?
> >
> >
> >  We thank the reviewer for this crucial question. In fact, as the reviewer noticed, the core goal of our paper is to see when the SGC is verified with smaller values of $\rho_K$. The smaller the SGC constant $\rho_K$ is, the more informative this condition is.
> >
> > The bound $\rho_k \le \frac{L_{(K)}}{\mu}$ relies on geometrical properties of $f$ that are difficult to control in practice. The bound $\rho_k \le \frac{N}{K}$ only relies on the number of gradients $N$ and the batch size $K$. These quantities are known in practice. The inequality $\frac{N}{K}\leq \frac{L_{(K)}}{\mu}$ is not true in general. However, in Example 2, this inequality has been shown ($N$ = 2 in this example).
> >
> > Finally, the bound $\rho_k \le \frac{N}{K}$ is more practical, since it  does not rely on geometrical properties of $f$ and it seems to be finer in practice. This is the reason why this bound can be considered as an improvement.
> >
> > + What is $\lambda$ in Figure 1?
> >
> > We thank the reviewer to point out that the presence of $\lambda$ on Figure 1 was not detailed in the previous main text. As mentioned in our Appendix A.1, $\lambda$ is a parameter that replaces the unknown strong growth condition. Grossly, small $\lambda$ values lead to more aggressive stepsizes, as $s = \frac{1}{L \lambda}$. High RACOGA values mean high correlation, and in this case we can take smaller $\lambda$ and converge faster (figure 1.b). Note that it is not the case with low RACOGA values (figure 1.a). To make it clearer in our paper, we added a paragraph about the role of $\lambda$ in Section 5.1, in the main text.
> >
> > [1] Vaswani et al. Fast and faster convergence of sgd for over-parameterized models and an accelerated perceptron, 2019

---

> > > ### Comment · Reviewer_wrcq · 2024-11-26
> > >
> > > Thank you so much for your detailed reply. I think the response together with the changes made to the paper has resolved most of my concerns. Therefore, I have raised my score to 6.
> > >
> > > I believe that what is eventually preventing me from giving a higher score to this paper is that the RACOGA condition seems to have limited applicability (as pointed in my Weakness #2). For instance, for neural network training it would be nearly impossible to verify the condition throughout the training process. Although the condition seems close to a necessary condition, I believe that a regulatory condition like this would have more value if it could be applied to more realistics scenarios. Nevertheless, the work itself presents a good contribution to the field, so I would still recommend acceptance.

---

> ### Comment · Area_Chair_V8cJ · 2024-11-24
>
> Dear Reviewer wrcq,
>
> The author discussion phase will be ending soon. The authors have provided detailed responses. Could you please reply to the authors whether they have addressed your concern and whether you will keep or modify your assessment of this submission?
>
> Thanks.
>
> Area Chair

---

### Official Review · Reviewer_Dovy · 2024-11-03

**Soundness:** 3
**Presentation:** 4
**Contribution:** 3
**Rating:** 6
**Confidence:** 3

**Summary:**

The authors study the convergence of SGD with momentum--Stochastic Nesterov Accelerated Gradient (SNAG) and obtained an improved rate compared to vanilla SGD. More precisely, they consider the strong growth condition that, intuitively, quantifies the amount of noise. Using this definition, they are able to achieve e.g. $o(\frac{1}{n^2})$ convergence rate (SGD has $o(\frac{1}{n})$) for convex functions. For certain objective functions, the authors propose a way to compute the strong growth condition and a new condition RACOGA. In addition, the authors have numerical experiments to verify their results.

**Strengths:**

1. The paper is well-organized and easy to read. The material in the supplement serves as a good complement to the main paper. Experimental results are presented in a clear way with nice plots and great details.

2. The main result is indeed very interesting to the community and gives some insight into a long-standing question. The theoretical contribution mainly comes from Theorem 4 which provides an almost surely convergence for SNAG showing a speed-up compared to SGD.

3. By proposing a new characterization of SGC, the authors improved the assumption so that it only depends on the size of the dataset and batch size. Using this, the authors proposed a new condition--RACOGA.

4. The authors also discuss the relation between batch size and gradient correlations which brings interesting insights into when to use stochastic and non-stochastic versions of these algorithms.

**Weaknesses:**

1. Proof for theorem 4 heavily relies on an existing result (Sebbouh et al. 2021, theorem 9), which one could argue it weakens the theoretical contributions of this work.

2. I appreciate that the authors made an effort to compare RACOGA with gradient diversity and gradient confusion and agree with the authors that they are not identical, but they do look quite similar.

**Questions:**

1. It would be nice to have a table that summarizes results from theorem 1-4 (perhaps including results from the literature) so that readers don't have to go back and forth to compare them.

2. Authors have remark 5 to explain the results from theorem 4 which does help me to understand it. I wonder if there is any intuition about why $\rho_k$ plays a different role in convex vs strongly convex cases. Also, for the strongly convex case, it seems we need less noisy data for SNAG to beat SGD, because we want $\rho_k$ to be small. Am I understanding this correctly? For continuous strongly convex function, there is a unique minimizer, meaning it won't stuck in some local minimizers. How does this fit into this theory?

---

> ### Author Response · Authors · 2024-11-19
>
> Thank you for your positive feedback and careful reading of our paper. Please see below our detailed answers
>
> + Proof for theorem 4 heavily relies on an existing result (Sebbouh et al. 2021, theorem 9), which one could
> argue it weakens the theoretical contributions of this work.
>
> We are not sure what the reviewer mean with this remark, as there is no Theorem 9 in [1]. If referring to the
> theorem 9 present in our paper, it is a result from [2], that is a classical tool to study almost sure convergence
> in stochastic optimization.
>
> What we meant is that we use this result to derive the almost sure rates. And to be transparent we mentioned
> that before us, Sebbouh et al. 2021 almost sure results also heavily rely on these results (from [2]), although
> they are applied to different algorithms. Finally, note that the strongly convex result of our Theorem 4 does
> not relie on the result of Robbins and Siegmund (1971), although it needs less developed tools.
>
> + I appreciate that the authors made an effort to compare RACOGA with gradient diversity and gradient
> confusion and agree with the authors that they are not identical, but they do look quite similar.
>
> We are glad the reviewer appreciate our attempt to relate our work with the literature. It is absolutely true
> that RACOGA and gradient diversity [3] are strongly related (Equation 39), although they correspond to
> different viewpoints. RACOGA is a direct measure of the gradient correlation. The gradient confusion [4]
> assumption is a less closed assumption, as it only measures anti correlation.
>
> + It would be nice to have a table that summarizes results from theorem 1-4 (perhaps including results from
> the literature) so that readers don’t have to go back and forth to compare them.
>
> We thank the reviewer for this suggestion. In order to make the comparison between SGD and SNAG under
> different conditions more clear we add a table in Section 3 (Table 1).
>
> + Authors have remark 5 to explain the results from theorem 4 which does help me to understand it. I wonder
> if there is any intuition about why ρK plays a different role in convex vs strongly convex cases. Also, for the
> strongly convex case, it seems we need less noisy data for SNAG to beat SGD , because we want to be small.
> Am I understanding this correctly ? For continuous strongly convex function, there is a unique minimizer,
> meaning it won’t stuck in some local minimizers. How does this fit into this theory ?
>
> This is a very interesting question, that involves deep optimization concepts. The different role of $\rho_K$ regarding the convexity or strongly convex functions stems from the different nature of momentum in these two cases.
>
> We believe a good intuition can be gained from the continuous variants of SNAG [5,6]. In the strongly convex case, best convergences are achieved choosing momentum to be constant. In our case, the momentum depends on the SGC constant $\rho_K$ making the role of $\rho_K$ crucial. However, in the convex case, as this class of functions include very flat functions (e.g $x \to x^{12}$), best convergences are achieved with increasing momentum. In this case, the SGC constant factor is asymptotically negligible.
>     As you noticed, we want $\rho_K$ in the SGC to be small in both cases.
>     In the convex case, the finite-time speed up depends on SGC constant $\rho_K$.
>     However, the asymptotic speed-up does not depend on $\rho_K$. In the case of strongly convex $f$, both finite-time and asymptotic speed-up depend on $\rho_K$.
>
> Convex functions do not have local minimizers (as mentioned just after Definition 2).
> In our study, note we that do not assume that the $f_i$ are convex, we only assume that $\frac{1}{N}\sum_{i=1}^N f_i$ is convex/strongly-convex. Therefore some $f_i$ could have many minimizers and not be convex.
>
> [1] Sebbouh and Gower and  Defazio, Almost sure convergence rates for Stochastic Gradient Descent and Stochastic Heavy Ball, 2021.
>
> [2] Robbins and Siegmund A convergence theorem for non negative almost supermartingales and some applications, Optimizing methods in statistics, pages 233-257, 1971.
>
> [3] Yin et al.  Gradient diversity:  a key ingredient for scalable distributed learning, 2018.
>
> [4]  Sankararaman et al.  The impact of neural network overparameterization on gradient confusion and stochastic gradient descent. 2020
>
> [5] Su, Boyd, Candès A Differential Equation for Modeling Nesterov's Accelerated Gradient Method: Theory and Insights 2016
>
> [6] Siegel Accelerated First-Order Methods: Differential Equations and Lyapunov Functions 2019

---

> > ### Comment · Reviewer_Dovy · 2024-11-22
> > **reply to authors**
> >
> > Thank you for the clarifications. I am keeping my positive score and recommend accepting this paper.

---

### Official Review · Reviewer_zYBC · 2024-11-03

**Soundness:** 3
**Presentation:** 2
**Contribution:** 3
**Rating:** 8
**Confidence:** 4

**Summary:**

The paper studies stochastic versions of Nesterov's accelerated gradient descent (NAG). These algorithms have been previously shown to converge at the same accelerated rates as NAG when the stochastic gradient estimates satisfy the so-called strong growth condition (SGC). Specifically for functions satisfying a finite sum structure, this paper finds a sufficient condition (RACOGA) in terms of gradient correlation that implies the strong growth condition, consequently implying that SNAG converges at an accelerated rate in those settings. Numerical experiments are provided to verify the implications of the RACOGA condition on accelerated convergence of stochastic algorithms.

**Strengths:**

Previous works have shown that stochastic versions of NAG converge at the same accelerated rates when the gradient estimates satisfy the strong growth condition (SGC). While they provide heuristics that suggest that SGC is a reasonable assumption in the context of overparametrized deep learning, it is not always clear when the condition is actually satisfied. This work addresses that gap in the literature. The authors show that for functions of the form $f=\sum_{i=1}^N f_i$, positive gradient correlation (i.e. $\langle f_i, f_j\rangle \geq 0$ for all $i,j$) is sufficient to guarantee the strong growth condition for the gradients. This result also gives a bound for the strong growth parameter ($\rho$) in terms of the batch size, which is important for choosing optimal parameters for the SNAG algorithms. The main contribution of the paper is a gradient correlation condition (RACOGA) which implies SGC for functions with a finite sum structure. This further implies that SNAG converges at an accelerated rate in those settings. I think this is a useful contribution and a step in the direction of better understanding why momentum-based stochastic gradient algorithms perform well in practice. The authors provide numerical experiments to back their claims, which I found interesting and insightful as well.

**Weaknesses:**

1. Line 70: "However, even the question of the possibility to accelerate with SNAG in the convex setting is not solved yet."
This is either unclear or inaccurate or both. There are several works which address the convergence of accelerated methods in the stochastic setting, both under SGC and with classical Robbins-Monro bounds, at least for smooth objectives. For a rigorous statement, the authors should specify the geometric assumptions, smoothness assumptions, and assumptions on the gradient oracle. Since the authors are aware of previous works on acceleration in convex optimization under SGC noise, it is unclear what meaning is intended.

2. More concerningly, the next sentence says "Finally, note that our core results (Propositions 1-2) do not assume convexity, and thus they could be used in nonconvex settings." Juxtaposed with the previous sentence, it gives the reader the impression that the authors have addressed the question of acceleration for non-convex functions, which is not true. It is true that the conditions PosCorr and RACOGA studied in Propositions 1 and 2 imply SGC even for non-convex functions. But SGC/PosCorr/RACOGA alone is not sufficient for any of the accelerated convergence results provided here or in previous works, some form of convexity is still required. The current phrasing is misleading since, again, it conflates conditions on the noise in the gradient estimates and on the geometry of the objective function, which are in general independent. If there is a relation in the setting the authors consider, they need to explain and emphasize this. I do not see the implication.

3. The authors claim one of their main contributions is "new almost sure convergence results (Theorem 4)". However, almost sure convergence is already covered by corollary 5 in Gupta et al. "Achieving acceleration despite very noisy gradients" arXiv:2302.05515. That paper studies a stochastic version of NAG under a condition similar to SGC. The authors should highlight the differences in their results.

4. The theorem 4 statement suggests that the authors recover a rate almost surely, but in the current presentation, it is unclear what precisely is meant. Even for $O(n^{-2})$: Is there a random variable C such that $f(x_n) - f(x^*) \leq C/n^2$ simultaneously for all $n$ (and almost surely in probability), or does the random constant $C$ depend on $n$? And, what is meant by $o(n^{-2})$? For a machine learning venue, they should state a non-asymptotic quantitative bound. Almost sure convergence is a notion of convergence which is *not* induced by a metric on a space of random variables. As such, there is no immediate way of making sense of the notion that $f(x_n)$ and $f(x^*)$ are $o(n^{-2})$-close in a specific sense. More explanation is needed. The same concern applies to Theorem 2.

5. The title of the paper is "Gradient correlation is **needed** to accelerate SGD with momentum", which makes it sound like gradient correlation is a necessary condition (i.e. if it is not satisfied then SGD with momentum does not converge at an accelerated rate). But I did not see a result proving that in the paper. The results actually claim that it is a sufficient condition. The title does not accurately reflect the main results.

6. $L_{(K)}$ acts as an "effective" Lipschitz-continuity parameter of the gradient, depending on the batch size $K$. The results for SGD (Theorems 1 and 2) are provided in terms of $L_{(K)}$ without assuming SGC but the results for SNAG (Theorems 3 and 4) are provided in terms of the SGC parameter $\rho_K$. Then these two results, derived under different conditions, are compared to conclude that SNAG does not accelerate over SGD unless $\rho_k<\frac{L_{(K)}}{\sqrt{L}}\cdot C$ (where $C$ is a constant that differs in the convex and strongly convex cases). This seems like an unfair and misleading comparison to me. Both, $L_{(K)}$ and $\rho_K$, measure the stochasticity of the gradient estimates but in different ways. The authors demonstrate in Appendix E.2 an example where $L_{(K)}$ is a tighter estimate of the effective Lipschitz constant than $L\rho_k$. That does show that if the smoothness parameter $L_i$ of each summand $f_i$ is known, then using $L_{(K)}$ would allow us to choose a larger step-size for SGD than the one provided by $1/L\rho_k$. However, a fair comparison between SGD and SNAG can only be made if the same assumptions and information are used to calculate the step size, but there are no convergence results available for SNAG that directly make use of $L_{(K)}$. This feels like comparing apples and oranges. If the authors want to argue that you can use a larger step size for SGD than for SNAG, they should justify why Nesterov would blow up with that step size.

**Questions:**

1. Was the Algorithm 2, in its given form, first introduced by Nesterov (2012) "Efficiency of coordinate descent methods on huge-scale optimization problems."? If yes, the authors should cite that paper. I appreciate Proposition 4 in the appendix showing that the more common two parameter NAG algorithm (Algorithm 8, with $\tau=0$) can be obtained as a special case of this algorithm with a reparametrization.

2. Proposition 2 suggests that RACOGA holding with $c>-0.5$ is sufficient to verify the SGC. But in Figure 1(a), SNAG does not accelerate over SGD despite the RACOGA values being greater than -0.12. Is there an explanation for this apparent discrepancy?

3. Just to confirm, in the experiments, were GD and NAG used with the full batch gradient at each step (e.g. were all of 50k images used for the CIFAR-10 experiment at each training step)? If yes, this might be worth specifying explicitly since most of the times in machine learning experiments, NAG refers to Algorithm 8, even when it is used with mini-batch gradients.

---

> ### Author Response · Authors · 2024-11-19
>
> Thank you for your positive feedback and careful reading of our paper. Please see below our detailed answers.
> + Line 70: "However, even the question of the possibility to accelerate with SNAG in the convex setting is not solved yet." This is either unclear or inaccurate or both. There are several works which address the convergence of accelerated methods in the stochastic setting, both under SGC and with classical Robbins-Monro bounds, at least for smooth objectives. For a rigorous statement, the authors should specify the geometric assumptions, smoothness assumptions, and assumptions on the gradient oracle. Since the authors are aware of previous works on acceleration in convex optimization under SGC noise, it is unclear what meaning is intended.
>
> We thank the reviewer to stress that this sentence might be confusing. In this paragraph of the introduction, we aim to stress the interest of studying the convex setting. The detailed explanation about existing works has been stated in the previous paragraphs of the introduction "Stochastic Nesterov Accelerated Gradient(SNAG)" and "What keeps us hopeful".
>
> We propose to reformulate this sentence as : "However, even in the convex setting,here is still work to do concerning the possibility of accelerating SGD with SNAG. For example, up to our knowledge, characterizing convex smooth functions that satisfy SGC has not been addressed yet.".
>
> + More concerningly, the next sentence says "Finally, note that our core results (Propositions 1-2) do not assume
> convexity, and thus they could be used in nonconvex settings." Juxtaposed with the previous sentence, it
> gives the reader the impression that the authors have addressed the question of acceleration for non-convex
> functions, which is not true. It is true that the conditions PosCorr and RACOGA studied in Propositions
> 1 and 2 imply SGC even for non-convex functions. But SGC/PosCorr/RACOGA alone is not sufficient for
> any of the accelerated convergence results provided here or in previous works, some form of convexity is still
> required. The current phrasing is misleading since, again, it conflates conditions on the noise in the gradient
> estimates and on the geometry of the objective function, which are in general independent. If there is a relation
> in the setting the authors consider, they need to explain and emphasize this. I do not see the implication
>
> We thank the reviewer to point out that this formulation might be misinterpreted. Our work does not solve the
> question of acceleration for non convex functions and we do not claim in our work to do so. We meant to say
> that further works, e.g. considering SGC in non convex setting, could also use RACOGA. Indeed, the essence
> of this definition and the properties we derive in Propositions 1 and 2 do not stem from convexity. We propose
> to reformulate this sentence as : "Finally, note that our core results about gradient correlation (Propositions
> 1-2) do not assume convexity, and thus could be used in future works beyond the convex setting.".
>
> + The authors claim one of their main contributions is "new almost sure convergence results (Theorem 4)".
> However, almost sure convergence is already covered by corollary 5 in Gupta et al. "Achieving acceleration
> despite very noisy gradients" arXiv :2302.05515. That paper studies a stochastic version of NAG under a
> condition similar to SGC. The authors should highlight the differences in their results.
>
> We indeed missed the recent result of [1], where the authors indeed give an almost sure convergence result. However note that if they show that $f(x_n) \to f(x^\ast)$ almost surely, they do not provide any convergence rates, as we do in our Theorem 4. We cite this work in our revision and add the sentence "Almost sure convergence has already been addressed in [1] without convergence rates." in the beginning of section 3.3.

---

> ### Author Response · Authors · 2024-11-19
>
> + The theorem 4 statement suggests that the authors recover a rate almost surely, but in the current presentation, it is unclear what precisely is meant. Even for $O(n^{-2})$: Is there a random variable C such that $f(x_n)-f^\ast \leq C/n^2$ simultaneously for all $n$ (and almost surely in probability), or does the random constant depend on $n$ ? And, what is meant by $o(n^{-2})$? For a machine learning venue, they should state a non-asymptotic quantitative bound. Almost sure convergence is a notion of convergence which is not induced by a metric on a space of random variables. As such, there is no immediate way of making sense of the notion that $f(x_n)$ and f$(x^\ast)$ are $o(n^{-2})$-close in a specific sense. More explanation is needed. The same concern applies to Theorem 2.
>
> We thank the reviewer to point out to  us that the notion of almost sure convergence has not been formally defined in our work. Almost sure convergence rate are standard in stochastic gradient algorithm analysis (see Theorem 8 in [2], Theorem 2-3 in [3], Corollary 5 in [4], Theorem 1-2-3 in [5] or  Convex Convergence Theorem, page 156 in [6]), so we do not recall the definition of this notion. More formally, if $\Omega$ is the set of realization of the noise, we say that $f(x_n) - f^\ast \overset{a.s.}{=} o\left( \frac{1}{n^2} \right)$ if and only if $\exists A \subset \Omega$, such that $\mathbb{P}(A) = 1$ and $\forall \omega \in A$, $\forall \epsilon > 0$, $\exists n_0 \in \mathbb{N}$, such that $\forall n \ge n_0$, $|f(x_n(\omega)) - f^\ast| \le \frac{\epsilon}{n^2}$. In order to make it clearer in the revised version of paper, we recall this definition in the beginning of section 3.
>
> For practical usage, non-asymptotic quantitative bounds are in fact more useful. This is the reason why Theorem 2 and 4 are together with Theorem 1 and 3 that provide finite time quantitative bounds in expectation. However, we are not aware of finite time convergence rates, asymptotic being the standard results in almost sure convergence studies, as it can be seen in above references. As the reviewer mentioned, even the task to define such a convergence is tough.
>
> Finally, we are not sure to understand this part of the reviewer remark : "Almost sure convergence is a notion of convergence which is not induced by a metric on a space of random variables.". Do our clarifications about the almost surely rate definition solve this concern ?
>
> + The title of the paper is "Gradient correlation is needed to accelerate SGD with momentum", which makes
> it sound like gradient correlation is a necessary condition (i.e. if it is not satisfied then SGD with momentum
> does not converge at an accelerated rate). But I did not see a result proving that in the paper. The results
> actually claim that it is a sufficient condition. The title does not accurately reflect the main results.
>
> We thank the reviewer to point out to us that our title might be confusing.
> In fact, our theoretical results show that positive gradient correlation, i.e. RACOGA verified for c > 0, ensures
> accelerated convergence rate for SNAG. Note that we look empirically at the reverse implication. On Figure
> 1)a), we highlighted that the uncorrelated data, i.e. RACOGA not verified for c > 0, leads to SGD being
> faster than SNAG.
>
> In order to suppress any logical ambiguities, we have decided to reformulate our title into : "Gradient correlation is a key ingredient to accelerate SGD with momentum"

---

> > ### Author Response · Authors · 2024-11-19
> >
> > + $L_{(K)}$ acts as an "effective" Lipschitz-continuity parameter of the gradient, depending on the batch size $K$. The results for SGD (Theorems 1 and 2) are provided in terms of $L_{(K)}$ without assuming SGC but the results for SNAG (Theorems 3 and 4) are provided in terms of the SGC parameter $\rho_K$. Then these two results, derived under different conditions, are compared to conclude that SNAG does not accelerate over SGD unless $\rho_k < \frac{L_{(K)}}{\sqrt{L}}$. $C$ (where $C$ is a constant that differs in the convex and strongly convex cases). This seems like an unfair and misleading comparison to me.
> >     Both, $L_{(K)}$ and $\rho_K$, measure the stochasticity of the gradient estimates but in different ways. The authors demonstrate in Appendix E.2 an example where $L_{(K)}$ is a tighter estimate of the effective Lipschitz constant than $L\rho_K$. That does show that if the smoothness parameter $L_i$ of each summand $f_i$ is known, then using $L_{(K)}$ would allow us to choose a larger step-size for SGD than the one provided by $\frac{1}{L\rho_K}$. However, a fair comparison between SGD and SNAG can only be made if the same assumptions and information are used to calculate the step size, but there are no convergence results available for SNAG that directly make use of $L_{(K)}$. This feels like comparing apples and oranges. If the authors want to argue that you can use a larger step size for SGD than for SNAG, they should justify why Nesterov would blow up with that step size.
> >
> > We thank the reviewer for this very relevant Remark.
> > In fact, between the submission and the review period, we have realized that there was a gap in our work on this question. We have added Appendix F to fill this gap.
> >
> > The question is to fairly compare the convergence rate of SGC and SNAG.
> >
> > First, in Appendix E.2, we justify that a convergence result for SGD under the SGC is not relevant. The reason is, although it allows to compare the two algorithms under the same assumptions, the convergence bound for SGD (Theorem 8) is always worse than the one for SNAG, which is misleading.
> >
> > Reversely, as the reviewer notice in his Remark, in the previous version of the paper, we did not justify why we did not consider a convergence result for SNAG under the same assumptions as we did for SGD in Theorem 1, $\textit{i.e.}$ using $L_{(K)}$. We answer this question with Theorem 9 in Appendix F. Note that the bounds of Theorem 9 are achieved by tuning the parameters such that we obtain the fastest decrease without SGC, which is of order $O(n^{-1})$. However, when assuming SGC, we can achieve with SNAG a convergence of $O(n^{-2})$ in the convex case: this indicates that SGC is the characterization of the noise that allows to achieve such a result, see our discussion (Remark 9 in particular) after Theorem 9 in Appendix F.
> >
> > In conclusion, in order to compare the convergence speed of SGD and SNAG, it is more relevant to use Theorem 1 and Theorem 3, although Theorem 3 makes use of SGC and Theorem 1 does not. Otherwise, comparing Theorem 1 and Theorem 9 indicates that SGD is (almost) always better than SNAG, and comparing Theorem 3 with Theorem 8 indicates that SNAG is always better than SGD, which in both cases is misleading (see Figure 1 for an experimental counterexample).
> >
> > To make these considerations clearer in the revised version of the paper, we modify Remark 4 to explain this briefly in the main text and we provide a detailed discussion in Appendix (Appendix F in particular).
> >
> >
> > +  Was the Algorithm 2, in its given form, first introduced by Nesterov (2012) "Efficiency of coordinate descent methods on huge-scale optimization problems."? If yes, the authors should cite that paper. I appreciate Proposition 4 in the appendix showing that the more common two parameter NAG algorithm (Algorithm 8, with $\tau = 0$) can be obtained as a special case of this algorithm with a reparametrization.
> >
> > We thank the reviewer to point out to us the paper [7] that we indeed did not cite. In our bibliography research, we did not
> > find older papers that present SNAG. Being able to identify and cite the seminal works is important so we
> > have added this citation in the revised version.

---

> > > ### Author Response · Authors · 2024-11-19
> > >
> > > + Proposition 2 suggests that RACOGA holding with $c>-0.5$ is sufficient to verify the SGC. But in Figure 1(a), SNAG does not accelerate over SGD despite the RACOGA values being greater than -0.12. Is there an explanation for this apparent discrepancy
> > >
> > > As the reviewer noticed, if having RACOGA holding with $c>-0.5$ is sufficient to verify the SGC, verifying SGC is not sufficient to have acceleration of SNAG over SGD. Remark 3 tells us that $\rho_K$ needs to be small enough to have that SNAG convergence bounds (Theorem 3) are better than SGD convergence bounds (Theorem 1). Importantly, according to our Section 4.2, in order to get $\rho_K$ small, we need to have RACOGA verified with $c$ large enough. On Figure 1.a, the fact that $c > -0.12$ is not sufficient to observe acceleration of SNAG over SGD. Therefore, the observed behaviour is consistent with our theoretical results.
> > >
> > > +  Just to confirm, in the experiments, were GD and NAG used with the full batch gradient at each step (e.g.
> > > were all of 50k images used for the CIFAR-10 experiment at each training step) ? If yes, this might be worth
> > > specifying explicitly since most of the times in machine learning experiments, NAG refers to Algorithm 8,
> > > even when it is used with mini-batch gradients.
> > >
> > > We thank the reviewer to highlight that our notations are not totally clear. GD and NAG were indeed used with
> > > full batch gradient at each step. For the linear regression experiments, we coded NAG ourselves (Algorithm
> > > 7). For the neural network experiments we used the PyTorch implementation. As mentioned in Section 5.2,
> > > NAG indeed refers to Algorithm 8. To make it clearer, when using Algorithm 8, we refer to Algorithm 3 full
> > > batch in the revised version of the caption of Figure 2.
> > >
> > > [1]  Gupta et al., Nesterov acceleration despite very noisy gradients, 2024.
> > >
> > > [2]  Sebbouh et al., Almost sure convergence rates for Stochastic Gradient Descent and Stochastic Heavy Ball, 2021.
> > >
> > > [3]  Mertikopoulos et al., On the Almost Sure Convergence of Stochastic Gradient Descent in Non-Convex Problems, 2020.
> > >
> > > [4]  Gupta et al., Nesterov acceleration despite very noisy gradients, 2024.
> > >
> > > [5]  Liu and  Yuan, On Almost Sure Convergence Rates of Stochastic Gradient Methods, 2022.
> > >
> > > [6] Bottou, Stochastic learning, Summer School on Machine Learning, pages 146-168, 2003.
> > >
> > > [7] Nesterov, Efficiency of Coordinate Descent Methods on Huge-Scale Optimization Problems, SIAM Journal on Optimization, volume 22, pages 341-362, 2012.

---

> ### Comment · Reviewer_zYBC · 2024-11-21
>
> Thanks for your response. I appreciate that the authors are receptive to feedback and have made a good effort to incorporate the feedback into their revision. I still have some doubts about the comparison between SGD and SNAG under different conditions. But now with the addition of remark 4 and appendix F, the authors are at least being more transparent about the nature of the comparison. It is interesting to see that without SGC, only a non-accelerated rate can be proved for SNAG. If the authors have the space for it, they can consider moving the discussion in Remark 9 to the main body of the article (this is only a suggestion and does not affect my recommendation for the paper).
>
> Nevertheless, I think that the overall contributions of the paper are interesting and useful enough. Most of my concerns were about the presentation, which have been addressed by the authors. I am happy to increase my score.

---

### Official Review · Reviewer_b9ZL · 2024-11-09

**Soundness:** 3
**Presentation:** 3
**Contribution:** 3
**Rating:** 6
**Confidence:** 3

**Summary:**

This paper studies the possibility of obtaining accelerated convergence of the Stochastic Nesterov Accelerated Gradient (SNAG) method. The authors provide a clear proof that the average correlation between gradients allows to verify the strong growth condition, which is essential for achieving accelerated convergence in convex optimization settings. Furthermore, the paper includes comprehensive numerical experiments in both linear regression and deep neural network optimization, empirically validating the theoretical findings. The experimental results are clear and concise. These contributions advance the understanding of momentum-based stochastic optimization techniques and demonstrate the practical effectiveness of SNAG in enhancing convergence rates.

**Strengths:**

1. Originality:
- Proposes the hypothesis that Stochastic Nesterov Accelerated Gradient (SNAG) can accelerate over Stochastic Gradient Descent (SGD) and proves that this hypothesis is valid when SNAG is under a Strong Growth Condition.
- Provides new asymptotic almost sure convergence results for SNAG.
- Gives the new characterization of the SGC constant by using the correlation between gradients.
- Introduces a new condition named Relaxed Averaged COrrelated Gradient Assumption (RACOGA).
2. Quality and clarity:
- Clearly shows when $f$ is convex and $\mu$-strongly convex, it shows the possibility of acceleration of SNAG over SGD is highly dependent on the SGC constant $\rho_K$, where $\rho_K < \sqrt{\frac{L^2_{(K)}}{\mu L}}$.
- Provides clear and explicit steps for proofs.
- The numerical results are readable and show a clear difference in convergence speed among different algorithms.
3. Significance:
- People can get faster and better results by applying the condition proposed in this paper.

**Weaknesses:**

* The text and formulas are a bit dense; the author can add a table to compare the convergence speed of SGD and SNAG under different conditions.
* The graphs look good. However, that would be better if the author gave more detail about the explanation for the graph, for example, what the "small values" of RACOGA mean on the graph.
* The colors in the right graph for Figure 1(a) are similar, author can use more contrasting colors.

**Questions:**

* While RACOGA has been demonstrated to facilitate the acceleration of SNAG over SGD in convex and strongly convex functions, how does RACOGA perform in non-convex optimization scenarios, such as those commonly found in deep neural network training? Can RACOGA be effectively applied to these more complex models, or are there additional considerations needed to achieve similar acceleration benefits?
* The paper highlights that large RACOGA values enable the acceleration of SGD with momentum. However, what practical methods or criteria can be used to identify or achieve large RACOGA values in real-world applications?
* How robust is SNAG's performance to variations in RACOGA across different types of datasets and optimization problems?

---

> ### Author Response · Authors · 2024-11-19
>
> Thank you for your positive feedback and careful reading of our paper. Please see below our detailed answers.
>
> + The text and formulas are a bit dense; the author can add a table to compare the convergence speed of SGD and SNAG under different conditions.
>
> We thank the reviewer for this suggestion. In order to make the comparison between SGD and SNAG under different conditions clearer we have added a table at the beginning of Section 3 (Table 1).
>
> + The graphs look good. However, that would be better if the author gave more detail about the explanation for the graph, for example, what the "small values" of RACOGA mean on the graph.
>
> We are glad that you appreciate our graphs. We also thank the reviewer for his suggestion to bring more clarity to our paper. We propose a new revised version where we add some details. First, we introduce the parameter $\lambda$ (Figure 1) in the main text. Then, we display a revised version of the histogram of RACOGA values (same scale in both sub-figures) in order to show that we mean "small values of RACOGA" in Figure 1a compared to the correlated case (Figure 1b).
>
> + The colors in the right graph for Figure 1(a) are similar, author can use more contrasting colors.
>
> We thank the reviewer for his feedback about Figure 1. In the revised version, we propose a new set of colors in order to make this figure clearer.
>
> + While RACOGA has been demonstrated to facilitate the acceleration of SNAG over SGD in convex and strongly convex functions, how does RACOGA perform in non-convex optimization scenarios, such as those commonly found in deep neural network training? Can RACOGA be effectively applied to these more complex models, or are there additional considerations needed to achieve similar acceleration benefits?
>
> We thank the reviewer for this question. Indeed, it is crucial to derive from this work future theoretical results that could be applied to realistic machine learning optimization tasks.
>
> First, the possibility to achieve acceleration relies heavily on geometrical assumptions of the function. Even in a deterministic setting, gradient descent is optimal in some cases, as for instance in the case of  $L$-smooth functions [1].
>
> It is therefore critical to find relevant geometrical properties that may relax convexity, and that allow to demonstrate an acceleration of SNAG over SGD. It is known that for some classes of non convex functions such as quasar convex functions [2] or functions with Lipschitz gradient and Lipschitz Hessian [3], momentum type algorithms allow to theoretically achieve faster convergence in a deterministic setting.
>
> Finally, we study in this work deep learning classification trainings (Figure 2), which is  a non-convex optimization problem, in order to get intuition of the possibility to extend RACOGA tools to this non-convex setting. We observe in Figure 2, that large RACOGA leads to faster convergence of SNAG over SGD. This suggests empirically that RACOGA could be useful even in this non-convex setting. We leave for future works the extension of   our theoretical results in this non-convex setting with the appropriate geometrical properties.
>
> + The paper highlights that large RACOGA values enable the acceleration of SGD with momentum. However, what practical methods or criteria can be used to identify or achieve large RACOGA values in real-world applications ?
>
> It is a very interesting and open question. As mentioned in our Remark 7, the authors of [4] show that when considering neural networks, some mechanisms such as increasing the width of the layers allows to increase the gradient confusion, a quantity which is related to RACOGA.
>
> However, the question of designing mechanisms that may exhibit positive correlation, quantified with RACOGA, is difficult (we discuss the linear case in our Appendix H), and it is still an open question as far as we know.
>
> In this work, we introduce RACOGA as a theoretical tool to understand the possibility of accelerating SGD with SNAG, but more works need to be done to make it a truly practical tool.
>
>  Finally, note that it appears with our experiments in Figure 2 show that classical neural network architectures such as MLP or CNN lead to high RACOGA values. This suggests that these architectures naturally produce high RACOGA values.

---

> ### Author Response · Authors · 2024-11-19
>
> + How robust is SNAG's performance to variations in RACOGA across different types of datasets and optimization problems?
>
> Again, it is a very interesting and open question. Identifying variations of RACOGA's behaviour across different datasets and optimization problems could help create link between model properties and appropriate optimizers. In all our experiments, both in linear regression (two datasets) and in deep learning classification (two datasets), we observe that large values of RACOGA allow to accelerate SGD with SNAG. Moreover, in the convex setting (linear regression), we show an example where low values of RACOGA lead to SGD being not accelerated with momentum. In our revised version, we added Figure 8 in Appendix A, where we plot the RACOGA values and convergence curves of SGD and SNAG for other classification datasets (MNIST, FashionMNIST, KMNIST, EMNIST), trained with a CNN. RACOGA values remain high for all datasets.
>
> [1]  Carmon et al., Lower Bounds for Finding Stationary Points I, 2019.
>
> [2]  Hinder et al., Near-Optimal Methods for Minimizing Star-Convex Functions and Beyond, 2023.
>
> [3]  Arjevani et al., Lower Bounds for Non-Convex Stochastic Optimization, 2022.
>
> [4]  Sankararaman et al. The Impact of Neural Network Overparameterization on Gradient Confusion and Stochastic Gradient Descent, 2020.

---

> > ### Author Response · Authors · 2024-11-21
> >
> > In order to investigate more deeply the following question of the reviewer :
> > + How robust is SNAG's performance to variations in RACOGA across different types of datasets and optimization problems?
> >
> > We added new numerical experiments to a new revised version. On Figure 4, we run GD, SGD, SNAG and NAG to solve a linear regression problem on a real world dataset, which allows to test the behaviour of the algorithms and the associated RACOGA values in an under-parameterized regime (outside our theory) where SNAG appears to be the fastest algorithm. On Figure 10, we study the behaviour of SNAG and SGD, together with RACOGA values, when performing logistic regression (non-convex) on the CIFAR-10 dataset, where both algorithms have the same convergence rate. This might be due to the non-convexity or low RACOGA values.

---

> > > ### Comment · Reviewer_b9ZL · 2024-11-25
> > >
> > > Thank you for the clear response. I will keep my score.

---

> > > > ### Author Response · Authors · 2024-11-25
> > > >
> > > > Thank you for your answer. We thought we took into account your remarks and questions, and we modified the paper accordingly (Table 1, new design of Figure 1). However your recommendation is still not positive about our paper. Could you please detail what are the remaining weaknesses of the paper ? We will be glad to answer your concerns.

---

> > > > > ### Comment · Reviewer_b9ZL · 2024-11-26
> > > > >
> > > > > Thank you for your reply. After consideration, I will increase my score and recommend to accept this paper.

---

> ### Comment · Area_Chair_V8cJ · 2024-11-24
>
> Dear Reviewer b9ZL,
>
> The author discussion phase will be ending soon. The authors have provided detailed responses. Could you please reply to the authors whether they have addressed your concern and whether you will keep or modify your assessment of this submission?
>
> Thanks.
>
> Area Chair

---

### Author Response · Authors · 2024-11-19
**Response to all reviewers**

We thank all the reviewers for their careful reading and all their comments. A new version of the paper has been loaded with modifications. These modifications appear in blue in the revised version.

The contributions of our work can be summarized as follows:


  1. We give a new characterization of the Strong Growth Condition (SGC) constant by using the correlation between gradients, quantified by RACOGA (Propositions 1-2), and we exploit this link to study the efficiency of SNAG.
  2. Using our framework, we study the theoretical impact of batch size on the algorithm performance, depending on the correlation between gradients (Theorem 5).
  3. We complete convergence results of [1,2] with new almost sure convergence rates (Theorem 4).
  4. We provide numerical experiments that show that RACOGA is a key ingredient to have good performances of SNAG compared to SGD.

[1] Vaswani et al. Fast and faster convergence of sgd for over-parameterized models and an accelerated perceptron, 2019

[2] Kanan et al. Achieving acceleration despite very noisy gradients,  2023.

---

### Meta-Review · Area_Chair_V8cJ · 2024-12-08

**Metareview:**

The paper shows that positive gradient correlation ensures  strong growth condition (SGC) for finite-sum functions. It introduces the RACOGA condition, linking SGC to accelerated convergence of Stochastic Nesterov Accelerated Gradient(SNAG). The results clarify why momentum-based stochastic methods perform well, supported by insightful numerical experiments, and thus make good contributions to optimization community.

**Additional Comments On Reviewer Discussion:**

I gave a lower weight on Reviewer b9ZL's opinion because I know this reviewer personally and he/she is junior PhD student and has not published any work on this area.


Reviewer zYBC had several concerns, most of which were about the presentation and have been addressed by the authors.

Reviewer wrcq expressed a concern on the (computational) limitation of the RACOGA condition, which is a minor issue in my opinion.

Reviewer Dovy had a concern that the proof in this paper uses a previous result (from [2]) which may reduce its novelty. However, I agree with the reviewer that the result used is mainly for proving the almost sure convergence. The main contribution, namely, RACOGA leading to accelerated, is still novel.

Three reviewers raised their scores after the rebuttal.

---

### Decision · Program_Chairs · 2025-01-22

Accept (Poster)